

# Top-down and bottom-up: Studying the SMEFT beyond leading order in $1/\Lambda^2$

Tyler Corbett

Faculty of Physics, University of Vienna, Boltzmanngasse 5, A-1090 Wien, Austria

corbett.t.s@gmail.com

## Abstract

In order to assess the relevance of higher order terms in the Standard Model effective field theory (SMEFT) expansion we consider four new physics models and their impact on the Drell Yan cross section. Of these four, one scalar model has no effect on Drell Yan, a model of fermions while appearing to generate a momentum expansion actually belongs to the vacuum expectation value expansion and so has a nominal effect on the process. The remaining two, a leptoquark and a $Z'$ model exhibit a momentum expansion. After matching these models to dimension-ten we study the how the inclusion of dimension-eight and dimension-ten operators in hypothetical effective field theory fits to the full ultraviolet models impacts fits. We do this both in the top-down approach, and in a very limited approximation to the bottom up approach of the SMEFT to infer the impact of a fully general fit to the SMEFT. We find that for the more weakly coupled models a strictly dimension-six fit is sufficient. In contrast when stronger interactions or lighter masses are considered the inclusion of dimension-eight operators becomes necessary. However, their Wilson coefficients perform the role of nuisance parameters with best fit values which can differ statistically from the theory prediction. In the most strongly coupled theories considered (which are already ruled out by data) the inclusion of dimension-ten operators allows for the measurement of dimension-eight operator coefficients consistent with theory predictions and the dimension-ten operator coefficients then behave as nuisance parameters. We also study the impact of the inclusion of partial next order results, such as dimension-six squared contributions, and find that in some cases they improve the convergence of the series while in others they hinder it.

| | |
|---|---|
| Received | 16-05-2024 |
| Accepted | 15-07-2024 |
| Published | 14-08-2024 |

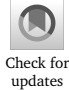
# 1  Introduction

The Standard Model effective field theory (SMEFT) has become one of the most important methodologies for studying physics beyond the Standard Model (SM) at the LHC. The SMEFT is formed on the fundamental principle that, so long as new resonances are sufficiently heavy, effective field theories yield the most general possible $S$-matrix consistent with the tenets of quantum field theory [1]. This powerful statement comes with some caveats, for example: there exists a region of validity based on the power counting and there is no reason (beyond aesthetics and limited experience) that the heavy physics imprints on the leading operators of the expansion.

In the field of the SMEFT, most studies are performed at dimension six or order $1/\Lambda^2$ where $\Lambda$ is the heavy scale of new physics. However, there has been substantial interest in understand beyond leading order effects in the SMEFT (or $1/\Lambda^2$) expansion. In some cases this can be because of unique signals first generated beyond leading order, for example triple neutral gauge couplings [2–7]. It has also been pointed out that many models generate similar dimension-six EFTs, but this degeneracy is broken beyond leading order in the SMEFT expansion [8]. This has motivated a shift toward including dimension-eight effects in the SMEFT which has resulted in many tools for calculation [9–13] as well as many phenomenological studies [14–36].

A natural question is: how relevant are terms of order $1/\Lambda^4$? Much of the community neglects them arguing the new physics scale is sufficiently high their effects are negligible. This approach is also pragmatic, in that it allows for dealing with far fewer parameters in a fit as there are already a seemingly intractable number of parameters at dimension-six [37]. This can be made tractable with assumptions such as minimal flavor violation [38] or flavor universality [39], which presents some opportunity to begin consistent studies of the SMEFT to order $1/\Lambda^4$, e.g. a fit including dimension-eight operators in [25]. If or when valid, truncation at order $1/\Lambda^2$ offers further appeal – one can embrace the "Energy helps accuracy" paradigm [40] where perceived growth of matrix elements due to the presence of dimension-six operator effects allows for more stringent constraints from data. This naturally brings us back to the

question posed at the beginning of this paragraph. Is it consistent to use high energy events to further constrain the parameters of the SMEFT at order $1/\Lambda^2$? An expansion in a large scale, while using high energy data, may naturally break down. An example of a more limited study of the breakdown of a top-down SMEFT analysis at dimension-six can be found in [41]. However, this study did not discuss how including higher order operators impacts a fit or the inclusion of additional operators in the IR which are not generated by the UV which we will perform below. Other studies related to the convergence of vev expansion in the scalar singlet extension of the SM can be found in for example [42] (SMEFT vs nonlinear EFTs) or [43] (D6 vs D8 SMEFT).

This article seeks to explore some concrete ultraviolet (UV) extensions of the SM, their imprint on the SMEFT, and the possible breakdown of the expansion specifically for the Drell Yan process at the LHC. As a study to dimension-eight naturally includes dimension-six-squared contributions, we also explore the reliability of fits that include the squares of dimension-six operators.

This article is organized as follows, in Section 2 we outline four new physics models of a single new field (possibly a multiplet of the SM gauge group), in Section 3 we derive the EFT for these models up to dimension ten ($1/\Lambda^6$), in Section 4 we consider how well the IR model at a given mass dimension compares to the UV model prediction, and in Section 5 we give our conclusions. The appendices include discussions of the $U(1)$ mixing model considered in this article, a parameterization of the SMEFT Drell Yan cross section, and additional tables. The ancillary files include Mathematica notebooks employing Matchete [44] to derive the effective Lagrangians used in this work.

In addition to the topics discussed in this article, we note that the foundation of any analysis at a hadronic collider are the pdfs. The pdf determination can hide the UV dynamics as state of the art pdf determinations *include* LHC data [45–49].

## 2 The models in the UV

We consider four different ultraviolet models that impact the Drell-Yan process. These include two scalar models, $\phi$ and $\Phi$, a fermion $\chi$, and a vector $V$. These models are elaborated after we establish our notation through reviewing the SM fields and Lagrangian.

For each model the Feynman Rules are generated using FeynRules [50], and the Drell Yan process is calculated using FeynArts and Formcalc [51, 52], then integrated in invariant mass bins for the final state leptons using the Vegas algorithm against the NNPDF3.0 NLO parton distribution functions (pdfs) with $\alpha_s = 0.118$. The factorization and renormalization scales are taken to be the central value of a given invariant mass bin. We assume a 13 TeV LHC, invariant mass bins are chosen according to the CMS Drell-Yan search with 140/fb integrated luminosity [53]. Care is taken to conform to the $\{\alpha, G_F, m_Z\}$ input parameter scheme, however this has a negligible effect on our results as the large mass of new particles required by the EFT approach requires mixing with the SM to be small.

### 2.1 The Standard Model

For clarity we briefly introduce the field content and the Lagrangian of the SM. The SM scalar and fermion fields and charges used in this article are:

$$H \sim (1,2)_{\frac{1}{2}},$$

$$L \sim (1,2)_{-\frac{1}{2}}, \qquad Q \sim (3,2)_{\frac{1}{6}}, \tag{1}$$

$$e \sim (1,1)_{-1}, \qquad d \sim (3,1)_{-\frac{1}{3}}, \qquad u \sim (3,1)_{\frac{2}{3}}.$$

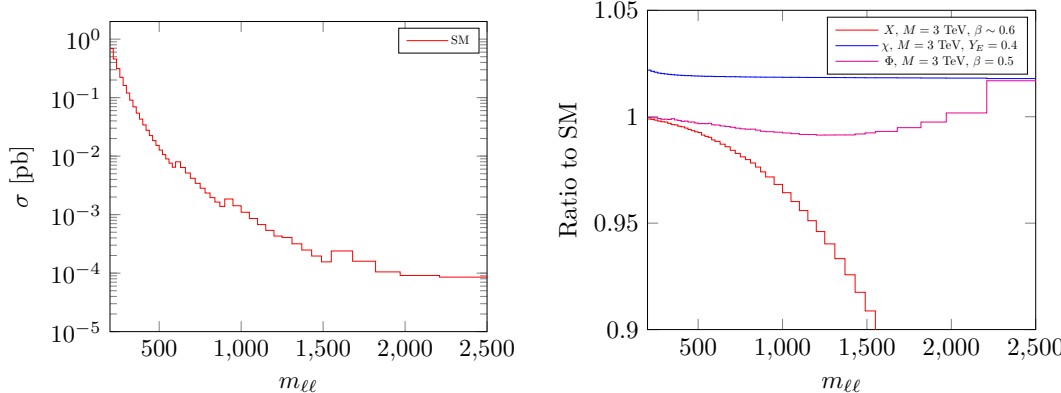

Figure 1: Cross section as a function of invariant mass bin for the SM. The ratio of the UV models with $\Phi$, $\chi$, or $X$ to the SM. Assumed mass and coupling for the UV model is given in the legend. In the SM plot, small jumps in the cross section are due to a change in the bin widths used in [53].

Implicit in the above notation is that the fermionic $SU(2)_L$ doublets are left handed, and the fermionic singlets are right handed. For simplicity we will neglect fermion masses, and therefore set the SM Yukawa couplings to zero. The SM Lagrangian is then given by:

$$
\begin{aligned}
\mathcal{L}_{\text{SM}} = &-\frac{1}{4}G^A_{\mu\nu}G^{A,\mu\nu} - \frac{1}{4}W^I_{\mu\nu}W^{I,\mu\nu} - \frac{1}{4}B_{\mu\nu}B^{\mu\nu} \\
&+ i\bar{L}\slashed{D}L + i\bar{Q}\slashed{D}Q + i\bar{e}\slashed{D}e + i\bar{d}\slashed{D}d + i\bar{u}\slashed{D}u \\
&+ (D_\mu H)^\dagger(D^\mu H) + \mu^2(H^\dagger H) - \lambda(H^\dagger H)^2\,.
\end{aligned}
\tag{2}
$$

Where the $B$, $W$, and $G$ fields are the familiar gauge fields of $U(1)_Y$, $SU(2)_L$, and $SU(3)_C$ with gauge coupling constants $g_1$, $g_2$, and $g_3$.

For the SM process calculation we use the input parameters used in previous studies of dimension-eight effects in the SMEFT [24, 29]:

$$
\alpha = \frac{1}{137.035999084(1-\Delta_\alpha)}\,,
\tag{3}
$$

$$
\Delta\alpha = 0.0590\,,
\tag{4}
$$

$$
G_F = 1.1663787 \cdot 10^{-5}/\text{GeV}^2\,,
\tag{5}
$$

$$
m_Z = 91.1876\,\text{GeV}\,.
\tag{6}
$$

The cross section for the tree-level SM Drell Yan process as a function of invariant mass bin is shown in Figure 1.

## 2.2 Scalar $\phi$

Next we consider the scalar $\phi$ with quantum numbers $(1,3)_0$ (referred to as $\Xi$ in [54]). This model was chosen as at dimension-six it only generates operators which result in finite renormalizations of the SM vertices contributing to Drell Yan and the $Z$ boson mass.

In this model we have, in addition to $\mathcal{L}_{\text{SM}}$ the terms:

$$
\Delta\mathcal{L}_\phi = \frac{1}{2}(D_\mu\phi^a)^\dagger(D_\mu\phi^a) - \frac{1}{2}M^2(\phi^a)^2 + \kappa H^\dagger\sigma^a H\phi^a - \lambda_{\phi H}(\phi^a)^2(H^\dagger H) - \lambda_\phi(\phi^a)^4\,.
\tag{7}
$$

As the heavy scalar $\phi$ does not couple directly to fermions, this model is a trivial example, i.e. it will not have any affect on Drell Yan. This is still a meaningful example of when NP will not affect a given process, albeit less interesting than those which follow. As the Drell Yan process is unaffected no plot for this model is included in Figure 1.

### 2.3 Scalar $\Phi$

We consider a scalar $\Phi$ with quantum numbers $(3,2)_{\frac{1}{6}}$, this is the $\Pi_1$ of [54]. This model was chosen as at leading order in the SMEFT it generates just one four-fermion operator.

We write the terms additional to the SM Lagrangian as,

$$\Delta\mathcal{L}_\Phi = (D_\mu\Phi)^\dagger(D_\mu\Phi) - M^2\Phi^\dagger\Phi + Y_\Phi\left[\bar{d}\left(\Phi\, i\sigma_2 L\right) + h.c.\right]. \tag{8}$$

For convenience we take $Y_\Phi$ to be real, but this need not be the case. As the $\Phi$ particle doesn't mix with the SM particle content the input parameters are the same as in the SM, plus the new parameter $Y_\Phi$. In Figure 1 we show the ratio of the cross section in the presence of a 3 TeV $\Phi$ with Yukawa coupling $Y_\Phi = 0.5$ to the SM cross section. The scalar $\Phi$ contributes to the process in the $t$–channel, and we can see from the plot that the disparity from the SM prediction is most pronounced in the highest invariant mass bins. That is, it contributes to the momentum expansion of the SMEFT.

### 2.4 Fermion $\chi$

We consider a vector-like fermion $\chi$ with quantum numbers $(1,1)_{-1}$ ($E$ of [54]). This model is chosen as it generates only "Class 7" operators ($\psi^2 H^2 D$) at dimension-six.

In addition to the SM Lagrangian we have:

$$\Delta\mathcal{L}_\chi = i\bar{\chi}\slashed{D}\chi - M\bar{\chi}\chi - Y_\chi\left[H^\dagger\bar{\chi}L + h.c.\right]. \tag{9}$$

Here we have the complication that the $\chi$ and $L$ fields mix, causing a shift in the definition of $G_F$. This shift is suppressed by $1/M^2$ and is numerically negligible. We include this shift, nonetheless, and Fig. 1 shows a plot for a 3 TeV $\chi$ with Yukawa coupling $Y_\chi = 0.4$. Notice the distinct difference from the case of $\Phi$. The largest difference here occurs in the low invariant mass range, for higher $m_{\ell\ell}$ the shift is approximately constant. This is because we have simply shifted the coupling to the leptons by a constant value. In the discussion of the theory in the infrared we will see this is directly attributable to the dimension six operator $(H^\dagger\overleftrightarrow{D}_\mu H)(i\bar{L}\gamma^\mu L)$, and all other operators do not contribute (in the $m_\ell = 0$ limit).

### 2.5 Vector $X$

Finally, we consider an additional gauge boson $X$. We begin with a vector $V$ transforming under a new $U(1)$ gauge symmetry, which will, upon diagonalizing the mass matrix, result in the vector $X$ with which we will work. We do not assign any charge to the SM fermions under this new $U(1)$ and therefore the vector's only interaction with the SM is through kinetic mixing with the SM $B$ field. This model is of interest as at each order in the heavy mass expansion it only generates operators of the form $(\partial_\mu B_{\mu\nu})\partial^{2n}(\partial_\rho B^{\rho\mu})$. These are the $(2n+4)$ dimensional analogues of the $\hat{Y}$ operator of [40]. The Lagrangian for this model is:

$$\Delta\mathcal{L}_V = -\frac{1}{4}V_{\mu\nu}V^{\mu\nu} + \frac{1}{2}M^2 V_\mu V^\mu - \frac{k}{2}B_{\mu\nu}V^{\mu\nu}. \tag{10}$$

Making appropriate field redefinitions (See App. A) results in the alternate UV Lagrangian:

$$\Delta \mathcal{L}_X = -\frac{1}{4} X_{\mu\nu} X^{\mu\nu} + \frac{1}{2} M_X^2 X_\mu X^\mu - g_1 Y_H \beta (H^\dagger i \overleftrightarrow{D}_\mu H) X^\mu + g_1^2 Y_H^2 \beta^2 (H^\dagger H) X_\mu X^\mu$$
$$- g_1 \sum_\psi Y_\psi \beta (\bar{\psi} \gamma_\mu \psi) X^\mu \,, \tag{11}$$

where $Y_H = 1/2$ is the Higgs hypercharge, $Y_\psi$ is the hypercharge for a given SM fermion field $\psi$, and:

$$\beta = \frac{-k}{\sqrt{1-k^2}} \,, \tag{12}$$

$$M_X = \frac{M_V}{\sqrt{1-k^2}} \,. \tag{13}$$

Note that part of this Lagrangian comes from transforming the $B$ field in $\mathcal{L}_{\text{SM}} + \Delta\mathcal{L}_V$ and the $B$ field is not the "same" $B$ field as in $\mathcal{L}_{\text{SM}} + \Delta\mathcal{L}_X$ when used. It is much easier to work with the EFT resulting from Eq. 11 as this greatly reduces the number of operators which induce unphysical poles in scattering amplitudes.[1] This is discussed more in the IR section below.

In this case, there is significant mixing with the SM and the input parameters are difficult to determine. We numerically solved for the corrections to input parameter relations and found they were again negligible. Nonetheless we include them in the calculation of the cross section in the UV. An example of the ratio of UV to SM cross sections for $M_X = 3$ TeV with the value of the parameter controlling the mixing $k \sim -0.5$ ($\beta = 0.6$) is shown in Fig. 1. We note that this model results in by far the largest deviation from the SM prediction, with this example resulting in over 15% corrections in the highest invariant mass bins.

## 3 The models in the IR

In this section we discuss the matching of the models of Sec. 2 to order $1/\Lambda^6$. The state of the art for matching onto the SMEFT is generally dimension-eight. Examples include [22, 24, 30, 56, 57]. The process of matching at tree level to *an* effective Lagrangian beyond leading order in the EFT expansion is in general not particularly difficult. However, the theory community has focused on rewriting these effective Lagrangians in terms of non-redundant bases by utilizing Integration By Parts (IBP) relations and field redefinitions. This step is fundamental to the bottom up approach embraced in phenomenological searches for beyond the SM physics as it removes redundancies in the operator basis and allows for a unified comparison between various groups' analyses. These steps are tedious and time consuming, rendering the process of matching even to dimension eight largely impractical.

However, for the purpose of this article we are particularly interested in understanding the implications of missing orders in the SMEFT power counting. As such we instead only make use of IBP identities to simplify our calculation of the Drell Yan cross section as much as possible.

Below we present the results of matching these models to the SMEFT up to order $1/\Lambda^4$. The matching has been performed to order $1/\Lambda^6$, however the resulting effective Lagrangians are generally not well suited for publication and so are relegated to the ancillary Mathematica notebooks. The matching was performed by hand following the procedure in [58], but checked

---

[1]For a discussion of how non-derivative field redefinitions in the UV relate to derivative dependent field redefinitions in the IR see [55].

using the `Matchete` package [44].[2] Integration by parts identities, when used, are applied using `Matchete`, these were not checked by hand due to the sheer scope of the number of identities required for some models.

In what follows we only write operators which affect the Drell Yan process. IBP identities are used in an effort to distribute derivatives across fields. For example,

$$
\begin{aligned}
&- (D_\mu H)^\dagger H (D_\mu D^2 H)^\dagger H + h.c. \\
&= \left[ H^\dagger (D^2 H) H^\dagger (D^2 H) + (D_\mu H)^\dagger (D^2 H) H^\dagger (D^\mu D^2 H) + H^\dagger (D^2 H)(D_\mu H)^\dagger (D^\mu H) + h.c. \right] .
\end{aligned}
\tag{14}
$$

Notice that after applying the IBP identity above we can neglect operators involving 3 or more cases of derivatives of the Higgs field (or in other cases also field strength tensors), as they require three or more bosons in an effective vertex and therefore will not contribute to the Drell Yan process at tree level. So for the example above, only the first term needs to be retained. This is also the method which allows the geoSMEFT to elaborate *all* operators which contribute to two- and three-point functions and derive certain results to all orders in the SMEFT power counting [10].

Predictions were made following the same routine outlined at the beginning of Sec. 2. In the case of the $X$ field, calculations involving four-fermion operators were performed by hand as Feynarts/Feyncalc remains unable to implement the Feynman rules for four-fermion operators with vector currents. In this section we give some interesting benchmark examples, primarily focusing on when dimension-six terms fail to correctly reproduce the $m_{\ell\ell}$ distribution. We leave a more general exploration of the model parameter spaces for the next section.

To clarify our nomenclature, we write a general squared amplitude in the IR as:

$$
|\mathcal{M}|^2 = |\mathcal{M}_{SM}|^2 + 2 \frac{c_6}{\Lambda^2} |\mathcal{M}_{SM} \mathcal{M}_6| + \frac{1}{\Lambda^4} \left( c_6^2 |\mathcal{M}_6|^2 + 2 c_8 |\mathcal{M}_{SM} \mathcal{M}_8| \right) + \cdots .
\tag{15}
$$

We will generally refer to the leading term as the dimension-six term and occasionally as the order $1/\Lambda^2$ term. The order $1/\Lambda^4$ term includes the dimension-six squared term ($c_6^2$) as well as the dimension eight term arising from the interference of the $1/\Lambda^4$ amplitude with the SM. We will also refer to results to order $1/\Lambda^4$ as "(up) to dimension eight." Similarly, but not written above, the $1/\Lambda^6$ terms include dimension-six amplitudes interfering with dimension-eight as well as dimension-ten amplitudes interfering with the SM amplitude. In general there are other contributions, such as three insertions of dimension-six operators in an amplitude interfering with the SM. We make simplifying assumptions below which remove these additional terms from consideration. We refer to calculations to order $1/\Lambda^6$ in analogously to those to order $1/\Lambda^4$.

**Scalar $\phi$**

This theory only affects the Drell-Yan process through shifts in the $Z$-mass. However, we use the $Z$-mass as an input parameter so these effects are absorbed into the definition of the $Z$-mass and result in shifts in other vertices which do not contribute to Drell Yan. Because of its relative

---

[2]While `Matchete` is largely known for its utility in matching at one-loop, it is also an extremely powerful tool for tree-level matching to higher orders in the EFT expansion as well as for IBP relations.

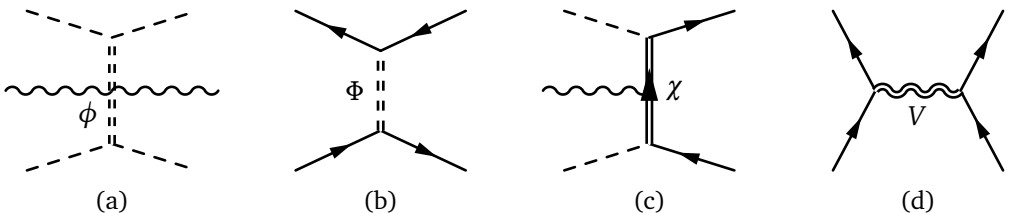

Figure 2: Example diagrams of the how the UV imprints on the IR for each model. These diagrams should be understood as an implicitly incomplete set, i.e. most diagrams are missing, and only used as qualitative guide. External vectors are sourced, in these cases, by covariant derivatives. The external Higgses in the diagram should be understood as $\langle v \rangle$ when contributing to the Drell-Yan process.

simplicity we can show the full result through dimension ten. The resulting Lagrangian is:

$$
\begin{aligned}
\mathcal{L}_{\text{IR}}^{\phi} = \mathcal{L}_{\text{SM}} &- \frac{\kappa^2}{M^4}\left[\frac{1}{2}Q_{HD} - \frac{1}{4}Q_{HD2}\right] - \frac{\kappa^2}{8M^4}\left[|H|^2(H^\dagger D^2 H) + h.c.\right] \\
&+ \frac{2\lambda_{\phi H}\kappa^2}{M^6}|H|^2 Q_{HD} - \frac{\lambda_{\phi H}\kappa^2}{M^6}|H|^2 Q_{HD2} + \frac{\lambda_{\phi H}\kappa^2}{2M^6}\left[|H|^4(H^\dagger D^2 H) + h.c.\right] \\
&+ \frac{\kappa^2}{M^8}\left[\frac{\lambda_\phi \kappa^2}{M^2} - 6\lambda_{\phi H}^2\right]|H|^4 Q_{HD} + \frac{\kappa^2}{6M^8}\left[16\lambda_{\phi H}^2 - 3\frac{\lambda_\phi \kappa^2}{M^2}\right]|H|^4 Q_{HD2} \\
&+ \frac{\kappa^2}{M^8}\left[\frac{\lambda_\phi \kappa^2}{4M^2} - \frac{5\lambda_{\phi H}^2}{3}\right]\left[|H|^6(H^\dagger D^2 H) + h.c.\right],
\end{aligned}
\tag{16}
$$

where we have defined:

$$
Q_{HD} = (H^\dagger D_\mu H)(D^\mu H)^\dagger H, \tag{17}
$$
$$
Q_{HD2} = (H^\dagger H)(D^\mu H)^\dagger (D_\mu H). \tag{18}
$$

Note that $Q_{HD2}$ is generally removed from the Warsaw basis in favor of $(H^\dagger H)\square(H^\dagger H)$. This serves as an example where there is no effect on the Drell Yan process. If nature realizes the $\phi$ model, then from a bottom up perspective we would be able to consistently use energy dependent distributions in the Drell Yan channel to constrain dimension-six operators in the SMEFT.

**Scalar $\Phi$**

This model exhibits the derivative expansion of the SMEFT very nicely as there are no Higgs boson dependent operators. It is important to note that $(\bar{d}L)$ is not gauge invariant, only the full product $(\bar{d}L)(\bar{L}d)$ is, otherwise one might attempt to simplify the derivatives into a form like $(\bar{d}L)\square(\bar{L}d)$ instead of $(\bar{d}L)D^2(\bar{L}d)$.

$$
\begin{aligned}
\mathcal{L}_{\text{IR}}^{\Phi} = \mathcal{L}_{\text{SM}} &+ \frac{Y_\Phi^2}{M^2}(\bar{d}L)(\bar{L}d) \\
&+ \frac{Y_\Phi^2}{M^4}\left[(\bar{d}D_\mu L)(\bar{L}D^\mu d) + (D_\mu \bar{d})L(\bar{L}D_\mu d) + (\bar{d}D_\mu L)(D_\mu \bar{L})d + (D_\mu \bar{d})L(D_\mu \bar{L})d\right].
\end{aligned}
\tag{19}
$$

The matching to dimension 10 can be found in the ancillary files. The result amounts to distributing four covariant derivatives among the four fermions of the dimension-six operator. Use of the Fierz identity,

$$
(\bar{\psi}_1 P_L \psi_2)(\bar{\psi}_3 P_R \psi_4) = -\frac{1}{2}(\bar{\psi}_1 \gamma_\mu P_R \psi_4)(\bar{\psi}_3 \gamma^\mu P_L \psi_2), \tag{20}
$$

recovers the dimension-six matching result of [54] which makes use of the Warsaw basis.

Performing the calculation in the UV, and for simplicity of writing the expressions, taking the limit $m_Z \to 0$, and denoting the partonic center of mass energy $\hat{s}$, we find:

$$\sigma(\bar{d}d \to e^+e^-) = \sigma_{\text{SM}}(\bar{d}d \to e^+e^-) - \alpha Y^2 \frac{\hat{s}(\hat{s} - 2M^2) + 2M^4 \log\left(1 + \frac{S}{M^2}\right)}{48N_c c_W^2 \hat{s}^3}$$

$$+ Y^4 \frac{\hat{s}(\hat{s} + 2M^2) - 2M^2(\hat{s} + M^2) \log\left(1 + \frac{S}{M^2}\right)}{64\pi N_c \hat{s}(\hat{s} + M^2)}. \tag{21}$$

In contrast, in the EFT we obtain:

$$\sigma(\bar{d}d \to e^+e^-) = \sigma_{\text{SM}}(\bar{d}d \to e^+e^-) - \frac{Y^2}{M^2} \frac{\alpha}{72\pi N_c c_W^2} \Delta_6 + \frac{Y^2}{M^4} \frac{\hat{s}\left(2\alpha\pi\Delta_8 + Y^2 c_W^2 \Delta_6^2\right)}{192\pi N_c c_W^2}$$

$$- \frac{Y^2}{M^6} \frac{\hat{s}^2\left(16\alpha\pi\Delta_{10} + 15Y^2 c_W^2 \Delta_6 \Delta_8\right)}{1920\pi N_c c_W^2}, \tag{22}$$

where we have denoted the contribution from the insertion of any dimension-d operator by $\Delta_d$. Notice that each subsequent order in the series essentially corrects the growth in $\hat{s}^n$ by a term with the opposite sign with growth in $\hat{s}^{n+1}$. That the order-by-order contributions contribute with opposite signs can be understood from the number of derivatives present at each order and that they source a factor of $ip^\mu$ in a given Feynman rule, i.e. a factor of 1 for dimension six, $-p^2$ at order $1/\Lambda^4$, and $p^4$ at $\mathcal{O}(1/\Lambda^6)$. Therefore, we expect that a fit to only dimension six operators will misestimate the UV model as $\hat{s}$ increases. The degree to which this occurs is controlled by the size of $M$ and $Y$. We also note the inclusion of the square of dimension-six amplitudes in calculations, *for this specific model,* while neglecting the dimension-eight contribution results in an opposite sign term which corrects the dimension-six term and results in better agreement as $\hat{s}$ grows. This will again fail at some $\hat{s}$ where the order $1/\Lambda^6$ terms are needed to again correct the growth.

Figure 3 shows three example plots of the convergence of the EFT expansion for $M_\Phi = 3$ TeV and $Y_\Phi = 0.5$ and 1.0, as well as the higher mass $M_\Phi = 7$ TeV with $Y_\Phi = 1.0$. These plots are chosen deliberately to show examples where the dimension-six prediction fails by more than 5% in the highest invariant mass bins. The plots nicely show how the opposite sign contribution at a given order $1/\Lambda^{2n+2}$ corrects for the growth at order $1/\Lambda^{2n}$, but eventually overcorrects requiring the order $1/\Lambda^{2n+4}$.

It is interesting to notice that the dimension-six squared contribution, which is of the same order as, but neglects the dimension-eight operators' contributions, outperforms the complete order $1/\Lambda^4$ contributions. We note that in the cases where $M_\Phi = 3$ TeV, the expansion fails to reproduce the full model result to better than 5% even when the dimension-ten contributions are considered. Unsurprisingly, we find as $M_\Phi$ tends to infinity, or $Y_\Phi$ to zero, the agreement between the dimension-six prediction and the UV model converge.

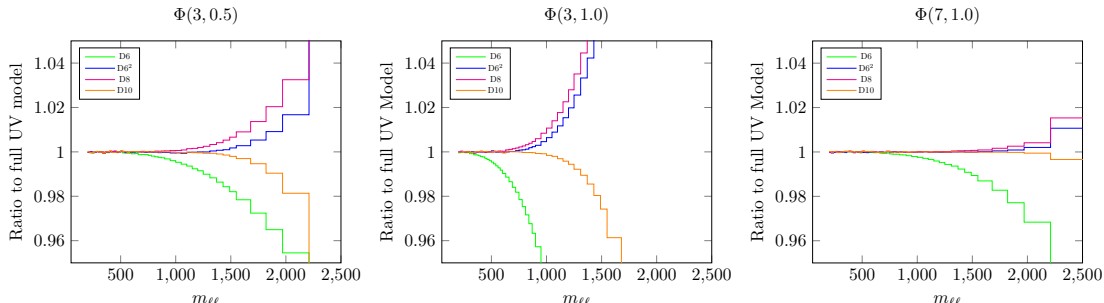

Figure 3: Ratios of the IR theory cross section prediction at a given order in the EFT expansion to the full UV model predictions. The data is binned in invariant mass of the leptons $m_{\ell\ell}$ following the CMS analysis found in [53]. The titles, $\Phi(M, Y_\Phi)$, indicate the mass $M$ in TeV and the Yukawa coupling of $\Phi$ to the SM fields, $Y_\Phi$.

## Fermion $\chi$

Integrating out the heavy fermion using the covariant derivative expansion, to dimension-eight, we find:

$$
\begin{aligned}
\mathcal{L}_{\text{IR}}^{\chi} = \mathcal{L}_{\text{SM}} &+ i\frac{Y_\chi^2}{2M^2}\Big[(H\bar{L})\gamma_\mu(D_\mu H)^\dagger L + (H\bar{L})\gamma_\mu(H^\dagger D_\mu L) - h.c.\Big] \\
&- i\frac{Y_\chi^2}{2M^4}\Big[(H\bar{L})\gamma_\mu\gamma_\nu\gamma_\rho(D_\mu D_\nu D_\rho H)^\dagger L + (H\bar{L})\gamma_\mu\gamma_\nu\gamma_\rho(D_\mu D_\nu H)^\dagger(D_\rho L) \\
&\quad + (H\bar{L})\gamma_\mu\gamma_\nu\gamma_\rho(D_\mu H)^\dagger(D_\nu D_\rho L) + (H\bar{L})\gamma_\mu\gamma_\nu\gamma_\rho H^\dagger(D_\mu D_\nu D_\rho L) \\
&\quad + (H\bar{L})\gamma_\mu\gamma_\nu\gamma_\rho(D_\mu D_\rho H)^\dagger(D_\nu L) + (H\bar{L})\gamma_\mu\gamma_\nu\gamma_\rho(D_\nu H)^\dagger(D_\mu D_\rho L) \\
&\quad + (H\bar{L})\gamma_\mu\gamma_\nu\gamma_\rho(D_\rho H)^\dagger(D_\mu D_\nu L) + (H\bar{L})\gamma_\mu\gamma_\nu\gamma_\rho(D_\nu D_\rho H)^\dagger(D_\mu L) - h.c.\Big]. \quad (23)
\end{aligned}
$$

This derivative expansion appears to imply that we could expect growth of the process with center of mass energy. However, by inspection we can see that most of the terms in Eq. 23 will result in a Feynman rule containing $\displaystyle{\not}p_V$ which will vanish for on-shell (massless) leptons. This is not obvious for some of the operators, however a more careful manipulation of the operators using IBP relations, removing terms generating rules with too many bosons, and deriving the Feynman rules reveals all operators of dimension-eight and higher do not contribute to the process. The only terms which remain after such an analysis are the dimension-six terms which simply renormalize the SM-like $A\bar{\ell}\ell$ and $Z\bar{\ell}\ell$ vertices. This is consistent with the plot in Fig. 1, where the dominant correction comes from the lower invariant mass bins.

To phrase this from a geometric perspective, the only operators affecting three-point functions are those classified in the geoSMEFT. The only operator appearing above which appears in the geoSMEFT is

$$
i(H\bar{L})\gamma_\mu(D^\mu H)^\dagger L - h.c. = \frac{1}{8}(H^\dagger i\overleftrightarrow{D}_\mu H)(\bar{L}\gamma^\mu L) + \frac{1}{2}(H^\dagger i\overleftrightarrow{D}^I_\mu H)(\bar{L}\gamma^\mu\tau^I L), \quad (24)
$$

where the $\tau^I$ are the generators of $SU(2)_L$, and

$$
H^\dagger i\overleftrightarrow{D}_\mu H = H^\dagger iD_\mu H - (iD_\mu H)^\dagger H, \quad (25)
$$

$$
H^\dagger i\overleftrightarrow{D}^I_\mu H = H^\dagger i\tau^I D_\mu H - (iD_\mu\tau^I H)^\dagger H. \quad (26)
$$

Therefore all other operators can be exchanged by equation of motion and IBP identities for four and higher point functions.

Unfortunately this means this effective Lagrangian is not particularly interesting for our purposes. That it has a dimension-six contribution to the Drell Yan process is a slight contrast to the first example, $\phi$, where for our input parameter choice there is no effect on the dynamics. In this case we again conclude a pure dimension-six analysis is sufficient. Although, looking at electroweak precision data (EWPD) [24, 25, 59–61] demonstrates that this model is severely constrained by $Z$-pole physics and we should not expect to obtain more stringent bounds from the Drell Yan process. We do not include a figure for this example as the dimension-six contribution fully reconciles the UV and IR predictions.

**Vector $X$**

In the infrared the Lagrangian for $X$ is by far the most complicated. As mentioned above, if we had simply integrated out the $V$ particle we would have obtained one operator at each order, $(\partial_\mu B_{\mu\nu})\partial^{2n}(\partial_\rho B^{\rho\mu})$. In order to avoid complications in this model from higher poles in the propagator resulting from the extra derivatives we made a field redefinition in the UV to arrive at a Lagrangian depending on $X$ field. This, and some IBP identities, allows us to arrive at an effective Lagrangian free of this complication:

$$
\begin{aligned}
\mathcal{L}_{\text{IR}}^X = \mathcal{L}_{\text{SM}} &- \frac{g_1^2\beta^2}{2M^2}\mathcal{H}_\mu\mathcal{H}^\mu - \frac{g_1^2\beta^2}{2M^2}\Psi_\mu\Psi^\mu - \frac{g_1^2\beta^2}{M^2}\mathcal{H}_\mu\Psi^\mu + \frac{g_1^4 Y_H^2\beta^4}{M^4}(H^\dagger H)\mathcal{H}_\mu\mathcal{H}^\mu \\
&+ \frac{g_1^2\beta^2}{M^4}\mathcal{H}_\mu(\Box\eta^{\mu\nu}-\partial^\mu\partial^\nu)\Psi_\nu + 2\frac{g_1^4 Y_H^2\beta^4}{M^4}(H^\dagger H)\mathcal{H}_\mu\Psi^\mu \\
&+ \frac{g_1^2\beta^2}{2M^4}\Psi_\mu(\Box\eta^{\mu\nu}-\partial^\mu\partial^\nu)\Psi_\nu + \frac{g_1^4 Y_H^2\beta^4}{M^4}(H^\dagger H)\Psi_\mu\Psi^\mu \\
&+ \frac{g_1^4 Y_H^4\beta^4}{M^4}\left[4(H^\dagger H)Q_{HD} + (H^\dagger H)Q_{HD,2}\right] \\
&+ \frac{g_1^4 Y_H^4\beta^4}{2M^4}\left[(H^\dagger H)^2(H^\dagger D^2 H)+h.c.\right] \\
&- \frac{g_1^2 Y_H^2\beta^2}{M^4}\left[\frac{g_1^2}{4}Q_{HB}^{(8)} + g_1 g_2 Q_{HWB}^{(8)} + g_2^2 Q_{HW,2}^{(8)}\right].
\end{aligned} \tag{27}
$$

We have made use of the following definitions to simplify the presentation:

$$
\mathcal{H}_\mu = Y_H(H^\dagger i\overleftrightarrow{D}_\mu H) = iY_H(H^\dagger D_\mu H) - iY_H(D_\mu H)^\dagger H \,, \tag{28}
$$

$$
\Psi_\mu = \sum_\psi Y_\psi \bar{\psi}\gamma_\mu\psi \,, \tag{29}
$$

$$
Q_{HB}^{(8)} = (H^\dagger H)^2 B_{\mu\nu}B^{\mu\nu} \,, \tag{30}
$$

$$
Q_{HWB}^{(8)} = (H^\dagger H)(H^\dagger\sigma^I H)W_{\mu\nu}^I B^{\mu\nu} \,, \tag{31}
$$

$$
Q_{HW,2}^{(8)} = (H^\dagger\sigma^I H)(H^\dagger\sigma^J H)W^{I,\mu\nu}W_{\mu\nu}^J \,. \tag{32}
$$

Where $Y_F$ is the hypercharge of a given field $F$, and the sum over $\psi$ is a sum over the SM (chiral) fermionic fields.

This Lagrangian neatly exhibits the momentum expansion which is illustrated by terms with the transverse projection operator $\Box\eta_{\mu\nu}-\partial_\mu\partial_\nu$. We can also see the vev expansion where dimension-six terms are accompanied by corresponding dimension eight operators rescaled by $(H^\dagger H)$. The operators, $Q_{HB}^{(8)}$, $Q_{HWB}^{(8)}$, and $Q_{HW,2}^{(8)}$ encode the corrections to the mixing between

the $X$ and $B$ particles in the UV. We note these corrections start at dimension-eight and so they would be missed by a leading order SMEFT study, which would therefore miss potentially stringent EWPD constraints derived from the $Q^{(8)}_{HWB}$ operator's contribution. The operator $\mathcal{H}_\mu \mathcal{H}^\mu$ is related to the operators $Q_{HD}$ and $Q_{HD,2}$ occurring in the $\phi$ model, and generates a shift in the $Z$-boson mass. Ultimately, as was found while determining the input parameters in the UV model, these operators have a negligible effect, when compared with the effects of the momentum expansion. That is, the cross section is dominated by the four-fermion operators. As such, only the four-fermion operators are included in our calculations of the cross sections. This is simply because these contributions dramatically slow the calculations due to the complexity of the expression for the partonic cross section, not because they are too cumbersome to implement.

Writing only the four-fermion operators, we can express the Lagrangian up to dimension ten:

$$\mathcal{L}^X_{IR} = -\frac{g_1^2 \beta^2}{2M^2} \Psi_\mu \Psi^\mu + \frac{g_1^2 \beta^2}{2M^4} \Psi_\mu \Pi^{\mu\nu} \Psi_\nu - \frac{g_1^2 \beta^2}{2M^6} \Psi_\mu \Pi^{\mu\nu} \Pi_{\nu\rho} \Psi^\rho \,, \tag{33}$$

where we have introduce $\Pi_{\mu\nu} = \Box \eta_{\mu\nu} - \partial_\mu \partial_\nu$. The operators present in Eq. 27 and neglected in Eq. 33 are generated by the following terms in the full UV model:

$$-g_1 Y_H \beta (H^\dagger i \overleftrightarrow{D}_\mu H) X^\mu + g_1^2 Y_H^2 \beta^2 (H^\dagger H)(X^\mu X_\mu) \,. \tag{34}$$

As mentioned, the contributions from the vev expansion are negligible in our analysis. For example, for a 3 TeV $X_\mu$ and $\beta = 3$ these operators contribute with an approximately $\mathcal{O}(10^{-2})$ effect in all invariant mass bins. This is contrasted with the momentum expansion where the effect is $\mathcal{O}(10^{-1.6})$ in the lowest invariant mass bins to $\mathcal{O}(100)$ in the highest invariant mass bins. As this example has the lowest mass and strongest mixing we expect all other potential parameter combinations to lead to similar or even more disparate contributions (the Wilson coefficients scale similarly, while the momentum dependence of the operators remains the same). To simplify our analysis and isolate the momentum expansion relevant to this study we will therefore only employ the IR Lagrangian of Eq. 33. We have tested that this has a negligible effect on the studies below.

Comparing Eq. 19 and Eq. 33 we see that the expansion in $p^2/M_X^2$ in the case of $X$ additively corrects the IR prediction in contrast with $\Phi$ where we found that at each order the sign of the contribution was flipped. This has dramatic effects for the dimension-six-squared contribution as, depending on the combination of parameters, it may improve the convergence or make it much worse. Figure 4 shows three benchmark examples for $\{M, \beta\} = \{3, 0.6\}$, $\{3, 1.2\}$, and $\{10, 3.0\}$. The chosen benchmarks are chosen to demonstrate the convergence of the series, but also that the dimension-six squared contributions sometimes fail and sometimes do much better which is simply accidental and due to the choice of the free parameters.

# 4 Comparing the IR with UV

With the models in the infrared derived to order $1/\Lambda^6$ we can compare the full UV model prediction and that of the EFT. In this section, our primary goal is to explore how well the EFT describes the Drell Yan processes order by order in its expansion. A complete SMEFT analysis is not possible as we consider only Drell Yan, but in the later part of this section we do estimate the impact of including additional operators not generated in the UV.

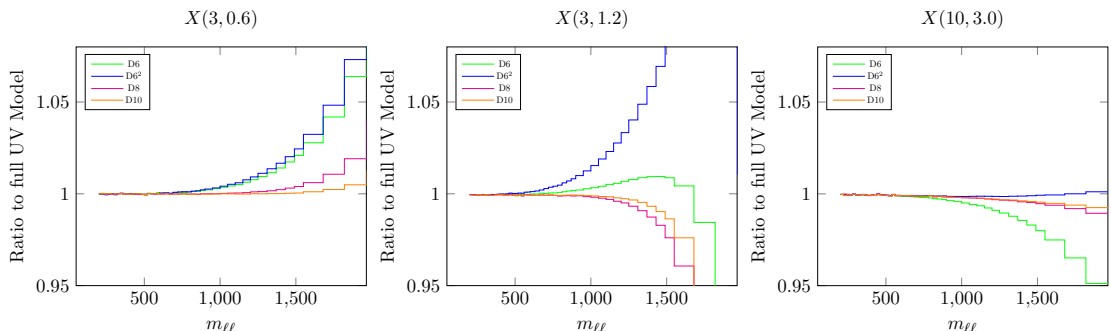

Figure 4: Plots comparing the accuracy of the EFT expansion as a ratio of the EFT to full UV theory cross section predictions. Benchmarks of $M = 3$ with $\beta = 0.6$ and $\beta = 1.2$, and $M = 10$ TeV with $\beta = 3.0$ are shown. The left-most plot nicely demonstrates the convergence of the expansion for a light mass and small mixing parameter. The middle plot shows that for a larger mixing parameter the convergence deteriorates. This plot shows a dramatic failure of the dimension-six squared contribution, while showing that due to accidental cancellations between the parameters of the EFT the pure dimension-six term does remarkably well to high invariant masses. The rightmost plot again shows the convergence for large mass and large mixing parameter, but also shows an example of when the dimension-six squared term accidentally makes a substantially better prediction than higher orders in the expansion.

## 4.1 Validity of the truncation

In figures 3 and 4 we implied that the truncation breaks down at a given order in $1/M^2$ by showing that the cross sections differ at the order of a few percent. While this is interesting theoretically, a more phenomenological approach requires us to consider if this breakdown is experimentally measurable. Our first measure of this breakdown is to multiply the differential cross sections by a given luminosity and comparing the number of events in the full UV model to the number of predicted events for a given order in the EFT expansion. We perform this analysis by:

1. Considering the integrated luminosity 140/fb for the 13 TeV LHC. This value is taken from the CMS paper cited above [53] from which we have taken the invariant mass binning. We then consider the HL-LHC scenario of 3/ab, maintaining 13 TeV center of mass energy.

2. We assume the simulated "experimental search" measures the UV model (i.e. the UV model is the signal), and that the error in a given bin is Poisson.

3. We compare with the theoretical prediction in the IR, for which we assume there is no theory error. This is consistent with how most global fits in the SMEFT are performed.

4. We already know the IR model as derived in Sec. 3. Admittedly, this is a rather weak assumption given state of the art studies of the SMEFT are done from a bottom up perspective. We estimate the impact of broadening this assumption in the next section.

5. We assume that the Wilson coefficients derived in the previous section are rescaled by a constant $c_d$ with $d$ the dimension of the operator. A given $c_d$ is a rescaling of all operators generated at a given dimension. For example in the $\Phi$ model the dimension-six Wilson

coefficient is rescaled as:

$$\frac{Y_\Phi^2}{M^2} \to c_6 \frac{Y_\Phi^2}{M^2}\,, \tag{35}$$

so a good fit should predict $c_6 = 1$. When more than one operator is obtained at a given dimension, they are all rescaled by a common $c_d$.

6. Best fit values of the Wilson coefficient(s) are obtained by performing a $\chi^2$ fit of the IR model to the simulated data (UV). The $\chi^2$ is given by:

$$\chi^2(c_6, c_8, c_{10}) = \sum_{m_{\ell\ell}} \left( \frac{N_{m_{\ell\ell}}^{\mathrm{UV}} - N_{m_{\ell\ell}}^{\mathrm{IR}}}{\sqrt{N_{m_{\ell\ell}}^{\mathrm{UV}}}} \right)^2\,, \tag{36}$$

with $N$ the number of events in a given invariant mass bin in the IR or UV. Minimizing the $\chi^2$ with respect to the $c_d$ yields the best fit values. We determine their $1\sigma$ errors by holding all but one $c_d$ constant at their best fit values and solving for the values of the remaining $c_d$ where $\chi^2 = \chi^2_{\min} + 1$. The fit is performed for full and partial contributions at each order in $1/M^2$ up to $1/M^6$. The highest invariant mass bin included is $m_{\ell\ell} \in \{1820, 1970\}$ GeV. We note that such a cutoff is not always (or even usually) employed in SMEFT studies, but we choose this to ensure the energy does not exceed the cutoff of the IR theory.

7. If the UV model result is more than one standard deviation away from the IR prediction we consider this a failure of the EFT truncation. That is, we need more terms in the EFT to correctly predict the UV physics.

We do not perform showering or detector simulations, nor do we consider e.g. acceptance cuts, this will serve to reduce the number of events and therefore hurt the statistics. However, in [53] they find for example the acceptance×efficiency is worst in low invariant mass bins, and best and approximately constant in high invariant mass bins (approximately 60%) for $Z'$s. Ultimately this does not affect our discussion of the convergence of the EFT expansion, but inclusion of these effects would serve to slightly alter our limited discussion of significance of measurements.

For the entirety of this article we only consider the processes at tree level. In [27] the authors use MCFM to obtain the SM cross section [62]. In [63–66] the authors consider one-loop contributions to the Drell Yan process at leading order in the SMEFT. However, as we will later compare the number of events in the UV to the number of events in the effective field theory, the missing contributions of the higher order corrections to the SM interfering with the tree-level effects of the new physics are still unknown and beyond the scope of this article. In absence of this, simply including the higher order corrections to the SM will not affect our results as these corrections do not contribute to the new physics signal captured by the EFTs. Inclusion of the SMEFT results beyond leading order without calculating the loop corrections in the UV would not be consistent, and the one-loop corrections to processes in the SMEFT have not been performed beyond order $1/\Lambda^2$. A comparison beyond leading order is beyond the scope of this work, however it would provide useful insights into the effects of theory errors on SMEFT analyses which are neglected in this work. Progress matching to higher orders in $1/\Lambda^2$ at one loop has been published in [67–69].

We do not focus on whether a given benchmark is already ruled out by LHC data as our goal is to understand the convergence of the series. In the case of the $\Phi$ model direct constraints are fairly loose, of order 1-2 TeV, as they must be pair produced [70]. For $Z'$ models more stringent constraints exist, requiring $M_Z'$ be larger than 3 or 4 TeV [70]. The case of our $X$ which has SM-like couplings to the fermions the constraints are more stringent. For the more weakly interacting models considered, a next generation collider may be able to draw the same conclusions as for the models with stronger interactions at the LHC.

### 4.1.1 $\Phi$ model

First we consider the theory of the extra $\Phi$ field for $M \in \{3, 10\}$ TeV in bins of 1 TeV and with $Y_\Phi \in \{0.1, 1.0\}$ in steps of 0.1. Due to the simplicity of our study the difference between 140/fb and 3/ab only results in different inferred $1\sigma$ error ranges. Table 1 illustrates the results of our study for 3 and 7 TeV with $Y_\Phi \in \{0.1, 0.5, 1.0\}$, App. C contains the full set of benchmarks in multiple tables.

In Tab. 1 we denote the (partial) order in the EFT expansion in the third column. The entries "D6," "D8," and "D10" refer to the full result to order $1/M_\Phi^2$, $1/M_\Phi^4$, $1/M_\Phi^6$ respectively. While "D6$^2$," and "D6D8" refer to the full result to order $1/M_\Phi^2$ or $1/M_\Phi^4$ supplemented by the partial results at order $1/M_\Phi^4$, $1/M_\Phi^6$. In this case we find that the dimension-six squared contribution *always* outperforms the consistent dimension-six contribution. As discussed below Eq. 22, this is because each subsequent order contributes with the opposite sign, so dimension-six squared neatly approximates the full $1/\Lambda^4$ contribution. The table also nicely demonstrates the convergence of the EFT expansion. Unsurprisingly for smaller $M_\Phi$ and larger $Y_\Phi$ more terms of the series are generally needed.

In the case of 140/fb the one-sigma error at dimension-six ($\delta c_6$) is sufficiently large in nearly all cases that the results are consistent with $c_6 = 0$ or 1 and no significant result is obtained. The exception is $M_\Phi = 3$ TeV with $Y_\Phi = 1$, in this case the dimension-six fit is consistent with $c_6 = 0$ and approximately $3\sigma$ away from the true value 1. It is important here to acknowledge that for this benchmark we would already have large deviations in the measured Drell Yan differential cross section.

For 3/ab the error shrinks by a factor of approximately 5. In this case the benchmark $M_\Phi = 3$ TeV with $Y_\Phi = 0.5$ could result in significant deviations. Considering this benchmark we can follow the expansion order by order to see the effects of fitting to higher orders:

- At dimension-six the fit gives a value $c_6 = 0.74 \pm 0.22$ This is just over $1\sigma$ away from the correct value. Fitting to only dimension-six is insufficient and can result in incorrect inferences about the nature of the new physics. If this occurred for a global fit in the bottom up approach one could envision that all measured Wilson coefficients are skewed in the same manner and perhaps this results in just misjudging the relationship between the mass and couplings of the new physics. However, in a more realistic example with multiple dimension-six operators with arbitrary coefficients, if the results were skewed in random directions and magnitudes away from the true values this could have more dire consequences. In this case we could fail to infer the UV model due to the pattern of matching from UV models being broken by our failure to expand to a sufficient order in the expansion. Largely related to this concern is that we would fail to identify symmetries of the UV which are naturally imprinted on the IR in the EFT approach.

- Partial results at dimension-six squared greatly improve the quality of the fit. The best fit point $0.96 \pm 0.30$ is fully consistent with the predicted value. While this is true for all benchmarks in the $\Phi$ model we will see below that it is not the case for the $X$ model and therefore we cannot assume that D6$^2$ terms will always help with the convergence.

- The full result at order $1/M_\Phi^4$ yields $c_6 = 0.96 \pm 0.30$ and is consistent with the predicted value. The Wilson coefficient of the dimension-eight operators is $-0.28 \pm 2.3$ and is consistent with both 0 and 1. We conclude that the dimension-eight operators' Wilson coefficients are absorbing the failure of the series to fully replicate the higher invariant mass bins. That is, they are playing a role of nuisance parameters instead of being measured as terms in the EFT expansion.[3] This is an inference from the behavior of the fits considered in this article,

---

[3]This claim should be tested in the context of multiple observables. It is possible that including multiple observables will change this picture. This is, however, beyond the scope of this project.

it is not a proof and should not be understood to refer to any formal definition of a nuisance parameter.

- The full dimension-eight contribution supplemented by the interference of the dimension-six and eight amplitudes continues to improve the convergence of the series. We also note that in this case the dimension-eight Wilson coefficient is also moving toward the true value.

- The full dimension-ten result yields excellent agreement at dimension-six while the Wilson coefficient of the dimension-eight result continues to move toward its true value. Adding the dimension-ten operators results in the fit to their Wilson coefficients being far from the true value ($c_{10} = -0.43 \pm 10$), but again gives the impression that these free parameters in the fit are absorbing our ignorance of the higher order terms in the series.

For a 7 TeV $\Phi$ the picture is largely the same, except that the relatively weaker influence of the $\Phi$ results in larger error bands. In the case of a top-like Yukawa coupling, $Y_\Phi = 1$, the measurement of $c_6$ has a significance of just over $2\sigma$. Again, a fit only at dimension six results in a best fit value of $c_6$ which is 20% below the true value.

In our simplified approach where we do not consider how the binning or possible cuts may change with increased luminosity, the only effect of increasing the luminosity is that the error in our measured Wilson coefficients shrink by $\sqrt{\mathcal{L}_{\text{old}}/\mathcal{L}_{\text{new}}}$. For a result more than one-sigma away from the SM ($c_i = 0$) in the $M_\Phi = 3$, $Y_\Phi = 0.1$ benchmark, we would need approximately 100/ab at the 13 TeV LHC. To notice the difference between truncating at dimension six versus order $1/\Lambda^4$, we would need nearly 20/zb. Such integrated luminosities are not possible even at future hadronic colliders.

### 4.1.2 X model

The first thing to notice about the $X$ model is that it is far more complicated than the $\Phi$ model. Considering Fig. 4 we can see that depending on the parameters of the model the dimension-six squared terms may hurt or improve the convergence of the series. The $\Phi$ particle only couples to left handed leptons and right handed down quarks, while the $X$ particle couples to all SM currents with coupling proportional to $g_1 Y_\psi \beta$. For certain choices of $\beta$ and $M_X$ accidental cancellations can cause the convergence of the series to behave poorly.

Another important aspect of the $X$ model is that in the UV it corresponds to an $s$-channel exchange. As such the number of events in the tails of the $m_{\ell\ell}$ distribution is higher. This results in better statistics overall in this model. This also means that some of the benchmarks discussed here may be ruled out by current data. We focus on the phenomenological implications and the behavior of the series and not on viability of the models given current LHC measurements.

We checked the benchmark values of $M_X \in \{3, 8\}$ TeV in bins of 1 TeV with $\beta \in \{0.3, 3\}$. Recalling the definition of beta in terms of the mixing parameter $k$,

$$\beta \equiv \frac{-k}{\sqrt{1-k^2}}, \tag{37}$$

the coupling of the $X$ particle to a given chiral fermion $\psi$ is $g_1 Y_\psi \beta$. For $\beta = 3$ this is roughly $2Y_\psi$, meaning the coupling of the $X$ is less than one for all fermions except the right handed leptons for which it is 2. A selection of benchmarks from the results are included in Tab. 2, while the full set can be found in App. C.

We can see similarities between the overall convergence between this example and that of the $\Phi$. However, we notice that in the case of the low mass with small coupling ($\beta = 0.3 \rightarrow k \sim -0.3$) the fit to dimension-six results in a best fit point off by 10%, though we note this is within the $1\sigma$ errors. This is resolved already at dimension-eight. Interestingly,

Table 1: Abridged table of benchmark $\Phi$ models. The full tables can be found in Appendix C. $\delta c_i$ is the one-sigma error in the measured value of $c_i$. The error in parenthesis is for 3/ab, while the other is for 140/fb. The value of $\chi^2_{\min}$ indicates the minimum value of $\chi^2$ obtained from the fit indicating that the goodness of fit improves order by order in the series. All numbers are rounded to two significant digits.

| $M_\Phi$ | $Y_\Phi$ | dim | $c_6$ | $\delta c_6$ | $c_8$ | $\delta c_8$ | $c_{10}$ | $\delta c_{10}$ | $\chi^2_{\min}$ |
|---|---|---|---|---|---|---|---|---|---|
| 3 | 0.1 | D6 | 0.93 | 26 (5.6) | - | - | - | - | $10^{-4}$ |
| - | - | D6$^2$ | 0.94 | 28 (5.7) | - | - | - | - | $10^{-4}$ |
| - | - | D8 | 0.99 | 28 (5.8) | 0.74 | 270 (57) | - | - | $10^{-6}$ |
| - | - | D6D8 | 1.0 | 28 (5.7) | 0.81 | 280 (61) | - | - | $10^{-6}$ |
| - | - | D10 | 1.0 | 28 (5.7) | 0.97 | 280 (61) | 0.58 | 1.2k (250) | $10^{-8}$ |
| 3 | 0.5 | D6 | 0.74 | 1.0 (0.22) | - | - | - | - | $10^0$ |
| - | - | D6$^2$ | 0.96 | 1.2 (0.30) | - | - | - | - | $10^{-1}$ |
| - | - | D8 | 0.96 | 1.2 (0.30) | -0.28 | 11 (2.3) | - | - | $10^{-2}$ |
| - | - | D6D8 | 0.99 | 1.2 (0.31) | 0.60 | 13 (2.8) | - | - | $10^{-2}$ |
| - | - | D10 | 1.0 | 1.2 (0.31) | 0.72 | 13 (2.8) | -0.43 | 47 (10) | $10^{-4}$ |
| 3 | 1.0 | D6 | 0.16 | 0.26 (0.057) | - | - | - | - | $10^2$ |
| - | - | D6$^2$ | 0.84 | 0.14 (0.030) | - | - | - | - | $10^1$ |
| - | - | D8 | 0.87 | 0.13 (0.029) | -0.62 | 2.8 (0.62) | - | - | $10^1$ |
| - | - | D6D8 | 0.97 | 0.14 (0.029) | 0.61 | 0.52(0.11) | - | - | $10^1$ |
| - | - | D10 | 0.98 | 0.13 (0.027) | 0.38 | 0.52 (0.11) | 6.6 | 15 (2.8) | $10^{-1}$ |
| 7 | 0.1 | D6 | 0.99 | 141 (31) | - | - | - | - | $10^{-7}$ |
| - | - | D6$^2$ | 0.99 | 152 (31) | - | - | - | - | $10^{-7}$ |
| - | - | D8 | 1.0 | 152 (31) | 0.96 | 7.9k (1.7k) | - | - | $10^{-7}$ |
| - | - | D6D8 | 1.0 | 152 (31) | 0.98 | 8.0k (1.7k) | - | - | $10^{-7}$ |
| - | - | D10 | 1.0 | 152 (31) | 1.1 | 8.0k (1.7k) | 2.6 | 190k (40k) | $10^{-7}$ |
| 7 | 0.5 | D6 | 0.94 | 5.6 (1.2) | - | - | - | - | $10^{-3}$ |
| - | - | D6$^2$ | 0.99 | 6.3 (1.3) | - | - | - | - | $10^{-4}$ |
| - | - | D8 | 1.0 | 6.3 (1.3) | 0.64 | 320 (68) | - | - | $10^{-6}$ |
| - | - | D6D8 | 1.0 | 6.3 (1.3) | 0.97 | 440 (94) | - | - | $10^{-6}$ |
| - | - | D10 | 1.0 | 6.3 (1.3) | 0.99 | 440 (94) | 0.34 | 7.5k (1.6k) | $10^{-8}$ |
| 7 | 1.0 | D6 | 0.80 | 1.4 (0.30) | - | - | - | - | $10^0$ |
| - | - | D6$^2$ | 0.99 | 1.6 (0.41) | - | - | - | - | $10^{-3}$ |
| - | - | D8 | 0.99 | 1.6 (0.41) | -0.16 | 79 (17) | - | - | $10^{-3}$ |
| - | - | D6D8 | 1.0 | 1.6 (0.41) | 0.90 | 160 (36) | - | - | $10^{-3}$ |
| - | - | D10 | 1.0 | 1.6 (0.41) | 0.94 | 160 (36) | -0.66 | 1.9k (400) | $10^{-7}$ |

for the more strongly mixed case of $\beta = 1.2$ with $M_X = 3$ TeV, the best fit value to dimension-six only performs better than the dimension six-squared fit. This is in contrast, for example with $M_X = 5$ TeV with $\beta = 3.0$ where the dimension-six squared fit performs better than the dimension-six only and even the full dimension-ten fit. We also note that the dimension-eight operator coefficient again plays a role similar to nuisance parameters, however since there are

more events for the $s$-channel process $c_8$ is usually closer to one since there are better statistics even in the weaker coupling limit.

Interestingly, a 5 TeV $X$ which strongly mixes ($\beta = 3.0$) with the SM $B$ field results in a sufficient number of events to precisely measure the dimension-six operator coefficient and for 3/ab integrated luminosity the dimension-eight operator coefficient can be measured with some degree of precision as well. In this case we see, in direct analogue with dimension-six vs dimension-eight fits, that the inclusion of the dimension-ten operators improves the agreement between the best fit point for $c_8$ and the theory prediction. This is interesting academically, however this model is likely ruled out already by the relatively fewer events observed in Run III of the LHC. An $X$ which strongly mixes into the SM with mass 8 TeV may still evade constraints from current measurements, however in this case the error on the dimension-eight Wilson coefficients is too large for a measurement with any meaningful significance to be achieved.

## 4.2 Fitting the SMEFT

As mentioned in the last section, our analysis so far is strictly top down. In order to attempt to understand how a bottom up EFT analysis such as the state of the art SMEFT global fits would perform, we now perform a similar analysis but include additional operators from the SMEFT. As we will see, this proves difficult for the limited amount of data coming from considering a single process so we will focus primarily on including only one additional operator at dimension six.

The following four-fermion operators contribute to the Drell-Yan process at tree level and order $1/\Lambda^2$ in the SMEFT:

$$
\begin{aligned}
\mathcal{L}_{\text{SMEFT}}^{4-\text{ferm}} = {} & c_{LQ}^{(1)}(\bar{L}\gamma_\mu L)(\bar{Q}\gamma^\mu Q) + c_{LQ}^{(3)}(\bar{L}\gamma_\mu \tau^I L)(\bar{Q}\gamma^\mu \tau^I Q) \\
& + c_{eu}(\bar{e}\gamma_\mu e)(\bar{u}\gamma^\mu u) + c_{ed}(\bar{e}\gamma_\mu e)(\bar{d}\gamma^\mu d) \\
& + c_{Lu}(\bar{L}\gamma_\mu L)(\bar{u}\gamma^\mu u) + c_{Ld}(\bar{L}\gamma_\mu L)(\bar{d}\gamma^\mu d) + c_{Qe}(\bar{Q}\gamma_\mu Q)(\bar{e}\gamma^\mu e) \\
& + c_{LedQ}(\bar{L}e)(\bar{d}Q) + c_{LeQu}(\bar{L}e)i\sigma_2(\bar{Q}u).
\end{aligned}
\tag{38}
$$

The operators appearing in the first line have chiral currents of the form $LL$, followed by $RR$, and $LR$, the final lines are scalar currents which will mix chiralities. The flipped chirality requires mass insertions and so we neglect the terms in the final line.[4]

Neglecting operators such as those in Eq. 24 we find for the contribution to the spin- and color-averaged partonic cross section in the $m_Z \to 0$ limit from interference between the SM amplitude and the $1/\Lambda^2$ amplitude:

$$
\begin{aligned}
\sigma(\hat{s})|_{\text{D6}} = \frac{2}{48\pi N_c}\Bigg[ & \left(g_L^q g_L^e + e^2 Q_q Q_e\right)\left(c_{LQ}^{(1)} \mp \frac{1}{4}c_{LQ}^{(3)}\right) + \left(g_L^q g_R^e + e^2 Q_q Q_e\right)c_{Qe} \\
& + \left(g_R^q g_L^e + e^2 Q_q Q_e\right)c_{Lq} + \left(g_R^q g_R^e + e^2 Q_q Q_e\right)c_{eq} \Bigg],
\end{aligned}
\tag{39}
$$

where the minus (plus) sign is for up (down) quarks, and lower-case $q$ should be taken to correspond to the right-handed $u$ or $d$ chiral quarks. This contribution, occuring at order $1/\Lambda^2$, is a constant with respect to the square of the partonic center of mass energy, $\hat{s}$. The

---

[4]Notice that Fierz identities such as the one appearing in Eq. 20 do not rectify this situation. The on-shell fermions will have opposite chiralities and therefore the interaction will select out mass terms from spin sums over on-shell spinors. This does not happen in the case of the IR theory of Eq. 19 as the fermion bilinears occurring in the operators are products of quark-lepton instead of quark-quark and lepton-lepton.

Table 2: Abridged table of benchmark $X$ models. The full tables can be found in Appendix C. $\delta c_i$ is the one-sigma error in the measured value of $c_i$. The error in parenthesis is for 3/ab, while the other is for 140/fb. The value of $\chi^2_{\min}$ indices the minimum value of $\chi^2$ obtained from the fit indicating that the goodness of fit improves order by order in the series. All numbers are rounded to two digits.

| $M_X$ | $\beta$ | dim | $c_6$ | $\delta c_6$ | $c_8$ | $\delta c_8$ | $c_{10}$ | $\delta c_{10}$ | $\chi^2_{\min}$ |
|---|---|---|---|---|---|---|---|---|---|
| 3 | 0.3 | D6 | 1.1 | 1.6 (0.35) | - | - | - | - | $10^{-1}$ |
| - | - | D6$^2$ | 1.1 | 1.7 (0.36) | - | - | - | - | $10^{-1}$ |
| - | - | D8 | 0.98 | 1.7 (0.36) | 1.5 | 11 (2.4) | - | - | $10^{-2}$ |
| - | - | D6D8 | 0.98 | 1.7 (0.36) | 1.5 | 11 (2.4) | - | - | $10^{-2}$ |
| - | - | D10 | 1.0 | 1.7 (0.36) | 0.85 | 11 (2.4) | 1.9 | 38 (8.1) | $10^{-4}$ |
| 3 | 1.2 | D6 | 1.0 | 0.096 (0.021) | - | - | - | - | $10^{0}$ |
| - | - | D6$^2$ | 1.1 | 0.11 (0.024) | - | - | - | - | $10^{1}$ |
| - | - | D8 | 1.0 | 0.11 (0.024) | 0.62 | 0.58 (0.13) | - | - | $10^{0}$ |
| - | - | D6D8 | 1.0 | 0.11 (0.025) | 1.1 | 0.94 (0.20) | - | - | $10^{0}$ |
| - | - | D10 | 0.99 | 0.11 (0.025) | 1.3 | 0.94 (0.20) | -0.22 | 1.9 (0.41) | $10^{-1}$ |
| 5 | 0.3 | D6 | 1.0 | 4.6 (0.98) | - | - | - | - | $10^{-3}$ |
| - | - | D6$^2$ | 1.0 | 4.6 (0.99) | - | - | - | - | $10^{-3}$ |
| - | - | D8 | 1.0 | 4.6 (0.99) | 1.1 | 86 (19) | - | - | $10^{-5}$ |
| - | - | D6D8 | 1.0 | 4.6 (0.99) | 1.1 | 87 (19) | - | - | $10^{-5}$ |
| - | - | D10 | 1.0 | 4.6 (0.99) | 0.83 | 87 (19) | 2.2 | 820 (180) | $10^{-5}$ |
| 5 | 1.2 | D6 | 1.0 | 0.28 (0.060) | - | - | - | - | $10^{-2}$ |
| - | - | D6$^2$ | 1.0 | 0.29 (0.064) | - | - | - | - | $10^{0}$ |
| - | - | D8 | 1.0 | 0.29 (0.064) | 0.93 | 5.1 (1.1) | - | - | $10^{-4}$ |
| - | - | D6D8 | 1.0 | 0.30 (0.064) | 1.1 | 5.9 (1.3) | - | - | $10^{-4}$ |
| - | - | D10 | 1.0 | 0.30 (0.064) | 1.0 | 5.9 (1.3) | 0.77 | 47 (10) | $10^{-5}$ |
| 5 | 3.0 | D6 | 0.81 | 0.041 (0.0089) | - | - | - | - | $10^{3}$ |
| - | - | D6$^2$ | 0.99 | 0.057 (0.012) | - | - | - | - | $10^{1}$ |
| - | - | D8 | 1.0 | 0.057 (0.012) | -0.13 | 0.70 (0.15) | - | - | $10^{0}$ |
| - | - | D6D8 | 0.98 | 0.058 (0.012) | 0.84 | 2.1 (0.46) | - | - | $10^{0}$ |
| - | - | D10 | 0.98 | 0.057 (0.012) | 0.98 | 2.1 (0.46) | -2.0 | 6.6 (1.4) | $10^{-2}$ |
| 8 | 1.5 | D6 | 1.0 | 0.46 (0.010) | - | - | - | - | $10^{-3}$ |
| - | - | D6$^2$ | 1.0 | 0.48 (0.10) | - | - | - | - | $10^{-2}$ |
| - | - | D8 | 1.0 | 0.48 (0.10) | 0.92 | 22 (4.7) | - | - | $10^{-5}$ |
| - | - | D6D8 | 1.0 | 0.48 (0.10) | 1.0 | 24 (5.2) | - | - | $10^{-5}$ |
| - | - | D10 | 1.0 | 0.48 (0.10) | 0.96 | 24 (5.2) | 1.3 | 530 (110) | $10^{-5}$ |
| 8 | 3.0 | D6 | 0.92 | 0.11 (0.024) | - | - | - | - | $10^{1}$ |
| - | - | D6$^2$ | 0.99 | 0.13 (0.027) | - | - | - | - | $10^{-1}$ |
| - | - | D8 | 0.98 | 0.13 (0.027) | 0.47 | 5.0 (1.1) | - | - | $10^{-2}$ |
| - | - | D6D8 | 0.98 | 0.13 (0.027) | 0.78 | 7.4 (1.6) | - | - | $10^{-2}$ |
| - | - | D10 | 0.98 | 0.13 (0.028) | 0.85 | 7.4 (1.6) | -1.0 | 120 (25) | $10^{-5}$ |

couplings of the $Z$-boson to the fermions are given by

$$g_L^\psi = \frac{g_Z}{2}(2Q_\psi s_W^2 - \sigma_3), \tag{40}$$

$$g_R^\psi = g_Z Q_\psi s_W^2, \tag{41}$$

and receive no corrections from the SMEFT under our simplifying assumption that only four-fermion operators contribute. We have also used $g_Z \equiv g_2/c_W$[5] and $\sigma_3$ corresponds to twice the weak isospin projection of a given left-handed fermion. For the dimension-six squared contribution we find that the partonic cross section grows proportional to $\hat{s}$:

$$\sigma(\hat{s})|_{\text{D}6^2} = \frac{\hat{s}}{48\pi N_c}\left[\left(c_{LQ}^{(1)} \mp \frac{1}{4}c_{LQ}^{(3)}\right)^2 + \left(c_{Qe}\right)^2 + \left(c_{Lq}\right)^2 + \left(c_{eq}\right)^2\right]. \tag{42}$$

This result neatly shows that in the massless fermion limit the different fermion chiralities do not interfere. Comparing Eqs. 39 and 42 we see that the inclusion of the squares of dimension-six contributions allows for some degree of distinguishing the Wilson coefficients when binning in the invariant mass of the leptons. We stress that such an approach is inconsistent with the bottom-up EFT approach as it is not the complete $1/\Lambda^4$ contribution, but as discussed elsewhere in this article it is common in the field and we include it for the sake of discussion and as we will include some dimension-eight operators in the following.

To make a comparison with the pseudo-data resulting from our UV scenarios, we integrate the full partonic cross sections folded with the pdfs. In doing so we do not take the limit $m_Z \to 0$ as in Eqs. 39 and 42. We use a parameterization of the full result as described in App. B.

A fit to all Wilson coefficients for the operators of Eq. 38 is technically possible with the (pseudo)data we have generated. For this we employ the same $\chi^2$ methodology outlined in Sec. 4.1. However, combinations of the Wilson coefficients can be used to approximate the SM to a great degree.

To start, we consider the SM Drell Yan process. That is, our signal is now simply the SM, and the constraints on the SMEFT should be consistent with all Wilson coefficients 0. We take the normalization of each Wilson coefficient to be,

$$c_i \to \frac{c_i}{(3\text{TeV})^2}, \tag{43}$$

and naively applying a Mathematica minimization routine, we obtain the following limits on the $c_i$ of the SMEFT to order $1/\Lambda^2$ for 3/ab integrated luminosity:

$$\begin{aligned}
c_{LQ}^{(1)} &= 0.20(1), & c_{LQ}^{(3)} &= -0.10(1), \\
c_{eu} &= -0.023(2), & c_{ed} &= 0.04(1), \\
c_{Lu} &= 0.57(3), & c_{Ld} &= 0.57(3), & c_{Qe} &= -0.38(1),
\end{aligned} \tag{44}$$

where we have retained the number of digits out to the $1\sigma$ error which is indicated by parenthesis. Considering this is for 3/ab, the results appear to indicate a significant deviation from the SM. The value of $\chi^2$ at the minimum is $\mathcal{O}(10^{-26})$. Since we have not added random noise into our data, the value of $\chi^2$ for all Wilson coefficients set to zero is identically zero. While the parameter space does indeed close, the limits on the Wilson coefficients are highly correlated and there exist narrow regions in the parameter space which are allowed within the $1\sigma$ bounds. Our method of obtaining the error in a given Wilson coefficient does not take this into

---

[5]In the general SMEFT this relationship is modified by operators such as $c_{HD}$.

accounting making the values above appear significant when they are not. This will prove a problem when we consider UV models below.

If we instead use a search specifying the SM point as the starting point the numerical search does not miss the true minimum and we obtain:

$$
\begin{aligned}
c_{LQ}^{(1)} &= 0 \pm 7 \cdot 10^{-3}, & c_{LQ}^{(3)} &= 0 \pm 8 \cdot 10^{-3}, \\
c_{eu} &= 0 \pm 2 \cdot 10^{-3}, & c_{ed} &= 0 \pm 1 \cdot 10^{-2}, \\
c_{Lu} &= 0 \pm 9 \cdot 10^{-3}, & c_{Ld} &= 0 \pm 3 \cdot 10^{-2}, & c_{Qe} &= 0 \pm 1 \cdot 10^{-2}.
\end{aligned}
\tag{45}
$$

Including the contributions of the squares of the dimension-six operators the absolute minimum is approximated well with best fit points of order $10^{-6}$ and we reproduce the errors of Eq. 45. This is a direct consequence of the quadratic terms in Eq. 42 which largely remove the narrow regions in parameters space where the Wilson coefficients' contributions add to approximately zero.

At dimension-six issues with these narrow regions may be assuaged through a global fit where these operators are further constrained by other data, such as $Z$-pole data. However, four-fermion operators are generally neglected on the $Z$-pole as their contribution is subdominant to others. We should note, in a more realistic fit to the SMEFT one would need to include bosonic operators which contribution through renormalization of the kinetic and mass terms of the SMEFT which would prevent the fit from being closed.

In what follows we consider a few interesting benchmarks from the $\Phi$ and $X$ models.

### 4.2.1 $\Phi$ model

Instead of using the SM Drell Yan results for our data, we next consider the $\Phi$ model. We further want to see the impact of including dimension-eight operator contributions on the fit. To achieve this we will consider more than one operator at dimension-six and the dimension-eight operators, with a single Wilson coefficient, derived in Sec. 3 for the $\Phi$ model. All dimension-six Wilson coefficients are normalized according to:

$$
c_i \rightarrow \frac{c_i Y_\Phi^2}{(3\,\text{TeV})^2},
\tag{46}
$$

with the exception of the dimension-eight Wilson coefficient which is normalized such that the theory value is 1. Table 3 shows the result for considering the combination of $c_{Ld}$, the single dimension-six operator generated by the UV, with any of the other Wilson coefficients of Eq. 39. We consider the cases of $M = 3$ TeV and $Y_\Phi = \{0.1, 0.5\}$ in order to exaggerate the effects of the fit. In the table we split results based on including or excluding dimension-six squared effects and including or excluding the dimension-eight operators. We only consider the dimension-eight operators generated by the UV. This is again a very over-simplified assumption, but our single-process analysis is too limiting to consider further dimension-eight operator contributions. When the dimension-eight operators are included we always include the dimension-six squared contribution for $c_{Ld}$.

In the case of $Y_\Phi = 0.1$ we find the errors are so large that the model is always consistent with the SM prediction of $c_i = 0$. Nonetheless, we see that the inclusion of the dimension eight operator combination appearing in Eq. 19, $c_8 \neq 0$, improves the central values of the model: the operator coefficient $c_{Ld}$ always moves toward 1, while the other operator coefficient moves toward 0. The most dramatic case is the fit to both $c_{Ld}$ and $c_{ed}$ where the best fit point for $c_{Ld}$ is close to zero, while $c_{ed}$ is closer to one-half. With the inclusion of dimension-eight operators the central values neatly move to be closer to their theory values $\{c_{Ld} = 1, c_{ed} = 0\}$. The inclusion of dimension-six squared contributions does nominally improve all fits, although in some cases the effect is smaller than the rounding employed.

Table 3: Example fits to two dimension-six operators for the $\Phi$ model. We consider $M = 3$ TeV and $Y_\Phi = \{0.1, 0.5\}$. We also consider the impact of the inclusion of dimension-six squared (Left vs Right) as well as the inclusion of the dimension-eight operators generated in integrating out the $\Phi$ (top vs bottom within a given row delineated by horizontal rules). When the dimension-eight operators are included, the dimension-six squared contribution from $c_{Ld}$ is always included. Results are rounded to one or two significant figures depending on how many decimal places are required to be nonzero. The use of $^*$ indicates that the differences from the model prediction of 1 or 0 for a given Wilson coefficient is greater than one sigma when not rounded.

| $M = 3$ TeV, $Y_\Phi = 0.1$ | | | | |
|---|---|---|---|---|
| | excluding d6$^2$ | | including d6$^2$ | |
| $c_8 = 0$ | $c_{Ld} = 1.2 \pm 5.6$ | $c_{LQ}^{(1)} = 0.08 \pm 1.3$ | $c_{Ld} = 1.2 \pm 5.8$ | $c_{LQ}^{(1)} = 0.06 \pm 1.3$ |
| $c_8 \neq 0$ | $c_{Ld} = 1.1 \pm 5.8$ | $c_{LQ}^{(1)} = 0.02 \pm 1.3$ | $c_{Ld} = 1.1 \pm 5.8$ | $c_{LQ}^{(1)} = 0.02 \pm 1.3$ |
| $c_8 = 0$ | $c_{Ld} = 2.1 \pm 5.6$ | $c_{LQ}^{(3)} = -0.3 \pm 1.7$ | $c_{Ld} = 1.7 \pm 5.8$ | $c_{LQ}^{(3)} = -0.2 \pm 1.7$ |
| $c_8 \neq 0$ | $c_{Ld} = 0.9 \pm 5.7$ | $c_{LQ}^{(3)} = 0.01 \pm 1.7$ | $c_{Ld} = 1.0 \pm 5.7$ | $c_{LQ}^{(3)} = 0.01 \pm 1.7$ |
| $c_8 = 0$ | $c_{Ld} = 1.6 \pm 5.6$ | $c_{eu} = -0.05 \pm 0.40$ | $c_{Ld} = 1.5 \pm 5.6$ | $c_{eu} = -0.03 \pm 0.4$ |
| $c_8 \neq 0$ | $c_{Ld} = 1.0 \pm 5.7$ | $c_{eu} = 0.003 \pm 0.40$ | $c_{Ld} = 1.0 \pm 5.7$ | $c_{eu} = 0.003 \pm 0.4$ |
| $c_8 = 0$ | $c_{Ld} = 0.09 \pm 5.6$ | $c_{ed} = 0.41 \pm 2.7$ | $c_{Ld} = 0.02 \pm 5.6$ | $c_{ed} = 0.44 \pm 2.7$ |
| $c_8 \neq 0$ | $c_{Ld} = 0.91 \pm 5.7$ | $c_{ed} = 0.04 \pm 2.7$ | $c_{Ld} = 0.91 \pm 5.7$ | $c_{ed} = 0.04 \pm 2.7$ |
| $c_8 = 0$ | $c_{Ld} = 1.5 \pm 5.6$ | $c_{Lu} = 0.19 \pm 1.8$ | $c_{Ld} = 1.5 \pm 5.8$ | $c_{Lu} = 0.13 \pm 1.8$ |
| $c_8 \neq 0$ | $c_{Ld} = 1.1 \pm 5.8$ | $c_{Lu} = 0.04 \pm 1.8$ | $c_{Ld} = 1.1 \pm 5.8$ | $c_{Lu} = 0.04 \pm 1.8$ |
| $c_8 = 0$ | $c_{Ld} = 2.2 \pm 5.6$ | $c_{Qe} = 0.5 \pm 2.2$ | $c_{Ld} = 1.6 \pm 5.8$ | $c_{Qe} = 0.4 \pm 2.1$ |
| $c_8 \neq 0$ | $c_{Ld} = 1.3 \pm 5.8$ | $c_{Qe} = 0.1 \pm 2.2$ | $c_{Ld} = 1.2 \pm 5.8$ | $c_{Qe} = 0.1 \pm 2.2$ |

| $M = 3$ TeV, $Y_\Phi = 0.5$ | | | | |
|---|---|---|---|---|
| | excluding d6$^2$ | | including d6$^2$ | |
| $c_8 = 0$ | $c_{Ld} = 1.9 \pm 0.2$ | $c_{LQ}^{(1)} = 0.29 \pm 0.05$ | $c_{Ld} = 0.9 \pm 0.3$ | $c_{LQ}^{(1)} = -0.01 \pm 0.05$ |
| $c_8 \neq 0$ | $c_{Ld} = 1.3 \pm 0.3^*$ | $c_{LQ}^{(1)} = 0.11 \pm 0.05$ | $c_{Ld} = 1.3 \pm 0.3^*$ | $c_{LQ}^{(1)} = 0.11 \pm 0.05$ |
| $c_8 = 0$ | $c_{Ld} = 4.9 \pm 0.2$ | $c_{LQ}^{(3)} = -1.2 \pm 0.1$ | $c_{Ld} = 0.9 \pm 0.3$ | $c_{LQ}^{(3)} = 0.01 \pm 0.07$ |
| $c_8 \neq 0$ | $c_{Ld} = 0.7 \pm 0.3^*$ | $c_{LQ}^{(3)} = 0.1 \pm 0.1^*$ | $c_{Ld} = 0.7 \pm 0.3^*$ | $c_{LQ}^{(3)} = 0.08 \pm 0.07$ |
| $c_8 = 0$ | $c_{Ld} = 3.3 \pm 0.2$ | $c_{eu} = -0.18 \pm 0.02$ | $c_{Ld} = 0.93 \pm 0.2$ | $c_{eu} = 0.00 \pm 0.02$ |
| $c_8 \neq 0$ | $c_{Ld} = 0.8 \pm 0.3$ | $c_{eu} = 0.01 \pm 0.02$ | $c_{Ld} = 0.8 \pm 0.3$ | $c_{eu} = 0.01 \pm 0.02$ |
| $c_8 = 0$ | $c_{Ld} = -2.5 \pm 0.2$ | $c_{ed} = 1.6 \pm 0.1$ | $c_{Ld} = 0.9 \pm 0.3$ | $c_{ed} = 0.02 \pm 0.11$ |
| $c_8 \neq 0$ | $c_{Ld} = 0.5 \pm 0.3$ | $c_{ed} = 0.2 \pm 0.1$ | $c_{Ld} = 0.5 \pm 0.3$ | $c_{ed} = 0.23 \pm 0.11$ |
| $c_8 = 0$ | $c_{Ld} = 2.9 \pm 0.2$ | $c_{Lu} = 0.7 \pm 0.1$ | $c_{Ld} = 0.9 \pm 0.3$ | $c_{Lu} = -0.01 \pm 0.07$ |
| $c_8 \neq 0$ | $c_{Ld} = 1.6 \pm 0.2$ | $c_{Lu} = 0.2 \pm 0.1$ | $c_{Ld} = 1.6 \pm 0.2$ | $c_{Lu} = 0.2 \pm 0.1$ |
| $c_8 = 0$ | $c_{Ld} = 5.5 \pm 0.2$ | $c_{Qe} = 1.9 \pm 0.1$ | $c_{Ld} = 0.9 \pm 0.3$ | $c_{Qe} = -0.01 \pm 0.09$ |
| $c_8 \neq 0$ | $c_{Ld} = 2.3 \pm 0.2$ | $c_{Qe} = 0.6 \pm 0.1$ | $c_{Ld} = 1.5 \pm 0.3$ | $c_{Qe} = 0.2 \pm 0.1$ |

It is more interesting to consider the case of $Y_\Phi = 0.5$ as there are far more signal events, and the errors in the best fit values for the Wilson coefficients are substantially smaller. In this case we see that the inclusion of dimension-six squared contributions has a much larger

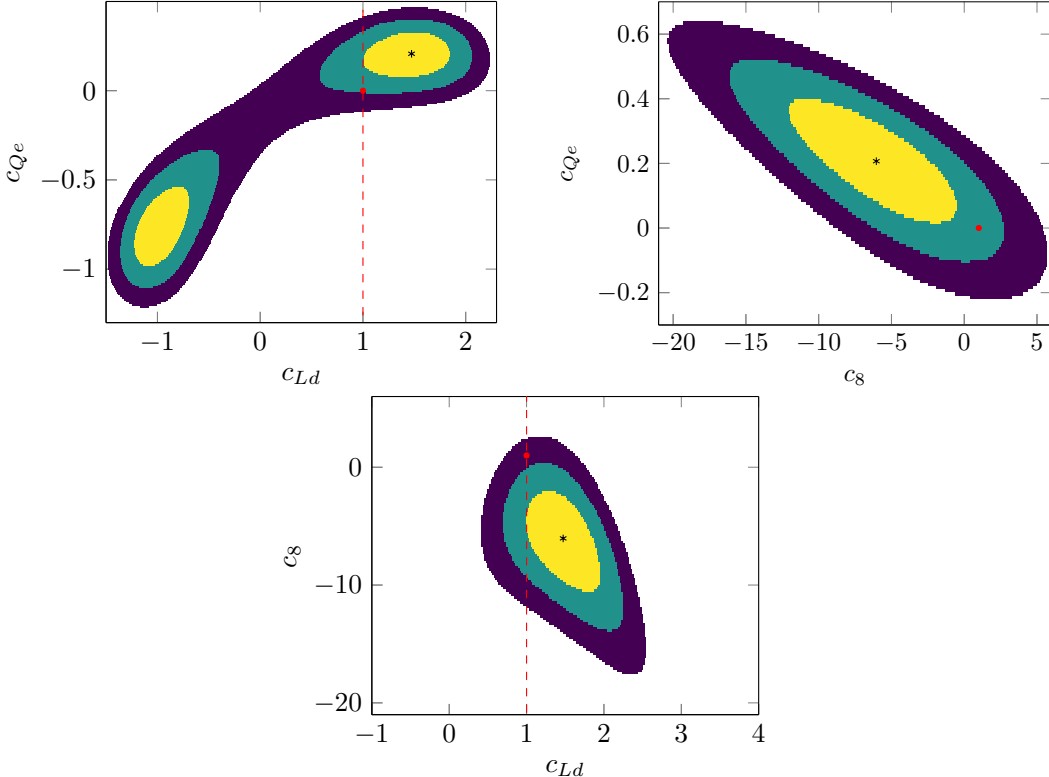

Figure 5: Plots of one-, two-, and three-sigma allowed regions of the Wilson coefficients $c_{Ld}$, $c_{Qe}$, and $c_8$ using the $\Phi$ model with $M = 3$ TeV and $Y_\Phi = 0.5$. The best fit point is labeled with $*$ and the theory prediction with $\bullet$ (red). The dashed line (red) indicates the theory prediction $c_{Ld} = 1$.

effect. When neglecting dimension-six squared contributions from the extra operator, we still find improvement in the fits. However, In contrast to the case $Y_\Phi = 0.1$, we find that the fits including the squared contributions of the extra dimension-six operator as well as the dimension-eight operators actually performs worse than when retaining the partial $1/\Lambda^4$ contributions and neglecting the dimension-eight operator coefficients. The reason for this is that the dimension-eight operator and extra dimension-six operator are both absorbing part of the signal that is missed by truncating at $\mathcal{O}(1/\Lambda^4)$, i.e. the extra dimension-six operator coefficient and the dimension-eight operators' coefficient are both playing the role of a nuisance parameter. This was not a problem for the more weakly interacting model with $Y_\Phi = 0.1$ as the truncation is more stable here.

Figure 5 shows a plot of the correlation between the dimension-eight operator coefficient, $c_8$, $c_{Ld}$, and $c_{Qe}$ for $Y_\Phi = 0.5$ and including the dimension-six squared contributions from $c_{Qe}$. The theory prediction, $\{c_{Ld} = 1, c_{Qe} = 0, c_8 = 1\}$ (red dot), is well outside the one-sigma contours. However, $c_{Ld} = 1$ (red dashed line) appears to be nearly consistent with the $1\sigma$ contours for values of $c_{Qe}$ and $c_8$ differing from the best fit point. Indeed, taking $c_8 = 1$ and $c_{Qe} = 0$ we have $c_{Ld} = 0.95 \pm 0.29$ with $\Delta\chi^2 = 0.35$, well within the one-sigma bound.

In this oversimplification of a full SMEFT fit, the extra dimension-six operator also plays the role of a nuisance parameter. This could be a potential hazard in a strictly dimension-six fit to a single channel. However, the majority of SMEFT bottom-up studies are global fits which use as much data as possible. If we imagine that our Drell-Yan measurement was complemented by another measurement constraining the extra dimension-six operator one would expect this would help to drive the Wilson coefficient to its theory value. If we reperform the same fit, but

Table 4: Example fits to two dimension-six operators for the $\Phi$ model as described in Tab. 3 but now including a fictitious constraint on the extra Wilson coefficient as described in the text.

| $M = 3$ TeV, $Y_\Phi = 0.5$ | | |
|---|---|---|
| | including d6$^2$ | |
| $c_8 = 0$ | $c_{Ld} = 0.9 \pm 0.3$ | $c_{LQ}^{(1)} = -0.01 \pm 0.05$ |
| $c_8 \neq 0$ | $c_{Ld} = 1.0 \pm 0.3$ | $c_{LQ}^{(1)} = 0.00 \pm 0.05$ |
| $c_8 = 0$ | $c_{Ld} = 0.9 \pm 0.3$ | $c_{LQ}^{(3)} = 0.01 \pm 0.07$ |
| $c_8 \neq 0$ | $c_{Ld} = 1.0 \pm 0.3$ | $c_{LQ}^{(3)} = 0.00 \pm 0.07$ |
| $c_8 = 0$ | $c_{Ld} = 0.93 \pm 0.2$ | $c_{eu} = 0.00 \pm 0.02$ |
| $c_8 \neq 0$ | $c_{Ld} = 1.0 \pm 0.3$ | $c_{eu} = 0.00 \pm 0.02$ |
| $c_8 = 0$ | $c_{Ld} = 0.9 \pm 0.3$ | $c_{ed} = 0.02 \pm 0.11$ |
| $c_8 \neq 0$ | $c_{Ld} = 1.0 \pm 0.3$ | $c_{ed} = 0.00 \pm 0.11$ |
| $c_8 = 0$ | $c_{Ld} = 0.9 \pm 0.3$ | $c_{Lu} = -0.01 \pm 0.07$ |
| $c_8 \neq 0$ | $c_{Ld} = 1.0 \pm 0.2$ | $c_{Lu} = 0.00 \pm 0.07$ |
| $c_8 = 0$ | $c_{Ld} = 0.9 \pm 0.3$ | $c_{Qe} = -0.01 \pm 0.09$ |
| $c_8 \neq 0$ | $c_{Ld} = 1.0 \pm 0.3$ | $c_{Qe} = 0.00 \pm 0.09$ |

add an extra term to the $\chi^2$ with a best fit value zero and error twice that coming from the dimension-six squared fit (neglecting the D8 operator) we find the results in Tab. 4. To clarify, the fit containing $\{c_{Ld}, c_{LQ}^{(1)}\}$ results in a best fit value for $c_{LQ}^{(1)}$ of $-0.01 \pm 0.05$ so we modify our $\chi^2$ as:

$$\chi^2 \to \chi^2 + \frac{(c_{LQ}^{(1)} - 0)^2}{(2 \cdot 0.05)^2}. \tag{47}$$

With this added "data" we find in all cases the extra dimension-six operator is driven to be consistent with the theory prediction, despite the relatively loose error assumed. We also find the best fit point $c_{Ld}$ coincides with the theory prediction.

We also perform a fit to all seven dimension-six Wilson coefficients simultaneously. As there are too many parameters the correct minimum cannot be found using Mathematica's limited minimization routines without the inclusion of dimension-six squared contributions, so we only consider this case. Including the squares of all dimension-six operators and neglecting dimension-eight operators we find:

$$\begin{aligned}
& c_{LQ}^{(1)} = 0.1 \pm 0.1\,, && c_{LQ}^{(3)} = 0.4 \pm 0.1\,, \\
& c_{eu} = -0.10 \pm 0.02\,, && c_{ed} = -0.4 \pm 0.1\,, \\
& c_{Lu} = -0.2 \pm 0.1\,, && c_{Ld} = 0.5 \pm 0.3\,, && c_{Qe} = -0.02 \pm 0.1\,.
\end{aligned} \tag{48}$$

And adding the dimension-eight operators to this:

$$\begin{aligned}
& c_{LQ}^{(1)} = 0.1 \pm 0.1\,, && c_{LQ}^{(3)} = 0.3 \pm 0.1\,, \\
& c_{eu} = -0.09 \pm 0.02\,, && c_{ed} = -0.4 \pm 0.1\,, \\
& c_{Lu} = 0.0 \pm 0.1\,, && c_{Ld} = 0.65 \pm 0.29\,, && c_{Qe} = -0.01 \pm 0.08\,.
\end{aligned} \tag{49}$$

Finally adding fictitious constraints as described above for all extra dimension-six operators we obtain:

$$
\begin{aligned}
c_{LQ}^{(1)} &= 0.0 \pm 0.1, & c_{LQ}^{(3)} &= 0.0 \pm 0.1, \\
c_{eu} &= 0.0 \pm 0.02, & c_{ed} &= 0.0 \pm 0.1, \\
c_{Lu} &= 0.0 \pm 0.1, & c_{Ld} &= 1.0 \pm 0.3, & c_{Qe} &= 0.0 \pm 0.1.
\end{aligned}
\tag{50}
$$

So we observe that for the inclusion of all seven operators we see the same behavior as in considering only one additional dimension-six operator. The inclusion of squares results in a fit that is not consistent with the UV theory. This result would seem to indicate an entirely different theory in the UV. Inclusion of the dimension-eight operators yields a largely similar fit as the extra parameters are all behaving as nuisance parameters, and finally the fictitious data meant to mimic a global fit results in excellent agreement with the theoretical prediction.

An important question to address is if the inclusion of the fictitious data without the dimension-eight operators performs just as well. Removing the dimension-eight operators yields the results:

$$
\begin{aligned}
c_{LQ}^{(1)} &= 0.0 \pm 0.1, & c_{LQ}^{(3)} &= 0.0 \pm 0.1, \\
c_{eu} &= 0.0 \pm 0.02, & c_{ed} &= 0.0 \pm 0.1, \\
c_{Lu} &= 0.0 \pm 0.1, & c_{Ld} &= 0.9 \pm 0.3, & c_{Qe} &= 0.0 \pm 0.1.
\end{aligned}
\tag{51}
$$

So the dimension-six operators which are not generated by the UV theory are still driven to their theory values, while the best fit value for $c_{Ld}$ is slightly pulled away from the theory value. This is consistent with the interpretation of the inclusion of dimension-eight operators as performing the role of nuisance parameters. The same analysis for $Y_\Phi = 10$ yields similar results.

Overall the results of our simplified analysis indicate:

1. Analyses of a single channel at dimension-six may result in results inconsistent with the actual UV realization as dimension-six operators not generated in the UV may misleadingly be playing the role of nuisance parameters. This appears be mitigated through global fits. This is already the practice in state of the art SMEFT bottom-up studies.

2. The dimension-six squared results may break approximate degeneracies in the parameter space allowing the fit to converge to the true values of the Wilson coefficients. This comes with a major caveat: recall that in the $U(1)$ mixing model dimension-six squared terms could actually hurt the convergence of the series (see Fig. 4).

3. Dimension-eight operators play the role of nuisance parameters which absorb our ignorance of higher order terms in the EFT expansion. As dimension-six squared terms improve the convergence for the $\Phi$ model this has a small impact relative to the size of the error in the measured Wilson coefficients.

### 4.2.2   *X* model

Unfortunately, it is very difficult to perform a similar analysis for the $X$ model as it generates all of the operators of Eq. 38 except that corresponding to the Wilson coefficient $Q_{LQ}^{(3)}$ (and the chiral flip operators). As such the most obvious approach would be to start with a six parameter fit which suffers from the same issue as the multiparameter fits for the SM and $\Phi$. That is, our minimization technique fails to converge to a minimum resembling the UV physics. Instead it picks a configuration in which various cancellations between the Wilson coefficients drives the best fit values away from the theory prediction and again falsely appears to predict a very different new physics scenario.

Instead we perform a two parameter fit where the dimension-six four-fermion operators appearing in Eq. 33 have a common Wilson coefficient and we also include $c_{LQ}^{(3)}$. Again we normalize the predicted operators so their Wilson coefficient $c_6$ should be 1, while we take:

$$c_{LQ}^{(3)} \to -\frac{g_1^2 \beta^2 c_{LQ}^{(3)}}{2M^2}. \tag{52}$$

We start with $M_X = 3$ TeV and $\beta = 0.3$ (Fig 6). Performing a fit excluding dimension-six squared contributions we find:

$$c_6 = 3.4 \pm 0.4, \qquad c_{LQ}^{(3)} = -9.2 \pm 1.4. \tag{53}$$

Again both Wilson coefficients appear to be statistically significant, but are also statistically far from their theory values. If we include the dimension-eight operators of Eq. 33 with a single coefficient we find:

$$c_6 = 0.54 \pm 0.36, \qquad c_{LQ}^{(3)} = 1.7 \pm 1.4. \tag{54}$$

The inclusion of the dimension-eight operator (and implicitly the dimension-six squared contributions for $c_6$) has ameliorated the situation slightly, but $c_6$ is still over one-sigma away from the theory value and $c_{LQ}^{(3)}$ also appears statistically nonzero. As was the case for $\Phi$ we infer that $c_{LQ}^{(3)}$ and $c_8$ are working in tandem as nuisance parameters.

Including the dimension-six squared contributions for $c_{LQ}^{(3)}$ while neglecting the dimension-eight operator gives:

$$c_6 = 2.3 \pm 0.4, \qquad c_{LQ}^{(3)} = -4.7 \pm 1.4. \tag{55}$$

We see a slight improvement of the fit from Eq. 53 where no quadratic terms were included. This is in contrast with Tab. 2 and Fig. 6 where the dimension-six squared contribution slightly hurt the fit. This supports the idea that the extra dimension-six operator is behaving as a nuisance parameter absorbing some of the physics neglected by truncation. However, it should be noted this two-parameter fit is overall much worse than the one-parameter fit in the table. This can be understood as $c_{LQ}^{(3)}$ does not have the correct kinematics to absorb these effects. Adding the dimension-eight operator then gives:

$$c_6 = 0.54 \pm 0.36, \qquad c_{LQ}^{(3)} = 1.7 \pm 1.4. \tag{56}$$

The fit has vastly improved, although $c_6$ and $c_{LQ}^{(3)}$ are still just over one-sigma away from their theory values. This improvement can be attributed to the fact the dimension-eight operator has some of the missing kinematics allowing its Wilson coefficient to absorb the physics neglected by truncating the EFT expansion. However, $c_{LQ}^{(3)}$ and $c_8$ are competing to absorb that physics resulting in the skewing of $c_6$ and $c_{LQ}^{(3)}$ away from their theory values. Notice that this fit gives the same results (with our rounding, slightly better when including further digits) as Eq. 54 indicating for this model inclusion of the dimension-eight operator is in fact driving the improvement.

Supposing some other experiment provides the constraint $c_{LQ}^{(3)} = 0 \pm 2.8$. This vastly improves our result to,

$$c_6 = 1.0 \pm 0.4, \qquad c_{LQ}^{(3)} = 0 \pm 1.4, \tag{57}$$

indicating any interpretation of the SMEFT should be made from a global fit, and not based on individual channels.

We can contrast the above with the case $M_X = 3$ TeV with $\beta = 1.2$. In this case, shown in Fig. 4, the dimension-six squared contribution actually hurts the fit. Fitting first only linearly in the dimension-six operators:

$$c_6 = 0.99 \pm 0.02, \qquad c_{LQ}^{(3)} = 0.09 \pm 0.08. \tag{58}$$

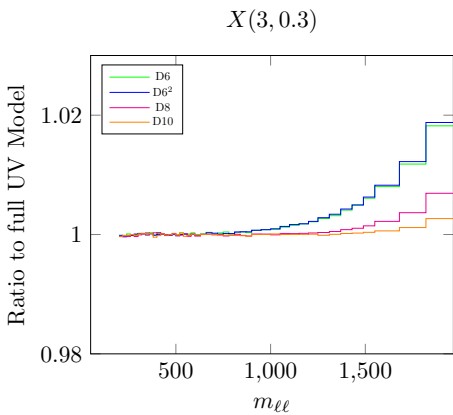

Figure 6: Ratio of $X$ model prediction to the SM for $M_X = 3$ TeV with $\beta = 0.3$. See caption of Fig. 4 for further details.

This agrees excellently with the theory prediction, we will see it actually is the best fit we obtain without including some (fictitious) extra measurement of $c_{LQ}^{(3)}$. The inclusion of the dimension-eight operators yields:

$$c_6 = 1.5 \pm 0.03\,, \qquad c_{LQ}^{(3)} = -1.8 \pm 0.1\,. \tag{59}$$

The best fit point for both Wilson coefficients have moved significantly away from the theory values. Adding the square of $c_{LQ}^{(3)}$ while neglecting dimension-eight contributions does not improve the fit:

$$c_6 = 0.59 \pm 0.02\,, \qquad c_{LQ}^{(3)} = 1.8 \pm 0.1\,. \tag{60}$$

Nor does the inclusion of both order $1/\Lambda^4$ contributions:

$$c_6 = 1.5 \pm 0.02\,, \qquad c_{LQ}^{(3)} = -1.8 \pm 0.1\,. \tag{61}$$

However, if an experiment were to measure $c_{LQ}^{(3)} = 0 \pm 0.2$ we would obtain:

$$c_6 = 1.021 \pm 0.024\,, \qquad c_{LQ}^{(3)} = 0 \pm 0.1\,, \qquad c_8 = 0.6 \pm 0.1\,. \tag{62}$$

This is in good agreement with the theory results for the dimension-six operators. We have included the dimension-eight operator coefficient constraints in the case to show that the result appears significant. However, we must bear in mind that the dimension-eight operator is behaving as a nuisance parameter and therefore this seemingly significant result should be neglected. The case $M_X = 3$ TeV with $\beta = 1.2$ stands in stark contrast with the other benchmark as well as the case of $\Phi$. By some accident the dimension-six operator fit actually outperforms all other considered fits. This can be understood immediately from Fig. 4 where we see that the dimension-six contribution more closely reproduces the full result than any of the other orders in $1/M_X^2$ considered. This example also further demonstrates the need to take care when including partial orders in $1/M^2$ in fits to the SMEFT.

## 5  Conclusions

We considered four separate ultraviolet extensions of the SM and matched them to dimension-ten. This was facilitated by remaining agnostic to the operator basis. We found that the case of the $(1,3)_0$ scalar, $\phi$, does not affect the Drell Yan process and is an example of a UV Model

where the truncation at dimension-six or inclusion of dimension-six squared contributions to the cross section has no impact on the fit procedure. The fermion, $\chi$, with charges $(1,1)_{-1}$ appeared to generate a momentum expansion. However, use of the EOM or field redefinitions allows all of these operators to be traded for operators in the vev expansion. As such, this model only affects low invariant mass bins for the Drell Yan process and therefore we concluded it is best constrained by EWPD. The cases of $\Phi = (3,2)_{1/6}$ and the U(1) mixing model ($X$) nicely exhibited a momentum expansion.

Next we studied the convergence of the $\Phi$ and $X$ models from a strictly top down perspective. We calculated the cross section as a function of invariant-mass bin for both the full UV model and the IR model truncated at a given order, including partial orders. We then compared the results of the two by performing a simple $\chi^2$ fit at a given order in the EFT expansion. We found that in the more strongly interacting cases and for lower masses the truncation at dimension-six fails. The inclusion of dimension-six squared contributions always helped the convergence in the case of $\Phi$ as it has the same sign contribution as the dimension-ten operator contribution which is opposite that of the purely $1/\Lambda^2$ contribution. In contrast, in the $X$ model the parameters can conspire such that the dimension-six squared term helps or hurts the fit. The parameters in a given benchmark model can actually cause the dimension-six term to better reproduce the full UV result than even the inclusion of terms up to dimension-ten. We also found that when a given order was insufficient to correctly determine the Wilson coefficients, the next order in the expansion performed the role of nuisance parameters absorbing our ignorance of the higher order terms contributing. This was most clearly demonstrated in the strongest interacting models where the determination of the dimension-eight operator coefficient was statistically significant, but the best fit point did not agree with the theory value.

Next we found that, even after limiting ourselves to strictly four-fermion operators, a bottom up SMEFT analysis of our UV models was difficult. This is because when fitting the SMEFT to our UV models the dimension-six operators exist in a strongly correlated parameter space and the minimization methods tested (those standard in Mathematica) appear to fail to find the true minimum. As a result, we largely limited ourselves to two-parameter fits which included the operator(s) generated in the UV with a single common Wilson coefficient and an additional SMEFT operator. When comparing the inclusion or exclusion of dimension-six squared as well as dimension-eight operators, we found that these additional operators filled the role of nuisance parameters. This had the effect that, in order to fit the new physics not accounted for at a given order in the expansion, our best fit values were skewed from the theory values (including the dimension-six operator generated in the UV). Dimension-six squared contributions again helped in the case of $\Phi$ but were not as dependable in the two benchmark $X$ models considered. Our method of determining the one-sigma error in the operator coefficients gave the impression that the theory values of the Wilson coefficients were not within one-sigma of the best fit point. However, in the case of the $\Phi$ model with $M_\Phi = 3$ TeV with $Y_\Phi = 0.5$ a more careful analysis demonstrated that this was a reflection of the strongly correlated parameter space, and the theory values were indeed consistent. The simple addition of a fictitious measurement of the operator not generated by the UV model with central value 0 and error twice the size of that resulting from the dimension-six squared fit immediately moved the best fit values to the theory values. This is promising as this issue with the central values appearing to differ significantly from the theory values may not be present for global fits.

Based on the discussion above and throughout the article we highlight some considerations that should be kept in mind for future work in the SMEFT.

1. For more weakly interacting and/or more strongly decoupled theories a strictly dimension-six fit appears to converge well to the theory values, given the large uncertainties.

2. Analyses which choose to include partial $1/\Lambda^4$ results, i.e. dimension-six squared contributions, should be accompanied by analyses properly truncated at a given order. The "consistency" of these contributions with UV physics is model dependent.

3. From a top down perspective, the highest order in the EFT considered appears to behave as nuisance parameters. In this context, a strictly dimension-six analysis behaves as a SM fit with errors defined consistent with the SMEFT approach. Notice that with a higher integrated luminosity, while not possible at the LHC, the error in our fits would shrink and the skewed dimension-six results would fail. Similarly, a dimension-eight analysis behaves as a dimension-six fit with the dimension-eight operator coefficients behaving as nuisance parameters which absorb our ignorance of higher order terms in the expansion. From a bottom up perspective, lower dimensional operators not generated by the ultraviolet physics also behave as nuisance parameters and this may skew the leading order results. For the more viable models considered which have weaker interactions/higher masses, issues with truncation could become a problem for a next generation precision experiment.

4. Single channel analyses may suffer as a result of the previous point. The results may be skewed from their theory values and the results may appear significant. Global fits are an industry standard for phenomenological studies and our simplified attempt to see the effects of including additional data (fictitious in our examples) appears to indicate that this problem will not persist in the context of global fits. It is important to note that as experimental analyses begin to perform detailed SMEFT analyses this issue could be present. Analyses of individual experimental channels should not be interpreted alone, but in the context of a global fit. Further, it is absolutely crucial that correlation matrices be included in experimental results (or any single/few channel determination) as the theory value which is not consistent with individual Wilson coefficient limits may be perfectly consistent when correlations are included.

While beyond the scope of this article, some future considerations that would further improve our understanding of the convergence of the SMEFT expansion include adding additional channels (for example foward-backward asymmetries in Drell Yan, EWPD, or other channels unrelated to Drell Yan), optimizing binning in the kinematic variables, exploring other distributions [65,66], and the inclusion of pdf fits as proposed in [45]. The inclusion of other distributions may have an interesting impact on the interpretation of higher dimension operators as nuisance parameters as at a given order they may not be able to as readily absorb affects such as angular distributions. Further, including loops in the UV and IR would not only result in more events and better statistics, it could also elucidate the role of the loop expansion in SMEFT analyses for realistic UV models. Such an analysis would have to include far more free parameters making such a study very difficult.

# Acknowledgments

The author thanks Adam Martin for his reading and feedback of the manuscript. The author also extends great thanks to Anders Eller Thomsen and Javier Fuentes-Martín for help with Matchete and discussions of its use.

# A Field redefinitions of the $U(1)$ mixing model

The $B$ and $V$ dependent Lagrangian (neglecting covariant derivative dependence on $B$) is:

$$\mathcal{L}_{BV} = -\frac{1}{4}B_{\mu\nu}B^{\mu\nu} - \frac{1}{4}V_{\mu\nu}V^{\mu\nu} - \frac{k}{2}B_{\mu\nu}X^{\mu\nu} + \frac{1}{2}M^2 V_\mu V^\mu \tag{A.1}$$

$$= -\frac{1}{4}\begin{pmatrix} B_{\mu\nu} \\ V_{\mu\nu} \end{pmatrix}^T \begin{pmatrix} 1 & k \\ k & 1 \end{pmatrix} \begin{pmatrix} B^{\mu\nu} \\ V^{\mu\nu} \end{pmatrix} + \frac{1}{2}M^2 V_\mu V^\mu . \tag{A.2}$$

The transformation,

$$\begin{pmatrix} B_\mu \\ V_\mu \end{pmatrix} \rightarrow \begin{pmatrix} \alpha & 0 \\ \beta & 1 \end{pmatrix} \begin{pmatrix} B'_\mu \\ V'_\mu \end{pmatrix}, \tag{A.3}$$

diagonalizes the kinetic mixing given,

$$\alpha = \frac{1}{\sqrt{1-k^2}}, \tag{A.4}$$

$$\beta = \frac{-k}{\sqrt{1-k^2}}. \tag{A.5}$$

This transformation results, however, in mass mixing of the form:

$$V^\mu V_\mu \rightarrow V'^\mu V'_\mu + 2\beta V'^\mu B'_\mu + \beta^2 B'_\mu B'^\mu . \tag{A.6}$$

Another field redefinition of the form,

$$\begin{pmatrix} B'_\mu \\ V'_\mu \end{pmatrix} \rightarrow \begin{pmatrix} c & s \\ -s & c \end{pmatrix} \begin{pmatrix} B''_\mu \\ V''_\mu \end{pmatrix}, \tag{A.7}$$

then diagonalizes the mass mixing for $s = -k$ and $c = \sqrt{1-k^2}$. Note this rotation commutes with that of Eq. A.3 and therefore does not affect the diagonalization of the kinetic terms. This results in a mass for $V''$,

$$M_{V''}^2 = \frac{M_V^2}{1-k^2}, \tag{A.8}$$

while leaving the $B''$ field massless (above EWSB). In the main text we refer to $V''$ as $X$.

Taking a $+$ sign convention for the covariant derivative and $V'' \rightarrow X$, this transformation gives:

$$(D_\mu H) \rightarrow (D_\mu H) + ig_1 Y_H \beta X_\mu H , \tag{A.9}$$

$$(D_\mu H)^\dagger (D_\mu H) \rightarrow (D_\mu H)^\dagger (D_\mu H) - g_1 Y_H \beta (H^\dagger i \overleftrightarrow{D}_\mu H) X^\mu + g_1^2 Y_H^2 \beta^2 (H^\dagger H) X_\mu X^\mu , \tag{A.10}$$

$$(D_\mu \psi) \rightarrow (D_\mu \psi) + ig_1 Y_\psi \beta X_\mu \psi , \tag{A.11}$$

$$i\bar{\psi}\slashed{D}\psi \rightarrow i\bar{\psi}\slashed{D}\psi - g_1 Y_\psi \beta X_\mu \bar{\psi}\gamma_\mu \psi . \tag{A.12}$$

# B Parameterization of SMEFT four-fermion operator contributions

We parameterize the full cross section bin-by-bin in $m_{\ell\ell}$ as follows:

$$
\begin{aligned}
\sigma = \sigma_0 &+ \left[ g_L^e \left( a_{dZ} g_L^d \sigma_{dZ} + a_{uZ} g_L^u \sigma_{uZ} \right) + e^2 Q_e \left( a_{dA} Q_d \sigma_{dA} + a_{uA} Q_u \sigma_{uA} \right) \right] c_{LQ}^{(1)} \\
&+ \left[ g_L^e \left( a_{dZ} g_L^d \sigma_{dZ} - a_{uZ} g_L^u \sigma_{uZ} \right) + e^2 Q_e \left( a_{dA} Q_d \sigma_{dA} - a_{uA} Q_u \sigma_{uA} \right) \right] \frac{c_{LQ}^{(3)}}{4} \\
&+ \left[ a_{uZ} g_R^e g_R^u \sigma_{uZ} + a_{uA} e^2 Q_e Q_u \sigma_{uA} \right] c_{eu} + \left[ a_{dZ} g_R^e g_R^d \sigma_{dZ} + a_{dA} e^2 Q_e Q_d \sigma_{dA} \right] c_{ed} \\
&+ \left[ a_{uZ} g_L^e g_R^u \sigma_{uZ} + a_{uA} e^2 Q_e Q_u \sigma_{uA} \right] c_{Lu} + \left[ a_{dZ} g_L^e g_R^d \sigma_{dZ} + a_{dA} e^2 Q_e Q_d \sigma_{dA} \right] c_{Ld} \\
&+ \left[ g_R^e \left( a_{dZ} g_L^d \sigma_{dZ} + a_{uZ} g_L^u \sigma_{uZ} \right) + e^2 Q_e \left( a_{dA} Q_d \sigma_{dA} + a_{uA} Q_u \sigma_{uA} \right) \right] c_{Qe} \\
&+ \left[ (b_d \sigma_{d\hat{s}} + b_u \sigma_{u\hat{s}}) \left( c_{LQ}^{(1)} + \frac{1}{16} c_{LQ}^{(3)} \right) + (b_d \sigma_{d\hat{s}} - b_u \sigma_{u\hat{s}}) \frac{1}{2} c_{LQ}^{(1)} c_{LQ}^{(3)} + b_u \sigma_{u\hat{s}} c_{eu}^2 + b_d \sigma_{d\hat{s}} c_{ed}^2 \right] \\
&+ \left[ b_u \sigma_{u\hat{s}} c_{Lu}^2 + b_d \sigma_{d\hat{s}} c_{Ld}^2 + (b_u \sigma_{u\hat{s}} + b_d \sigma_{d\hat{s}}) c_{Qe}^2 \right],
\end{aligned}
\tag{B.1}
$$

where $\sigma_0$ is the SM cross section for a given invariant mass bin. The $\sigma_{qZ}$ for $q = \{u, d\}$ correspond to the pdf integration for the lowest invariant mass bin of,

$$
\hat{\sigma}_{qZ} = \frac{2}{48\pi N_c} \frac{\hat{s}(\hat{s} - m_Z^2)}{(\hat{s} - m_Z^2)^2 + \Gamma_Z^2 m_Z^2},
\tag{B.2}
$$

the $\sigma_{qA}$ correspond to the integration of the partonic cross section,

$$
\hat{\sigma}_{qA} = \frac{2}{48\pi N_c},
\tag{B.3}
$$

and the $\sigma_{q\hat{s}}$ correspond to the integration of,

$$
\hat{\sigma}_{q\hat{s}} = \frac{\hat{s}}{48\pi N_c}.
\tag{B.4}
$$

These correspond to, up to their normalization, the kinematic part of the $1/\Lambda^2$ and $1/\Lambda^4$ squared amplitudes. This choice of normalization, along with the explicit SM coupling dependence of Eq. B.1 requires that the $a_i$ and $b$ are identically one for the lowest invariant mass. The $a_i$ and $b$ are then determined for each subsequent bin. Due to the normalization to the $\sigma_i$ this demonstrates how each bin compares with the bin with lowest $m_{\ell\ell}$. The $a_i$ and $b$ for each bin can be found in Tab. 5. **Please note: the table does not contain nearly enough significant digits for the applications described in the main text. The full tables used including error in the integration can be requested from the author.**

Table 5: Values of the $a_i$ and the $b_i$ for a given invariant mass bin $m_{\ell\ell}$.

| $m_{\ell\ell}$ | $\sigma_0$ | $a_{dZ}$ | $a_{dA}$ | $a_{uZ}$ | $a_{uA}$ | $b_d$ | $b_u$ | $m_{\ell\ell}$ | $\sigma_0$ | $a_{dZ}$ | $a_{dA}$ | $a_{uZ}$ | $a_{uA}$ | $b_d$ | $b_u$ |
|---|---|---|---|---|---|---|---|---|---|---|---|---|---|---|---|
| $200-220$ | 0.69 | 1 | 1 | 1 | 1 | 1 | 1 | $810-840$ | $2.0\cdot10^{-3}$ | 0.036 | 0.044 | 0.045 | 0.054 | 0.68 | 0.84 |
| $220-240$ | 0.46 | 0.78 | 0.81 | 0.79 | 0.82 | 0.98 | 0.99 | $840-870$ | $1.6\cdot10^{-3}$ | 0.032 | 0.040 | 0.040 | 0.049 | 0.66 | 0.81 |
| $240-260$ | 0.31 | 0.63 | 0.67 | 0.64 | 0.69 | 0.95 | 0.97 | $870-900$ | $1.4\cdot10^{-3}$ | 0.029 | 0.036 | 0.036 | 0.044 | 0.63 | 0.80 |
| $260-280$ | 0.22 | 0.51 | 0.56 | 0.53 | 0.58 | 0.93 | 0.96 | $900-950$ | $1.8\cdot10^{-3}$ | 0.042 | 0.052 | 0.053 | 0.065 | 1.0 | 1.3 |
| $280-300$ | 0.16 | 0.43 | 0.47 | 0.44 | 0.49 | 0.90 | 0.94 | $950-1000$ | $1.4\cdot10^{-3}$ | 0.036 | 0.044 | 0.045 | 0.056 | 0.94 | 1.2 |
| $300-320$ | 0.12 | 0.36 | 0.40 | 0.38 | 0.42 | 0.88 | 0.93 | $1000-1050$ | $1.1\cdot10^{-3}$ | 0.030 | 0.037 | 0.039 | 0.048 | 0.89 | 1.1 |
| $320-340$ | 0.091 | 0.30 | 0.35 | 0.32 | 0.37 | 0.86 | 0.91 | $1050-1100$ | $8.6\cdot10^{-4}$ | 0.026 | 0.032 | 0.034 | 0.041 | 0.83 | 1.1 |
| $340-360$ | 0.070 | 0.26 | 0.30 | 0.28 | 0.32 | 0.84 | 0.89 | $1100-1150$ | $6.8\cdot10^{-4}$ | 0.022 | 0.027 | 0.029 | 0.036 | 0.78 | 1.0 |
| $360-380$ | 0.054 | 0.23 | 0.26 | 0.24 | 0.28 | 0.81 | 0.88 | $1150-1200$ | $5.4\cdot10^{-4}$ | 0.019 | 0.023 | 0.025 | 0.031 | 0.74 | 0.97 |
| $380-400$ | 0.043 | 0.20 | 0.23 | 0.21 | 0.25 | 0.79 | 0.86 | $1200-1250$ | $4.3\cdot10^{-4}$ | 0.017 | 0.020 | 0.022 | 0.027 | 0.69 | 0.92 |
| $400-420$ | 0.034 | 0.17 | 0.20 | 0.19 | 0.22 | 0.77 | 0.85 | $1250-1310$ | $4.1\cdot10^{-4}$ | 0.017 | 0.021 | 0.023 | 0.028 | 0.77 | 1.0 |
| $420-440$ | 0.028 | 0.15 | 0.18 | 0.17 | 0.20 | 0.75 | 0.83 | $1310-1370$ | $3.2\cdot10^{-4}$ | 0.014 | 0.018 | 0.019 | 0.024 | 0.72 | 0.97 |
| $440-460$ | 0.022 | 0.13 | 0.16 | 0.15 | 0.18 | 0.73 | 0.81 | $1370-1430$ | $2.5\cdot10^{-4}$ | 0.012 | 0.015 | 0.017 | 0.021 | 0.66 | 0.91 |
| $460-480$ | 0.018 | 0.12 | 0.14 | 0.13 | 0.16 | 0.71 | 0.80 | $1430-1490$ | $2.0\cdot10^{-4}$ | 0.010 | 0.013 | 0.014 | 0.018 | 0.62 | 0.85 |
| $480-500$ | 0.015 | 0.11 | 0.13 | 0.12 | 0.14 | 0.70 | 0.78 | $1490-1550$ | $1.6\cdot10^{-4}$ | $8.8\cdot10^{-3}$ | 0.011 | 0.012 | 0.015 | 0.57 | 0.80 |
| $500-520$ | 0.013 | 0.096 | 0.11 | 0.11 | 0.13 | 0.68 | 0.77 | $1550-1680$ | $2.4\cdot10^{-4}$ | 0.015 | 0.019 | 0.021 | 0.026 | 1.1 | 1.6 |
| $520-540$ | 0.011 | 0.086 | 0.10 | 0.099 | 0.12 | 0.66 | 0.75 | $1680-1820$ | $1.6\cdot10^{-4}$ | 0.012 | 0.014 | 0.017 | 0.021 | 0.99 | 1.4 |
| $540-560$ | $8.9\cdot10^{-3}$ | 0.078 | 0.094 | 0.090 | 0.11 | 0.64 | 0.74 | $1820-1970$ | $1.0\cdot10^{-4}$ | $8.7\cdot10^{-3}$ | 0.011 | 0.013 | 0.016 | 0.87 | 1.3 |
| $560-580$ | $7.6\cdot10^{-3}$ | 0.071 | 0.085 | 0.082 | 0.099 | 0.63 | 0.72 | $1970-2210$ | $9.1\cdot10^{-5}$ | $8.8\cdot10^{-3}$ | 0.011 | 0.014 | 0.017 | 1.1 | 1.7 |
| $580-600$ | $6.5\cdot10^{-3}$ | 0.064 | 0.077 | 0.075 | 0.090 | 0.61 | 0.71 | $2210-6070$ | $8.5\cdot10^{-5}$ | 0.011 | 0.014 | 0.021 | 0.037 | 2.2 | 4.7 |
| $600-630$ | $7.9\cdot10^{-3}$ | 0.086 | 0.10 | 0.10 | 0.12 | 0.89 | 1.0 | | | | | | | | |
| $630-660$ | $6.4\cdot10^{-3}$ | 0.075 | 0.090 | 0.088 | 0.11 | 0.85 | 1.0 | | | | | | | | |
| $660-690$ | $5.1\cdot10^{-3}$ | 0.066 | 0.080 | 0.078 | 0.095 | 0.82 | 0.98 | | | | | | | | |
| $690-720$ | $4.2\cdot10^{-3}$ | 0.058 | 0.070 | 0.069 | 0.084 | 0.79 | 0.95 | | | | | | | | |
| $720-750$ | $3.4\cdot10^{-3}$ | 0.051 | 0.062 | 0.062 | 0.075 | 0.76 | 0.92 | | | | | | | | |
| $750-780$ | $2.8\cdot10^{-3}$ | 0.045 | 0.055 | 0.055 | 0.067 | 0.74 | 0.89 | | | | | | | | |
| $780-810$ | $2.3\cdot10^{-3}$ | 0.040 | 0.049 | 0.050 | 0.060 | 0.71 | 0.87 | | | | | | | | |

# C   Full tables

Table 6: Table of fit $\chi^2$ fit values for the $\Phi$ model as described in Sec. 4 and Tab. 1. Table is continued below.

| $M_\Phi$ | $Y_\Phi$ | dim | $c_6$ | $\delta c_6$ | $c_8$ | $\delta c_8$ | $c_{10}$ | $\delta c_{10}$ | $\chi^2_{\min}$ |
|---|---|---|---|---|---|---|---|---|---|
| 3 | 1 | D6 | 0.93424 | 5.60594 | - | - | - | - | $10^{-4}$ |
| - | - | D6$^2$ | 0.9448 | 5.74959 | - | - | - | - | $10^{-4}$ |
| - | - | D8 | 0.993788 | 5.75154 | 0.743965 | 57.4362 | - | - | $10^{-6}$ |
| - | - | D6D8 | 0.995559 | 5.73829 | 0.806826 | 61.2331 | - | - | $10^{-6}$ |
| - | - | D10 | 0.999841 | 5.73613 | 0.967029 | 61.2505 | 0.579804 | 252.998 | $10^{-8}$ |
| 3 | 2 | D6 | 0.909417 | 1.40087 | - | - | - | - | $10^{-2}$ |
| - | - | D6$^2$ | 0.951751 | 1.52662 | - | - | - | - | $10^{-3}$ |
| - | - | D8 | 0.989425 | 1.52705 | 0.587535 | 14.3434 | - | - | $10^{-4}$ |
| - | - | D6D8 | 0.996651 | 1.51446 | 0.840976 | 18.9536 | - | - | $10^{-4}$ |
| - | - | D10 | 0.9988 | 1.5134 | 0.924136 | 18.9663 | 0.24698 | 63.169 | $10^{-6}$ |
| 3 | 3 | D6 | 0.86835 | 0.622225 | - | - | - | - | $10^{-1}$ |
| - | - | D6$^2$ | 0.963071 | 0.744628 | - | - | - | - | $10^{-3}$ |
| - | - | D8 | 0.982763 | 0.742972 | 0.347476 | 6.36636 | - | - | $10^{-3}$ |
| - | - | D6D8 | 0.999257 | 0.733219 | 0.929252 | 12.9978 | - | - | $10^{-3}$ |
| - | - | D10 | 0.998047 | 0.733737 | 0.871944 | 12.9849 | -0.137005 | 28.0337 | $10^{-5}$ |
| 3 | 4 | D6 | 0.810801 | 0.349784 | - | - | - | - | $10^{-1}$ |
| - | - | D6$^2$ | 0.971767 | 0.457819 | - | - | - | - | $10^{-2}$ |
| - | - | D8 | 0.973297 | 0.45712 | 0.045924 | 3.57801 | - | - | $10^{-2}$ |
| - | - | D6D8 | 0.991193 | 0.458877 | 0.649279 | 9.14815 | - | - | $10^{-2}$ |
| - | - | D10 | 0.996819 | 0.457252 | 0.801076 | 9.09787 | -0.429437 | 15.7581 | $10^{-4}$ |
| 3 | 5 | D6 | 0.737046 | 0.223775 | - | - | - | - | $10^{0}$ |
| - | - | D6$^2$ | 0.957963 | 0.295181 | - | - | - | - | $10^{-1}$ |
| - | - | D8 | 0.961134 | 0.300567 | -0.28196 | 2.29139 | - | - | $10^{-2}$ |
| - | - | D6D8 | 0.992251 | 0.311135 | 0.601045 | 2.78106 | - | - | $10^{-2}$ |
| - | - | D10 | 0.995231 | 0.312243 | 0.717649 | 2.76599 | -0.427874 | 10.1 | $10^{-4}$ |
| 3 | 6 | D6 | 0.647671 | 0.155423 | - | - | - | - | $10^{1}$ |
| - | - | D6$^2$ | 0.916583 | 0.185084 | - | - | - | - | $10^{-1}$ |
| - | - | D8 | 0.946374 | 0.189307 | -0.592875 | 1.59667 | - | - | $10^{-1}$ |
| - | - | D6D8 | 0.993485 | 0.205298 | 0.636112 | 1.0534 | - | - | $10^{-1}$ |
| - | - | D10 | 0.993361 | 0.205093 | 0.628639 | 1.0536 | 0.0488359 | 7.0515 | $10^{-3}$ |
| 3 | 7 | D6 | 0.543721 | 0.114306 | - | - | - | - | $10^{1}$ |
| - | - | D6$^2$ | 0.875785 | 0.114358 | - | - | - | - | $10^{0}$ |
| - | - | D8 | 0.929116 | 0.112274 | -0.836343 | 1.18199 | - | - | $10^{-1}$ |
| - | - | D6D8 | 0.991343 | 0.124374 | 0.647188 | 0.503273 | - | - | $10^{-1}$ |
| - | - | D10 | 0.991152 | 0.121127 | 0.540727 | 0.503405 | 1.11525 | 5.23869 | $10^{-3}$ |
| 3 | 8 | D6 | 0.426885 | 0.0877142 | - | - | - | - | $10^{2}$ |
| - | - | D6$^2$ | 0.851512 | 0.0708649 | - | - | - | - | $10^{0}$ |
| - | - | D8 | 0.909601 | 0.0672631 | -0.957877 | 0.916951 | - | - | $10^{0}$ |
| - | - | D6D8 | 0.985744 | 0.0733825 | 0.639042 | 0.278839 | - | - | $10^{0}$ |
| - | - | D10 | 0.988637 | 0.0695142 | 0.462873 | 0.277806 | 2.75038 | 4.08696 | $10^{-2}$ |
| 3 | 9 | D6 | 0.299459 | 0.0695681 | - | - | - | - | $10^{2}$ |
| - | - | D6$^2$ | 0.841054 | 0.0448985 | - | - | - | - | $10^{0}$ |
| - | - | D8 | 0.888191 | 0.0425246 | -0.903184 | 0.73905 | - | - | $10^{0}$ |
| - | - | D6D8 | 0.979356 | 0.0450526 | 0.623438 | 0.170961 | - | - | $10^{0}$ |
| - | - | D10 | 0.985899 | 0.0419425 | 0.404659 | 0.169595 | 4.7407 | 3.3208 | $10^{-2}$ |
| 3 | 10 | D6 | 0.164122 | 0.056661 | - | - | - | - | $10^{2}$ |
| - | - | D6$^2$ | 0.838137 | 0.0295516 | - | - | - | - | $10^{1}$ |
| - | - | D8 | 0.865272 | 0.0285521 | -0.621193 | 0.615176 | - | - | $10^{1}$ |
| - | - | D6D8 | 0.97485 | 0.0292257 | 0.611437 | 0.11285 | - | - | $10^{1}$ |
| - | - | D10 | 0.982939 | 0.0270792 | 0.376095 | 0.11177 | 6.61241 | 2.79402 | $10^{-1}$ |
| 4 | 1 | D6 | 0.960669 | 9.96675 | - | - | - | - | $10^{-5}$ |
| - | - | D6$^2$ | 0.966612 | 10.1325 | - | - | - | - | $10^{-5}$ |
| - | - | D8 | 0.997753 | 10.1338 | 0.839327 | 181.555 | - | - | $10^{-7}$ |
| - | - | D6D8 | 0.99835 | 10.1256 | 0.877693 | 188.152 | - | - | $10^{-7}$ |
| - | - | D10 | 1.00011 | 10.1247 | 0.994565 | 188.164 | 0.773141 | 1421.77 | $10^{-8}$ |
| 4 | 2 | D6 | 0.945321 | 2.49102 | - | - | - | - | $10^{-3}$ |
| - | - | D6$^2$ | 0.969123 | 2.62174 | - | - | - | - | $10^{-4}$ |
| - | - | D8 | 0.996166 | 2.62254 | 0.7365 | 45.3568 | - | - | $10^{-6}$ |
| - | - | D6D8 | 0.99857 | 2.6146 | 0.8894 | 52.6731 | - | - | $10^{-6}$ |
| - | - | D10 | 0.999855 | 2.61396 | 0.975652 | 52.6839 | 0.51242 | 355.143 | $10^{-7}$ |
| 4 | 3 | D6 | 0.919782 | 1.10667 | - | - | - | - | $10^{-2}$ |
| - | - | D6$^2$ | 0.973352 | 1.23226 | - | - | - | - | $10^{-3}$ |
| - | - | D8 | 0.993603 | 1.23218 | 0.572117 | 20.1385 | - | - | $10^{-5}$ |
| - | - | D6D8 | 0.999081 | 1.22494 | 0.915512 | 28.996 | - | - | $10^{-5}$ |
| - | - | D10 | 0.999549 | 1.22472 | 0.948031 | 29.0017 | 0.160245 | 157.659 | $10^{-7}$ |
| 4 | 4 | D6 | 0.883971 | 0.6222 | - | - | - | - | $10^{-1}$ |
| - | - | D6$^2$ | 0.978711 | 0.744754 | - | - | - | - | $10^{-3}$ |
| - | - | D8 | 0.989971 | 0.743771 | 0.355433 | 11.3164 | - | - | $10^{-4}$ |
| - | - | D6D8 | 0.999731 | 0.738418 | 0.960993 | 23.114 | - | - | $10^{-4}$ |
| - | - | D10 | 0.999171 | 0.738616 | 0.916416 | 23.1034 | -0.187235 | 88.5858 | $10^{-6}$ |

Table 7: As described in Tab. 6.

| $M_\Phi$ | $Y_\Phi$ | dim | $c_6$ | $\delta c_6$ | $c_8$ | $\delta c_8$ | $c_{10}$ | $\delta c_{10}$ | $\chi^2_{\min}$ |
|---|---|---|---|---|---|---|---|---|---|
| 4 | 5 | D6 | 0.838033 | 0.39802 | - | - | - | - | $10^{-1}$ |
| - | - | D6² | 0.983155 | 0.51164 | - | - | - | - | $10^{-3}$ |
| - | - | D8 | 0.985459 | 0.510922 | 0.100735 | 7.23776 | - | - | $10^{-3}$ |
| - | - | D6D8 | 0.997768 | 0.510438 | 0.847198 | 19.7422 | - | - | $10^{-3}$ |
| - | - | D10 | 0.998921 | 0.509894 | 0.88296 | 19.7399 | -0.461172 | 56.6657 | $10^{-5}$ |
| 4 | 6 | D6 | 0.781973 | 0.276306 | - | - | - | - | $10^0$ |
| - | - | D6² | 0.981025 | 0.36903 | - | - | - | - | $10^{-2}$ |
| - | - | D8 | 0.97989 | 0.370955 | -0.17615 | 5.02723 | - | - | $10^{-2}$ |
| - | - | D6D8 | 0.996654 | 0.37538 | 0.724048 | 8.93017 | - | - | $10^{-2}$ |
| - | - | D10 | 0.998524 | 0.375524 | 0.841921 | 8.89957 | -0.566179 | 39.3804 | $10^{-5}$ |
| 4 | 7 | D6 | 0.716039 | 0.202982 | - | - | - | - | $10^0$ |
| - | - | D6² | 0.966841 | 0.264586 | - | - | - | - | $10^{-2}$ |
| - | - | D8 | 0.973301 | 0.26889 | -0.453938 | 3.69954 | - | - | $10^{-2}$ |
| - | - | D6D8 | 0.997358 | 0.277802 | 0.744921 | 3.74595 | - | - | $10^{-2}$ |
| - | - | D10 | 0.998017 | 0.278291 | 0.796493 | 3.74167 | -0.392955 | 29.0139 | $10^{-4}$ |
| 4 | 8 | D6 | 0.640663 | 0.155459 | - | - | - | - | $10^1$ |
| - | - | D6² | 0.944611 | 0.184996 | - | - | - | - | $10^{-1}$ |
| - | - | D8 | 0.965726 | 0.187392 | -0.708307 | 2.84308 | - | - | $10^{-1}$ |
| - | - | D6D8 | 0.997508 | 0.198147 | 0.761652 | 1.86648 | - | - | $10^{-1}$ |
| - | - | D10 | 0.9974 | 0.197947 | 0.74912 | 1.86679 | 0.146217 | 22.3425 | $10^{-4}$ |
| 4 | 9 | D6 | 0.556532 | 0.122942 | - | - | - | - | $10^1$ |
| - | - | D6² | 0.924373 | 0.126767 | - | - | - | - | $10^{-1}$ |
| - | - | D8 | 0.957261 | 0.125719 | -0.912706 | 2.26109 | - | - | $10^{-1}$ |
| - | - | D6D8 | 0.996908 | 0.134364 | 0.768091 | 1.05707 | - | - | $10^{-1}$ |
| - | - | D10 | 0.996738 | 0.13326 | 0.702948 | 1.05732 | 1.09674 | 17.8252 | $10^{-3}$ |
| 4 | 10 | D6 | 0.46456 | 0.0997459 | - | - | - | - | $10^1$ |
| - | - | D6² | 0.911049 | 0.0864334 | - | - | - | - | $10^{-1}$ |
| - | - | D8 | 0.947969 | 0.084098 | -1.03874 | 1.84983 | - | - | $10^{-1}$ |
| - | - | D6D8 | 0.995492 | 0.0894328 | 0.765825 | 0.655434 | - | - | $10^{-1}$ |
| - | - | D10 | 0.996018 | 0.0879801 | 0.660509 | 0.654981 | 2.45085 | 14.649 | $10^{-3}$ |
| 5 | 1 | D6 | 0.973751 | 15.5735 | - | - | - | - | $10^{-6}$ |
| - | - | D6² | 0.977546 | 15.767 | - | - | - | - | $10^{-6}$ |
| - | - | D8 | 0.998565 | 15.7679 | 0.884563 | 443.287 | - | - | $10^{-8}$ |
| - | - | D6D8 | 0.998816 | 15.7625 | 0.90997 | 453.477 | - | - | $10^{-8}$ |
| - | - | D10 | 0.999705 | 15.762 | 1.00184 | 453.486 | 0.961592 | 5424.16 | $10^{-8}$ |
| 5 | 2 | D6 | 0.96382 | 3.89269 | - | - | - | - | $10^{-4}$ |
| - | - | D6² | 0.979031 | 4.0301 | - | - | - | - | $10^{-4}$ |
| - | - | D8 | 0.998482 | 4.0308 | 0.823035 | 110.768 | - | - | $10^{-7}$ |
| - | - | D6D8 | 0.999501 | 4.0254 | 0.925387 | 121.646 | - | - | $10^{-7}$ |
| - | - | D10 | 1.00025 | 4.02503 | 1.00314 | 121.655 | 0.760778 | 1355.25 | $10^{-7}$ |
| 5 | 3 | D6 | 0.946589 | 1.7296 | - | - | - | - | $10^{-3}$ |
| - | - | D6² | 0.980829 | 1.85773 | - | - | - | - | $10^{-4}$ |
| - | - | D8 | 0.997221 | 1.85805 | 0.704869 | 49.1945 | - | - | $10^{-6}$ |
| - | - | D6D8 | 0.999515 | 1.85292 | 0.932768 | 61.4015 | - | - | $10^{-6}$ |
| - | - | D10 | 0.999978 | 1.8527 | 0.981674 | 61.4084 | 0.42552 | 601.812 | $10^{-7}$ |
| 5 | 4 | D6 | 0.922452 | 0.97255 | - | - | - | - | $10^{-2}$ |
| - | - | D6² | 0.983284 | 1.09788 | - | - | - | - | $10^{-4}$ |
| - | - | D8 | 0.995507 | 1.09768 | 0.547838 | 27.6484 | - | - | $10^{-5}$ |
| - | - | D6D8 | 0.999605 | 1.09313 | 0.948863 | 42.214 | - | - | $10^{-5}$ |
| - | - | D10 | 0.99971 | 1.09308 | 0.9603 | 42.2161 | 0.0842902 | 338.19 | $10^{-7}$ |
| 5 | 5 | D6 | 0.891531 | 0.622187 | - | - | - | - | $10^{-2}$ |
| - | - | D6² | 0.98626 | 0.744806 | - | - | - | - | $10^{-3}$ |
| - | - | D8 | 0.993505 | 0.744163 | 0.358444 | 17.6807 | - | - | $10^{-4}$ |
| - | - | D6D8 | 0.999927 | 0.74078 | 0.977124 | 36.1198 | - | - | $10^{-4}$ |
| - | - | D10 | 0.999659 | 0.740866 | 0.944658 | 36.1119 | -0.212014 | 216.257 | $10^{-7}$ |
| 5 | 6 | D6 | 0.853728 | 0.43191 | - | - | - | - | $10^{-1}$ |
| - | - | D6² | 0.988748 | 0.548177 | - | - | - | - | $10^{-3}$ |
| - | - | D8 | 0.991025 | 0.547612 | 0.142171 | 12.2717 | - | - | $10^{-3}$ |
| - | - | D6D8 | 0.999441 | 0.546618 | 0.936311 | 33.0775 | - | - | $10^{-3}$ |
| - | - | D10 | 0.999565 | 0.546457 | 0.925456 | 33.079 | -0.462527 | 150.115 | $10^{-6}$ |
| 5 | 7 | D6 | 0.809113 | 0.317224 | - | - | - | - | $10^0$ |
| - | - | D6² | 0.988894 | 0.42023 | - | - | - | - | $10^{-3}$ |
| - | - | D8 | 0.988098 | 0.420805 | -0.0911835 | 9.01592 | - | - | $10^{-3}$ |
| - | - | D6D8 | 0.998488 | 0.422669 | 0.815106 | 20.2035 | - | - | $10^{-3}$ |
| - | - | D10 | 0.999452 | 0.422572 | 0.903282 | 20.174 | -0.608631 | 110.33 | $10^{-6}$ |
| 5 | 8 | D6 | 0.757763 | 0.242835 | - | - | - | - | $10^0$ |
| - | - | D6² | 0.984161 | 0.324698 | - | - | - | - | $10^{-2}$ |
| - | - | D8 | 0.984653 | 0.327061 | -0.330407 | 6.90866 | - | - | $10^{-3}$ |
| - | - | D6D8 | 0.998631 | 0.331679 | 0.812342 | 9.62964 | - | - | $10^{-3}$ |
| - | - | D10 | 0.999216 | 0.331896 | 0.876372 | 9.61919 | -0.572289 | 84.6071 | $10^{-5}$ |
| 5 | 9 | D6 | 0.699942 | 0.191881 | - | - | - | - | $10^0$ |
| - | - | D6² | 0.974035 | 0.247576 | - | - | - | - | $10^{-2}$ |
| - | - | D8 | 0.980815 | 0.250563 | -0.562432 | 5.46985 | - | - | $10^{-2}$ |
| - | - | D6D8 | 0.998842 | 0.257434 | 0.825855 | 5.02087 | - | - | $10^{-2}$ |
| - | - | D10 | 0.999024 | 0.257616 | 0.850135 | 5.01933 | -0.316331 | 67.0724 | $10^{-5}$ |
| 5 | 10 | D6 | 0.63592 | 0.155481 | - | - | - | - | $10^1$ |
| - | - | D6² | 0.961377 | 0.184817 | - | - | - | - | $10^{-1}$ |
| - | - | D8 | 0.976548 | 0.186309 | -0.773073 | 4.44659 | - | - | $10^{-2}$ |
| - | - | D6D8 | 0.998865 | 0.193824 | 0.834855 | 2.91539 | - | - | $10^{-2}$ |
| - | - | D10 | 0.998805 | 0.193707 | 0.823539 | 2.91565 | 0.206568 | 54.6306 | $10^{-5}$ |
| 6 | 1 | D6 | 0.982203 | 22.4263 | - | - | - | - | $10^{-6}$ |
| - | - | D6² | 0.984844 | 22.6536 | - | - | - | - | $10^{-6}$ |
| - | - | D8 | 1.00049 | 22.6543 | 0.947624 | 919.245 | - | - | $10^{-7}$ |
| - | - | D6D8 | 1.00062 | 22.6503 | 0.966506 | 933.852 | - | - | $10^{-7}$ |
| - | - | D10 | 1.00155 | 22.6498 | 1.10589 | 933.866 | 2.11512 | 16197.4 | $10^{-7}$ |

Table 8: As described in Tab. 6.

| $M_\Phi$ | $Y_\Phi$ | dim | $c_6$ | $\delta c_6$ | $c_8$ | $\delta c_8$ | $c_{10}$ | $\delta c_{10}$ | $\chi^2_{\min}$ |
|---|---|---|---|---|---|---|---|---|---|
| 6 | 2 | D6 | 0.974148 | 5.60586 | - | - | - | - | $10^{-5}$ |
| - | - | D6² | 0.984688 | 5.75152 | - | - | - | - | $10^{-5}$ |
| - | - | D8 | 0.998934 | 5.75208 | 0.865913 | 229.731 | - | - | $10^{-7}$ |
| - | - | D6D8 | 0.999426 | 5.74824 | 0.937829 | 244.981 | - | - | $10^{-7}$ |
| - | - | D10 | 0.999703 | 5.7481 | 0.979229 | 244.985 | 0.599885 | 4047.63 | $10^{-7}$ |
| 6 | 3 | D6 | 0.961997 | 2.49098 | - | - | - | - | $10^{-4}$ |
| - | - | D6² | 0.985733 | 2.62242 | - | - | - | - | $10^{-4}$ |
| - | - | D8 | 0.998539 | 2.62279 | 0.785238 | 102.047 | - | - | $10^{-7}$ |
| - | - | D6D8 | 0.999657 | 2.61905 | 0.947009 | 118.529 | - | - | $10^{-7}$ |
| - | - | D10 | 0.999929 | 2.61892 | 0.988007 | 118.534 | 0.548514 | 1797.77 | $10^{-8}$ |
| 6 | 4 | D6 | 0.944843 | 1.4008 | - | - | - | - | $10^{-3}$ |
| - | - | D6² | 0.987058 | 1.52791 | - | - | - | - | $10^{-4}$ |
| - | - | D8 | 0.997768 | 1.52802 | 0.670093 | 57.3627 | - | - | $10^{-6}$ |
| - | - | D6D8 | 0.999766 | 1.5245 | 0.955444 | 75.8753 | - | - | $10^{-6}$ |
| - | - | D10 | 0.99993 | 1.52442 | 0.98067 | 75.8792 | 0.300588 | 1010.44 | $10^{-7}$ |
| 6 | 5 | D6 | 0.922785 | 0.896236 | - | - | - | - | $10^{-2}$ |
| - | - | D6² | 0.988726 | 1.02133 | - | - | - | - | $10^{-4}$ |
| - | - | D8 | 0.996812 | 1.02112 | 0.528235 | 36.6856 | - | - | $10^{-5}$ |
| - | - | D6D8 | 0.999972 | 1.01802 | 0.97194 | 58.439 | - | - | $10^{-5}$ |
| - | - | D10 | 1.00003 | 1.018 | 0.980591 | 58.4407 | 0.0891306 | 646.156 | $10^{-7}$ |
| 6 | 6 | D6 | 0.895717 | 0.62218 | - | - | - | - | $10^{-2}$ |
| - | - | D6² | 0.990433 | 0.744833 | - | - | - | - | $10^{-4}$ |
| - | - | D8 | 0.995475 | 0.74438 | 0.359762 | 25.4593 | - | - | $10^{-4}$ |
| - | - | D6D8 | 1. | 0.742051 | 0.985516 | 52.0144 | - | - | $10^{-4}$ |
| - | - | D10 | 0.999858 | 0.742094 | 0.960871 | 52.0083 | -0.231166 | 448.412 | $10^{-7}$ |
| 6 | 7 | D6 | 0.863775 | 0.456966 | - | - | - | - | $10^{-1}$ |
| - | - | D6² | 0.991938 | 0.574728 | - | - | - | - | $10^{-4}$ |
| - | - | D8 | 0.993968 | 0.574295 | 0.173361 | 18.6962 | - | - | $10^{-4}$ |
| - | - | D6D8 | 0.999859 | 0.573295 | 0.971486 | 49.0246 | - | - | $10^{-4}$ |
| - | - | D10 | 0.999807 | 0.573248 | 0.949791 | 49.0231 | -0.450538 | 329.325 | $10^{-7}$ |
| 6 | 8 | D6 | 0.826949 | 0.349769 | - | - | - | - | $10^{-1}$ |
| - | - | D6² | 0.992431 | 0.45827 | - | - | - | - | $10^{-3}$ |
| - | - | D8 | 0.992223 | 0.458373 | -0.0262067 | 14.3129 | - | - | $10^{-3}$ |
| - | - | D6D8 | 0.999277 | 0.459095 | 0.881243 | 36.3069 | - | - | $10^{-3}$ |
| - | - | D10 | 0.999715 | 0.458989 | 0.934032 | 36.2908 | -0.603806 | 252.184 | $10^{-6}$ |
| 6 | 9 | D6 | 0.785327 | 0.27631 | - | - | - | - | $10^{0}$ |
| - | - | D6² | 0.990882 | 0.370067 | - | - | - | - | $10^{-3}$ |
| - | - | D8 | 0.990278 | 0.371204 | -0.232293 | 11.314 | - | - | $10^{-3}$ |
| - | - | D6D8 | 0.999255 | 0.373643 | 0.859793 | 20.0073 | - | - | $10^{-3}$ |
| - | - | D10 | 0.999664 | 0.373705 | 0.92004 | 19.9924 | -0.648599 | 199.452 | $10^{-6}$ |
| 6 | 10 | D6 | 0.738976 | 0.2238 | - | - | - | - | $10^{0}$ |
| - | - | D6² | 0.98643 | 0.297869 | - | - | - | - | $10^{-2}$ |
| - | - | D8 | 0.98807 | 0.299946 | -0.437665 | 9.17532 | - | - | $10^{-3}$ |
| - | - | D6D8 | 0.999332 | 0.304039 | 0.866607 | 11.0054 | - | - | $10^{-3}$ |
| - | - | D10 | 0.999543 | 0.304166 | 0.902511 | 11.0012 | -0.524536 | 161.888 | $10^{-6}$ |
| 7 | 1 | D6 | 0.986938 | 30.525 | - | - | - | - | $10^{-7}$ |
| - | - | D6² | 0.988875 | 30.7921 | - | - | - | - | $10^{-7}$ |
| - | - | D8 | 1.00054 | 30.7926 | 0.96156 | 1703.07 | - | - | $10^{-7}$ |
| - | - | D6D8 | 1.00061 | 30.7896 | 0.97554 | 1722.87 | - | - | $10^{-7}$ |
| - | - | D10 | 1.00122 | 30.7893 | 1.09923 | 1722.88 | 2.56513 | 40845.3 | $10^{-7}$ |
| 7 | 2 | D6 | 0.98076 | 7.63053 | - | - | - | - | $10^{-5}$ |
| - | - | D6² | 0.988493 | 7.78603 | - | - | - | - | $10^{-6}$ |
| - | - | D8 | 0.999242 | 7.78648 | 0.888092 | 425.655 | - | - | $10^{-7}$ |
| - | - | D6D8 | 0.999505 | 7.78363 | 0.940939 | 446.086 | - | - | $10^{-7}$ |
| - | - | D10 | 0.999506 | 7.78363 | 0.941212 | 446.086 | 0.00547358 | 10208. | $10^{-7}$ |
| 7 | 3 | D6 | 0.971765 | 3.39083 | - | - | - | - | $10^{-4}$ |
| - | - | D6² | 0.98919 | 3.52639 | - | - | - | - | $10^{-5}$ |
| - | - | D8 | 0.999345 | 3.52674 | 0.843669 | 189.101 | - | - | $10^{-7}$ |
| - | - | D6D8 | 0.999962 | 3.52389 | 0.965659 | 210.715 | - | - | $10^{-7}$ |
| - | - | D10 | 1.00022 | 3.52376 | 1.01836 | 210.721 | 0.997397 | 4534.62 | $10^{-7}$ |
| 7 | 4 | D6 | 0.958937 | 1.90695 | - | - | - | - | $10^{-3}$ |
| - | - | D6² | 0.989928 | 2.03613 | - | - | - | - | $10^{-5}$ |
| - | - | D8 | 0.998905 | 2.03633 | 0.754454 | 106.312 | - | - | $10^{-6}$ |
| - | - | D6D8 | 1. | 2.03359 | 0.96967 | 129.779 | - | - | $10^{-6}$ |
| - | - | D10 | 1.00021 | 2.03349 | 1.01176 | 129.784 | 0.732708 | 2549.09 | $10^{-7}$ |
| 7 | 5 | D6 | 0.942347 | 1.22015 | - | - | - | - | $10^{-3}$ |
| - | - | D6² | 0.990765 | 1.34668 | - | - | - | - | $10^{-4}$ |
| - | - | D8 | 0.998204 | 1.34669 | 0.639728 | 67.9983 | - | - | $10^{-6}$ |
| - | - | D6D8 | 0.999915 | 1.34415 | 0.970715 | 94.2612 | - | - | $10^{-6}$ |
| - | - | D10 | 1.00002 | 1.34411 | 0.992323 | 94.2648 | 0.336998 | 1630.26 | $10^{-8}$ |
| 7 | 6 | D6 | 0.922167 | 0.847092 | - | - | - | - | $10^{-2}$ |
| - | - | D6² | 0.991865 | 0.971958 | - | - | - | - | $10^{-4}$ |
| - | - | D8 | 0.997547 | 0.971774 | 0.510155 | 47.1918 | - | - | $10^{-5}$ |
| - | - | D6D8 | 1.00003 | 0.969544 | 0.983115 | 77.6058 | - | - | $10^{-5}$ |
| - | - | D10 | 1.00005 | 0.969535 | 0.988123 | 77.6068 | 0.0687991 | 1131.34 | $10^{-7}$ |
| 7 | 7 | D6 | 0.898246 | 0.622176 | - | - | - | - | $10^{-2}$ |
| - | - | D6² | 0.992948 | 0.744845 | - | - | - | - | $10^{-4}$ |
| - | - | D8 | 0.996654 | 0.744511 | 0.360327 | 34.6522 | - | - | $10^{-5}$ |
| - | - | D6D8 | 1.00001 | 0.742807 | 0.991074 | 70.7971 | - | - | $10^{-5}$ |
| - | - | D10 | 0.999938 | 0.742829 | 0.973345 | 70.7927 | -0.226011 | 830.718 | $10^{-7}$ |
| 7 | 8 | D6 | 0.870661 | 0.476223 | - | - | - | - | $10^{-1}$ |
| - | - | D6² | 0.993897 | 0.594926 | - | - | - | - | $10^{-4}$ |
| - | - | D8 | 0.995647 | 0.594592 | 0.196872 | 26.52 | - | - | $10^{-4}$ |
| - | - | D6D8 | 0.999904 | 0.593708 | 0.981212 | 67.6682 | - | - | $10^{-4}$ |
| - | - | D10 | 0.99984 | 0.593695 | 0.959344 | 67.6651 | -0.444864 | 635.812 | $10^{-7}$ |

Table 9: As described in Tab. 6.

| $M_\Phi$ | $Y_\Phi$ | dim | $c_6$ | $\delta c_6$ | $c_8$ | $\delta c_8$ | $c_{10}$ | $\delta c_{10}$ | $\chi^2_{\min}$ |
|---|---|---|---|---|---|---|---|---|---|
| 7 | 9 | D6 | 0.839479 | 0.376183 | - | - | - | - | $10^{-1}$ |
| - | - | D6$^2$ | 0.994443 | 0.487958 | - | - | - | - | $10^{-4}$ |
| - | - | D8 | 0.994606 | 0.487896 | 0.0242301 | 20.9511 | - | - | $10^{-4}$ |
| - | - | D6D8 | 0.9997 | 0.488088 | 0.929098 | 56.1551 | - | - | $10^{-4}$ |
| - | - | D10 | 0.999885 | 0.488016 | 0.95529 | 56.1499 | -0.595993 | 502.404 | $10^{-7}$ |
| 7 | 10 | D6 | 0.804648 | 0.304652 | - | - | - | - | $10^0$ |
| - | - | D6$^2$ | 0.993948 | 0.405634 | - | - | - | - | $10^{-3}$ |
| - | - | D8 | 0.993373 | 0.406158 | -0.155669 | 16.9743 | - | - | $10^{-3}$ |
| - | - | D6D8 | 0.999574 | 0.40747 | 0.895448 | 35.6145 | - | - | $10^{-3}$ |
| - | - | D10 | 0.999831 | 0.407469 | 0.944221 | 35.5994 | -0.661499 | 407.199 | $10^{-7}$ |
| 8 | 1 | D6 | 0.989651 | 39.8697 | - | - | - | - | $10^{-7}$ |
| - | - | D6$^2$ | 0.99113 | 40.1825 | - | - | - | - | $10^{-7}$ |
| - | - | D8 | 0.999735 | 40.1829 | 0.926302 | 2905.42 | - | - | $10^{-7}$ |
| - | - | D6D8 | 0.999774 | 40.1807 | 0.936565 | 2931.2 | - | - | $10^{-7}$ |
| - | - | D10 | 1.0002 | 40.1805 | 1.04937 | 2931.21 | 3.06346 | 91013.2 | $10^{-7}$ |
| 8 | 2 | D6 | 0.985085 | 9.9667 | - | - | - | - | $10^{-6}$ |
| - | - | D6$^2$ | 0.991 | 10.1336 | - | - | - | - | $10^{-6}$ |
| - | - | D8 | 0.999355 | 10.1339 | 0.900847 | 726.205 | - | - | $10^{-7}$ |
| - | - | D6D8 | 0.999509 | 10.1317 | 0.941463 | 752.621 | - | - | $10^{-7}$ |
| - | - | D10 | 0.999564 | 10.1317 | 0.956 | 752.623 | 0.384795 | 22747.6 | $10^{-7}$ |
| 8 | 3 | D6 | 0.97811 | 4.42911 | - | - | - | - | $10^{-5}$ |
| - | - | D6$^2$ | 0.991433 | 4.56954 | - | - | - | - | $10^{-5}$ |
| - | - | D8 | 0.999465 | 4.56984 | 0.86929 | 322.652 | - | - | $10^{-8}$ |
| - | - | D6D8 | 0.999823 | 4.56764 | 0.962466 | 350.213 | - | - | $10^{-8}$ |
| - | - | D10 | 0.999962 | 4.56757 | 0.99942 | 350.217 | 0.936345 | 10106. | $10^{-8}$ |
| 8 | 4 | D6 | 0.968243 | 2.49097 | - | - | - | - | $10^{-4}$ |
| - | - | D6$^2$ | 0.991944 | 2.62267 | - | - | - | - | $10^{-5}$ |
| - | - | D8 | 0.999337 | 2.62288 | 0.805992 | 181.413 | - | - | $10^{-7}$ |
| - | - | D6D8 | 0.99998 | 2.62072 | 0.97182 | 210.724 | - | - | $10^{-7}$ |
| - | - | D10 | 1.0001 | 2.62066 | 1.00395 | 210.728 | 0.764383 | 5681.68 | $10^{-7}$ |
| 8 | 5 | D6 | 0.955466 | 1.59391 | - | - | - | - | $10^{-3}$ |
| - | - | D6$^2$ | 0.992516 | 1.72194 | - | - | - | - | $10^{-5}$ |
| - | - | D8 | 0.999046 | 1.72204 | 0.721846 | 116.046 | - | - | $10^{-6}$ |
| - | - | D6D8 | 1.00007 | 1.71997 | 0.981244 | 147.899 | - | - | $10^{-6}$ |
| - | - | D10 | 1.00023 | 1.71989 | 1.0256 | 147.906 | 0.970839 | 3634.1 | $10^{-7}$ |
| 8 | 6 | D6 | 0.939784 | 1.10663 | - | - | - | - | $10^{-2}$ |
| - | - | D6$^2$ | 0.993134 | 1.23278 | - | - | - | - | $10^{-4}$ |
| - | - | D8 | 0.998564 | 1.23275 | 0.615123 | 80.5432 | - | - | $10^{-6}$ |
| - | - | D6D8 | 1.00002 | 1.23084 | 0.982674 | 116.017 | - | - | $10^{-6}$ |
| - | - | D10 | 1.00008 | 1.23082 | 0.99788 | 116.019 | 0.300343 | 2522.09 | $10^{-7}$ |
| 8 | 7 | D6 | 0.921244 | 0.812837 | - | - | - | - | $10^{-2}$ |
| - | - | D6$^2$ | 0.993822 | 0.937502 | - | - | - | - | $10^{-4}$ |
| - | - | D8 | 0.997995 | 0.937343 | 0.493329 | 59.1427 | - | - | $10^{-5}$ |
| - | - | D6D8 | 0.999975 | 0.93568 | 0.985574 | 99.6911 | - | - | $10^{-5}$ |
| - | - | D10 | 0.999973 | 0.935681 | 0.984953 | 99.691 | -0.0109677 | 1851.87 | $10^{-8}$ |
| 8 | 8 | D6 | 0.899895 | 0.622173 | - | - | - | - | $10^{-2}$ |
| - | - | D6$^2$ | 0.994585 | 0.744852 | - | - | - | - | $10^{-4}$ |
| - | - | D8 | 0.997419 | 0.744595 | 0.360089 | 45.2594 | - | - | $10^{-5}$ |
| - | - | D6D8 | 1.00001 | 0.743297 | 0.993138 | 92.4681 | - | - | $10^{-5}$ |
| - | - | D10 | 0.999963 | 0.743309 | 0.980253 | 92.4648 | -0.214335 | 1417.14 | $10^{-7}$ |
| 8 | 9 | D6 | 0.875709 | 0.491475 | - | - | - | - | $10^{-1}$ |
| - | - | D6$^2$ | 0.995279 | 0.610833 | - | - | - | - | $10^{-4}$ |
| - | - | D8 | 0.996787 | 0.610569 | 0.216587 | 35.7478 | - | - | $10^{-4}$ |
| - | - | D6D8 | 1.00002 | 0.609804 | 0.993614 | 89.0923 | - | - | $10^{-4}$ |
| - | - | D10 | 0.999969 | 0.609801 | 0.974937 | 89.0889 | -0.42456 | 1119.39 | $10^{-8}$ |
| 8 | 10 | D6 | 0.84869 | 0.398008 | - | - | - | - | $10^{-1}$ |
| - | - | D6$^2$ | 0.995711 | 0.511905 | - | - | - | - | $10^{-4}$ |
| - | - | D8 | 0.996071 | 0.511789 | 0.0641191 | 28.9509 | - | - | $10^{-4}$ |
| - | - | D6D8 | 0.999906 | 0.51173 | 0.961513 | 78.9495 | - | - | $10^{-4}$ |
| - | - | D10 | 0.999976 | 0.511688 | 0.970967 | 78.9489 | -0.58538 | 906.709 | $10^{-7}$ |
| 9 | 1 | D6 | 0.992295 | 50.4604 | - | - | - | - | $10^{-7}$ |
| - | - | D6$^2$ | 0.993465 | 50.8251 | - | - | - | - | $10^{-7}$ |
| - | - | D8 | 1.0008 | 50.8254 | 0.999832 | 4653.98 | - | - | $10^{-7}$ |
| - | - | D6D8 | 1.00083 | 50.8236 | 1.00863 | 4686.59 | - | - | $10^{-7}$ |
| - | - | D10 | 1.00137 | 50.8233 | 1.18949 | 4686.61 | 6.22714 | 184513. | $10^{-7}$ |
| 9 | 2 | D6 | 0.988606 | 12.6144 | - | - | - | - | $10^{-6}$ |
| - | - | D6$^2$ | 0.993286 | 12.7943 | - | - | - | - | $10^{-6}$ |
| - | - | D8 | 1.00037 | 12.7946 | 0.965824 | 1163.3 | - | - | $10^{-7}$ |
| - | - | D6D8 | 1.00048 | 12.7927 | 1.00082 | 1196.54 | - | - | $10^{-7}$ |
| - | - | D10 | 1.00113 | 12.7924 | 1.21951 | 1196.56 | 7.37889 | 46119. | $10^{-7}$ |
| 9 | 3 | D6 | 0.982512 | 5.60584 | - | - | - | - | $10^{-5}$ |
| - | - | D6$^2$ | 0.993028 | 5.75187 | - | - | - | - | $10^{-6}$ |
| - | - | D8 | 0.999467 | 5.75213 | 0.880738 | 516.887 | - | - | $10^{-7}$ |
| - | - | D6D8 | 0.999684 | 5.75039 | 0.953195 | 551.208 | - | - | $10^{-7}$ |
| - | - | D10 | 0.999602 | 5.75043 | 0.925523 | 551.205 | -0.902349 | 20490.7 | $10^{-7}$ |
| 9 | 4 | D6 | 0.974543 | 3.15288 | - | - | - | - | $10^{-4}$ |
| - | - | D6$^2$ | 0.993243 | 3.28751 | - | - | - | - | $10^{-5}$ |
| - | - | D8 | 0.999272 | 3.28771 | 0.828786 | 290.646 | - | - | $10^{-7}$ |
| - | - | D6D8 | 0.999663 | 3.286 | 0.957701 | 326.642 | - | - | $10^{-7}$ |
| - | - | D10 | 0.999643 | 3.28601 | 0.951186 | 326.641 | -0.202258 | 11521.1 | $10^{-7}$ |
| 9 | 5 | D6 | 0.964463 | 2.01751 | - | - | - | - | $10^{-3}$ |
| - | - | D6$^2$ | 0.993704 | 2.14727 | - | - | - | - | $10^{-5}$ |
| - | - | D8 | 0.999251 | 2.1474 | 0.769484 | 185.935 | - | - | $10^{-7}$ |
| - | - | D6D8 | 0.999876 | 2.14572 | 0.973086 | 224.313 | - | - | $10^{-7}$ |
| - | - | D10 | 0.999922 | 2.1457 | 0.988786 | 224.315 | 0.456807 | 7369.8 | $10^{-7}$ |

Table 10: As described in Tab. 6.

| $M_\Phi$ | $Y_\Phi$ | dim | $c_6$ | $\delta c_6$ | $c_8$ | $\delta c_8$ | $c_{10}$ | $\delta c_{10}$ | $\chi^2_{\min}$ |
|---|---|---|---|---|---|---|---|---|---|
| 9 | 6 | D6 | 0.952123 | 1.40079 | - | - | - | - | $10^{-3}$ |
| - | - | D6$^2$ | 0.99426 | 1.52813 | - | - | - | - | $10^{-5}$ |
| - | - | D8 | 0.999187 | 1.52818 | 0.694104 | 129.061 | - | - | $10^{-6}$ |
| - | - | D6D8 | 1.00011 | 1.52657 | 0.989926 | 170.732 | - | - | $10^{-6}$ |
| - | - | D10 | 1.00019 | 1.52653 | 1.01798 | 170.736 | 0.752688 | 5115.09 | $10^{-8}$ |
| 9 | 7 | D6 | 0.93734 | 1.02894 | - | - | - | - | $10^{-2}$ |
| - | - | D6$^2$ | 0.994685 | 1.15479 | - | - | - | - | $10^{-5}$ |
| - | - | D8 | 0.998782 | 1.15474 | 0.592083 | 94.7737 | - | - | $10^{-6}$ |
| - | - | D6D8 | 1.00003 | 1.15327 | 0.988138 | 140.922 | - | - | $10^{-6}$ |
| - | - | D10 | 1.00007 | 1.15325 | 1.00163 | 140.924 | 0.329028 | 3755.93 | $10^{-8}$ |
| 9 | 8 | D6 | 0.920304 | 0.78761 | - | - | - | - | $10^{-2}$ |
| - | - | D6$^2$ | 0.995174 | 0.912112 | - | - | - | - | $10^{-4}$ |
| - | - | D8 | 0.998365 | 0.911974 | 0.48056 | 72.5267 | - | - | $10^{-5}$ |
| - | - | D6D8 | 0.999986 | 0.910686 | 0.989533 | 124.691 | - | - | $10^{-5}$ |
| - | - | D10 | 0.999983 | 0.910687 | 0.988484 | 124.691 | -0.0231992 | 2874.14 | $10^{-8}$ |
| 9 | 9 | D6 | 0.901044 | 0.622172 | - | - | - | - | $10^{-2}$ |
| - | - | D6$^2$ | 0.995732 | 0.74486 | - | - | - | - | $10^{-4}$ |
| - | - | D8 | 0.997972 | 0.744657 | 0.360517 | 57.2809 | - | - | $10^{-5}$ |
| - | - | D6D8 | 1.00003 | 0.743632 | 0.996549 | 117.031 | - | - | $10^{-5}$ |
| - | - | D10 | 1. | 0.743639 | 0.987559 | 117.029 | -0.189147 | 2269.97 | $10^{-7}$ |
| 9 | 10 | D6 | 0.879495 | 0.503851 | - | - | - | - | $10^{-1}$ |
| - | - | D6$^2$ | 0.996208 | 0.623678 | - | - | - | - | $10^{-4}$ |
| - | - | D8 | 0.997503 | 0.623466 | 0.231618 | 46.3826 | - | - | $10^{-5}$ |
| - | - | D6D8 | 1.00002 | 0.622811 | 0.996653 | 113.328 | - | - | $10^{-5}$ |
| - | - | D10 | 0.999982 | 0.622811 | 0.98118 | 113.325 | -0.407428 | 1838.18 | $10^{-8}$ |
| 10 | 1 | D6 | 0.993549 | 62.297 | - | - | - | - | $10^{-7}$ |
| - | - | D6$^2$ | 0.994497 | 62.7196 | - | - | - | - | $10^{-8}$ |
| - | - | D8 | 1.0008 | 62.7199 | 1.06003 | 7093.48 | - | - | $10^{-8}$ |
| - | - | D6D8 | 1.00082 | 62.7183 | 1.06755 | 7133.69 | - | - | $10^{-8}$ |
| - | - | D10 | 1.0012 | 62.7181 | 1.22298 | 7133.7 | 6.61533 | 347198. | $10^{-8}$ |
| 10 | 2 | D6 | 0.990706 | 15.5735 | - | - | - | - | $10^{-6}$ |
| - | - | D6$^2$ | 0.994495 | 15.7679 | - | - | - | - | $10^{-7}$ |
| - | - | D8 | 1.00035 | 15.7681 | 0.985429 | 1773.13 | - | - | $10^{-8}$ |
| - | - | D6D8 | 1.00042 | 15.7666 | 1.01394 | 1813.96 | - | - | $10^{-8}$ |
| - | - | D10 | 1.00079 | 15.7664 | 1.1659 | 1813.97 | 6.36299 | 86785.3 | $10^{-8}$ |
| 10 | 3 | D6 | 0.985607 | 6.92102 | - | - | - | - | $10^{-5}$ |
| - | - | D6$^2$ | 0.994117 | 7.07335 | - | - | - | - | $10^{-6}$ |
| - | - | D8 | 0.999337 | 7.07357 | 0.880505 | 787.887 | - | - | $10^{-8}$ |
| - | - | D6D8 | 0.999477 | 7.07218 | 0.938416 | 829.77 | - | - | $10^{-8}$ |
| - | - | D10 | 0.999427 | 7.0722 | 0.917419 | 829.768 | -0.855342 | 38561. | $10^{-8}$ |
| 10 | 4 | D6 | 0.979654 | 3.89265 | - | - | - | - | $10^{-4}$ |
| - | - | D6$^2$ | 0.994815 | 4.03078 | - | - | - | - | $10^{-6}$ |
| - | - | D8 | 1.0001 | 4.03097 | 0.893974 | 443.056 | - | - | $10^{-7}$ |
| - | - | D6D8 | 1.00037 | 4.0295 | 1.00552 | 486.609 | - | - | $10^{-7}$ |
| - | - | D10 | 1.00062 | 4.02938 | 1.10805 | 486.621 | 4.01508 | 21682.9 | $10^{-8}$ |
| 10 | 5 | D6 | 0.971129 | 2.49097 | - | - | - | - | $10^{-4}$ |
| - | - | D6$^2$ | 0.994808 | 2.62277 | - | - | - | - | $10^{-5}$ |
| - | - | D8 | 0.999612 | 2.62291 | 0.818614 | 283.455 | - | - | $10^{-7}$ |
| - | - | D6D8 | 1.00003 | 2.62151 | 0.98696 | 329.256 | - | - | $10^{-7}$ |
| - | - | D10 | 1.0001 | 2.62147 | 1.01644 | 329.26 | 1.09605 | 13871.1 | $10^{-8}$ |
| 10 | 6 | D6 | 0.961023 | 1.72957 | - | - | - | - | $10^{-3}$ |
| - | - | D6$^2$ | 0.995134 | 1.85822 | - | - | - | - | $10^{-5}$ |
| - | - | D8 | 0.999501 | 1.8583 | 0.751808 | 196.764 | - | - | $10^{-7}$ |
| - | - | D6D8 | 1.00011 | 1.85694 | 0.994318 | 245.626 | - | - | $10^{-7}$ |
| - | - | D10 | 1.0002 | 1.8569 | 1.03128 | 245.631 | 1.28776 | 9628.09 | $10^{-7}$ |
| 10 | 7 | D6 | 0.948945 | 1.27048 | - | - | - | - | $10^{-3}$ |
| - | - | D6$^2$ | 0.995368 | 1.39733 | - | - | - | - | $10^{-5}$ |
| - | - | D8 | 0.999135 | 1.39735 | 0.659486 | 144.499 | - | - | $10^{-6}$ |
| - | - | D6D8 | 0.999944 | 1.39608 | 0.980288 | 197.418 | - | - | $10^{-6}$ |
| - | - | D10 | 0.999964 | 1.39607 | 0.98884 | 197.419 | 0.275652 | 7070.13 | $10^{-8}$ |
| 10 | 8 | D6 | 0.935095 | 0.972527 | - | - | - | - | $10^{-2}$ |
| - | - | D6$^2$ | 0.995734 | 1.09813 | - | - | - | - | $10^{-5}$ |
| - | - | D8 | 0.99891 | 1.09808 | 0.570479 | 110.583 | - | - | $10^{-6}$ |
| - | - | D6D8 | 0.999973 | 1.09691 | 0.987468 | 168.874 | - | - | $10^{-6}$ |
| - | - | D10 | 0.999988 | 1.0969 | 0.993603 | 168.875 | 0.181087 | 5410.37 | $10^{-8}$ |
| 10 | 9 | D6 | 0.919405 | 0.768265 | - | - | - | - | $10^{-2}$ |
| - | - | D6$^2$ | 0.996141 | 0.892628 | - | - | - | - | $10^{-5}$ |
| - | - | D8 | 0.998658 | 0.892509 | 0.470451 | 87.3373 | - | - | $10^{-6}$ |
| - | - | D6D8 | 1.00001 | 0.891482 | 0.993735 | 152.599 | - | - | $10^{-6}$ |
| - | - | D10 | 1.00001 | 0.891482 | 0.993769 | 152.599 | 0.000898094 | 4272.91 | $10^{-7}$ |
| 10 | 10 | D6 | 0.901816 | 0.62217 | - | - | - | - | $10^{-2}$ |
| - | - | D6$^2$ | 0.996482 | 0.744854 | - | - | - | - | $10^{-5}$ |
| - | - | D8 | 0.998283 | 0.744691 | 0.357783 | 70.7167 | - | - | $10^{-5}$ |
| - | - | D6D8 | 0.999924 | 0.743874 | 0.985524 | 144.47 | - | - | $10^{-5}$ |
| - | - | D10 | 0.999892 | 0.743882 | 0.970747 | 144.466 | -0.383684 | 3459.76 | $10^{-7}$ |

Table 11: Table of fit $\chi^2$ fit values for the $X$ model as described in Sec. 4 and Tab. 2. Table is continued below.

| $M_X$ | $\beta$ | dim | $c_6$ | $\delta c_6$ | $c_8$ | $\delta c_8$ | $c_{10}$ | $\delta c_{10}$ | $\chi^2_{\min}$ |
|---|---|---|---|---|---|---|---|---|---|
| 3 | 3 | D6 | 1.12788 | 0.353318 | - | - | - | - | $10^{-1}$ |
| - | - | D6$^2$ | 1.13203 | 0.356668 | - | - | - | - | $10^{-1}$ |
| - | - | D8 | 0.980971 | 0.356442 | 1.45476 | 2.38857 | - | - | $10^{-2}$ |
| - | - | D6D8 | 0.979411 | 0.357435 | 1.49589 | 2.44542 | - | - | $10^{-2}$ |
| - | - | D10 | 1.00498 | 0.357087 | 0.850995 | 2.44694 | 1.8517 | 8.11232 | $10^{-4}$ |
| 3 | 6 | D6 | 1.10815 | 0.0870852 | - | - | - | - | $10^{0}$ |
| - | - | D6$^2$ | 1.12538 | 0.0905217 | - | - | - | - | $10^{0}$ |
| - | - | D8 | 0.987427 | 0.0903332 | 1.29106 | 0.572319 | - | - | $10^{-2}$ |
| - | - | D6D8 | 0.981081 | 0.0913225 | 1.45242 | 0.631675 | - | - | $10^{-2}$ |
| - | - | D10 | 1.00166 | 0.0910396 | 0.943074 | 0.633042 | 1.34751 | 1.92171 | $10^{-4}$ |
| 3 | 9 | D6 | 1.07091 | 0.0378047 | - | - | - | - | $10^{1}$ |
| - | - | D6$^2$ | 1.11237 | 0.0413785 | - | - | - | - | $10^{1}$ |
| - | - | D8 | 1.00085 | 0.0412695 | 1.00748 | 0.238168 | - | - | $10^{-2}$ |
| - | - | D6D8 | 0.986527 | 0.0422007 | 1.35066 | 0.303184 | - | - | $10^{-2}$ |
| - | - | D10 | 0.997849 | 0.0420497 | 1.07877 | 0.304116 | 0.622311 | 0.787609 | $10^{-3}$ |
| 3 | 12 | D6 | 1.00923 | 0.0206546 | - | - | - | - | $10^{0}$ |
| - | - | D6$^2$ | 1.08999 | 0.0243725 | - | - | - | - | $10^{1}$ |
| - | - | D8 | 1.02154 | 0.0243805 | 0.616649 | 0.12527 | - | - | $10^{0}$ |
| - | - | D6D8 | 0.997353 | 0.0251004 | 1.14917 | 0.202589 | - | - | $10^{0}$ |
| - | - | D10 | 0.992205 | 0.0251541 | 1.26864 | 0.201985 | -0.222281 | 0.411097 | $10^{-1}$ |
| 3 | 15 | D6 | 0.926368 | 0.0129133 | - | - | - | - | $10^{2}$ |
| - | - | D6$^2$ | 1.06471 | 0.0167086 | - | - | - | - | $10^{1}$ |
| - | - | D8 | 1.04683 | 0.0167598 | 0.180373 | 0.0785712 | - | - | $10^{1}$ |
| - | - | D6D8 | 1.01233 | 0.0171405 | 0.866336 | 0.179652 | - | - | $10^{1}$ |
| - | - | D10 | 0.984882 | 0.0171841 | 1.49054 | 0.174775 | -0.983607 | 0.26331 | $10^{0}$ |
| 3 | 18 | D6 | 0.842816 | 0.00888742 | - | - | - | - | $10^{3}$ |
| - | - | D6$^2$ | 1.05498 | 0.0126621 | - | - | - | - | $10^{2}$ |
| - | - | D8 | 1.06967 | 0.0125303 | -0.20599 | 0.0574116 | - | - | $10^{2}$ |
| - | - | D6D8 | 1.01002 | 0.0127117 | 1.18099 | 0.169176 | - | - | $10^{2}$ |
| - | - | D10 | 0.977171 | 0.0122382 | 1.69327 | 0.166058 | -1.47187 | 0.203686 | $10^{1}$ |
| 3 | 21 | D6 | 0.771161 | 0.00654558 | - | - | - | - | $10^{3}$ |
| - | - | D6$^2$ | 1.06933 | 0.0101629 | - | - | - | - | $10^{3}$ |
| - | - | D8 | 1.08616 | 0.00970972 | -0.508954 | 0.0464134 | - | - | $10^{2}$ |
| - | - | D6D8 | 0.993852 | 0.00885815 | 2.06728 | 0.119261 | - | - | $10^{2}$ |
| - | - | D10 | 0.96982 | 0.00879942 | 1.84537 | 0.122417 | -1.65923 | 0.177694 | $10^{1}$ |
| 3 | 24 | D6 | 0.710911 | 0.00505784 | - | - | - | - | $10^{4}$ |
| - | - | D6$^2$ | 1.10199 | 0.0082592 | - | - | - | - | $10^{3}$ |
| - | - | D8 | 1.09643 | 0.0076162 | -0.744376 | 0.0398252 | - | - | $10^{3}$ |
| - | - | D6D8 | 0.972171 | 0.00620991 | 2.37603 | 0.0755521 | - | - | $10^{3}$ |
| - | - | D10 | 0.962486 | 0.00644627 | 1.94384 | 0.0768416 | -1.62891 | 0.165102 | $10^{1}$ |
| 3 | 27 | D6 | 0.658785 | 0.00405035 | - | - | - | - | $10^{4}$ |
| - | - | D6$^2$ | 1.13855 | 0.00654542 | - | - | - | - | $10^{3}$ |
| - | - | D8 | 1.10128 | 0.00600319 | -0.934216 | 0.0354559 | - | - | $10^{3}$ |
| - | - | D6D8 | 0.957567 | 0.00465887 | 2.35815 | 0.0502491 | - | - | $10^{3}$ |
| - | - | D10 | 0.954571 | 0.00486197 | 1.98925 | 0.050523 | -1.51136 | 0.158513 | $10^{2}$ |
| 3 | 30 | D6 | 0.612118 | 0.00333483 | - | - | - | - | $10^{4}$ |
| - | - | D6$^2$ | 1.16495 | 0.00506135 | - | - | - | - | $10^{3}$ |
| - | - | D8 | 1.10119 | 0.00476017 | -1.09737 | 0.0323498 | - | - | $10^{3}$ |
| - | - | D6D8 | 0.946954 | 0.00364767 | 2.27092 | 0.0356695 | - | - | $10^{3}$ |
| - | - | D10 | 0.945524 | 0.00379038 | 1.97817 | 0.0357588 | -1.49372 | 0.154855 | $10^{2}$ |
| 4 | 3 | D6 | 1.06432 | 0.629608 | - | - | - | - | $10^{-2}$ |
| - | - | D6$^2$ | 1.06679 | 0.632873 | - | - | - | - | $10^{-2}$ |
| - | - | D8 | 0.997294 | 0.632768 | 1.19634 | 7.6072 | - | - | $10^{-4}$ |
| - | - | D6D8 | 0.996878 | 0.633219 | 1.21572 | 7.7096 | - | - | $10^{-4}$ |
| - | - | D10 | 1.00371 | 0.633126 | 0.908069 | 7.71031 | 1.5892 | 46.0403 | $10^{-5}$ |
| 4 | 6 | D6 | 1.05434 | 0.15631 | - | - | - | - | $10^{-1}$ |
| - | - | D6$^2$ | 1.06434 | 0.159615 | - | - | - | - | $10^{-1}$ |
| - | - | D8 | 0.997126 | 0.159518 | 1.14089 | 1.86438 | - | - | $10^{-3}$ |
| - | - | D6D8 | 0.995423 | 0.159976 | 1.21881 | 1.96918 | - | - | $10^{-3}$ |
| - | - | D10 | 1.00099 | 0.159899 | 0.969922 | 1.96979 | 1.23001 | 11.2285 | $10^{-5}$ |
| 4 | 9 | D6 | 1.03827 | 0.0686636 | - | - | - | - | $10^{0}$ |
| - | - | D6$^2$ | 1.06135 | 0.072045 | - | - | - | - | $10^{0}$ |
| - | - | D8 | 0.999435 | 0.0719638 | 1.02919 | 0.802011 | - | - | $10^{-4}$ |
| - | - | D6D8 | 0.995538 | 0.0724228 | 1.20253 | 0.911658 | - | - | $10^{-4}$ |
| - | - | D10 | 1.00004 | 0.0723604 | 1.00338 | 0.912217 | 0.911276 | 4.7932 | $10^{-5}$ |
| 4 | 12 | D6 | 1.01404 | 0.0379978 | - | - | - | - | $10^{-1}$ |
| - | - | D6$^2$ | 1.05654 | 0.0414815 | - | - | - | - | $10^{0}$ |
| - | - | D8 | 1.00303 | 0.0414276 | 0.870016 | 0.431865 | - | - | $10^{-2}$ |
| - | - | D6D8 | 0.996004 | 0.0418767 | 1.17039 | 0.549872 | - | - | $10^{-2}$ |
| - | - | D10 | 0.998972 | 0.0418359 | 1.04102 | 0.55031 | 0.528222 | 2.55695 | $10^{-4}$ |
| 4 | 15 | D6 | 0.979859 | 0.023833 | - | - | - | - | $10^{0}$ |
| - | - | D6$^2$ | 1.04913 | 0.0274298 | - | - | - | - | $10^{1}$ |
| - | - | D8 | 1.00774 | 0.0274152 | 0.666482 | 0.26301 | - | - | $10^{-1}$ |
| - | - | D6D8 | 0.996731 | 0.0278271 | 1.11312 | 0.395232 | - | - | $10^{-1}$ |
| - | - | D10 | 0.997378 | 0.0278188 | 1.08536 | 0.395355 | 0.0971082 | 1.544 | $10^{-3}$ |

Table 12: As described in Tab. 11.

| $M_X$ | $\beta$ | dim | $c_6$ | $\delta c_6$ | $c_8$ | $\delta c_8$ | $c_{10}$ | $\delta c_{10}$ | $\chi^2_{\min}$ |
|---|---|---|---|---|---|---|---|---|---|
| 4 | 18 | D6 | 0.934817 | 0.016197 | - | - | - | - | $10^1$ |
| - | - | D6$^2$ | 1.03896 | 0.0198842 | - | - | - | - | $10^1$ |
| - | - | D8 | 1.01327 | 0.0199112 | 0.426714 | 0.174712 | - | - | $10^0$ |
| - | - | D6D8 | 0.997761 | 0.0202352 | 1.01528 | 0.330998 | - | - | $10^0$ |
| - | - | D10 | 0.994955 | 0.0202625 | 1.1344 | 0.330258 | -0.347859 | 1.02276 | $10^{-2}$ |
| 4 | 21 | D6 | 0.881011 | 0.0116809 | - | - | - | - | $10^2$ |
| - | - | D6$^2$ | 1.0278 | 0.0153866 | - | - | - | - | $10^1$ |
| - | - | D8 | 1.01883 | 0.0154218 | 0.167914 | 0.125567 | - | - | $10^1$ |
| - | - | D6D8 | 0.99813 | 0.0156184 | 0.898149 | 0.316049 | - | - | $10^1$ |
| - | - | D10 | 0.991452 | 0.0156319 | 1.18145 | 0.313794 | -0.771298 | 0.742141 | $10^{-1}$ |
| 4 | 24 | D6 | 0.824176 | 0.0088429 | - | - | - | - | $10^3$ |
| - | - | D6$^2$ | 1.01956 | 0.0124572 | - | - | - | - | $10^1$ |
| - | - | D8 | 1.02311 | 0.0124232 | -0.0876536 | 0.0974413 | - | - | $10^1$ |
| - | - | D6D8 | 0.991305 | 0.0125123 | 1.05743 | 0.29586 | - | - | $10^1$ |
| - | - | D10 | 0.986642 | 0.0124081 | 1.21565 | 0.295296 | -1.16688 | 0.591249 | $10^{-1}$ |
| 4 | 27 | D6 | 0.769791 | 0.00696577 | - | - | - | - | $10^3$ |
| - | - | D6$^2$ | 1.01711 | 0.0103706 | - | - | - | - | $10^2$ |
| - | - | D8 | 1.02478 | 0.0102001 | -0.326512 | 0.080718 | - | - | $10^1$ |
| - | - | D6D8 | 0.977765 | 0.0100053 | 1.52128 | 0.221022 | - | - | $10^1$ |
| - | - | D10 | 0.980274 | 0.00996182 | 1.22494 | 0.220395 | -1.59332 | 0.510147 | $10^{-1}$ |
| 4 | 30 | D6 | 0.719886 | 0.00566208 | - | - | - | - | $10^3$ |
| - | - | D6$^2$ | 1.01993 | 0.00874775 | - | - | - | - | $10^2$ |
| - | - | D8 | 1.02303 | 0.00844388 | -0.550255 | 0.0701249 | - | - | $10^2$ |
| - | - | D6D8 | 0.965893 | 0.0079549 | 1.7356 | 0.147362 | - | - | $10^2$ |
| - | - | D10 | 0.972029 | 0.00807064 | 1.1986 | 0.145733 | -2.19289 | 0.465412 | $10^0$ |
| 5 | 3 | D6 | 1.04054 | 0.984653 | - | - | - | - | $10^{-3}$ |
| - | - | D6$^2$ | 1.04214 | 0.987883 | - | - | - | - | $10^{-3}$ |
| - | - | D8 | 1.00196 | 0.987822 | 1.08267 | 18.6226 | - | - | $10^{-5}$ |
| - | - | D6D8 | 1.00181 | 0.988081 | 1.09388 | 18.7829 | - | - | $10^{-5}$ |
| - | - | D10 | 1.00562 | 0.98803 | 0.825618 | 18.7835 | 2.17667 | 176.233 | $10^{-5}$ |
| 5 | 6 | D6 | 1.03348 | 0.245126 | - | - | - | - | $10^{-2}$ |
| - | - | D6$^2$ | 1.03991 | 0.248375 | - | - | - | - | $10^{-2}$ |
| - | - | D8 | 0.999576 | 0.248316 | 1.07771 | 4.60098 | - | - | $10^{-4}$ |
| - | - | D6D8 | 0.998928 | 0.248584 | 1.12409 | 4.76346 | - | - | $10^{-4}$ |
| - | - | D10 | 1.00132 | 0.248551 | 0.956224 | 4.76386 | 1.32458 | 43.4194 | $10^{-5}$ |
| 5 | 9 | D6 | 1.02388 | 0.108177 | - | - | - | - | $10^{-1}$ |
| - | - | D6$^2$ | 1.03854 | 0.111473 | - | - | - | - | $10^{-1}$ |
| - | - | D8 | 0.999888 | 0.111418 | 1.01896 | 2.00528 | - | - | $10^{-5}$ |
| - | - | D6D8 | 0.998403 | 0.111689 | 1.12332 | 2.17231 | - | - | $10^{-5}$ |
| - | - | D10 | 1.00038 | 0.111662 | 0.985568 | 2.17267 | 1.03623 | 18.8382 | $10^{-5}$ |
| 5 | 12 | D6 | 1.00998 | 0.0602473 | - | - | - | - | $10^{-2}$ |
| - | - | D6$^2$ | 1.03656 | 0.0636086 | - | - | - | - | $10^0$ |
| - | - | D8 | 1.0007 | 0.063563 | 0.930206 | 1.09799 | - | - | $10^{-4}$ |
| - | - | D6D8 | 0.998018 | 0.0638348 | 1.11413 | 1.2723 | - | - | $10^{-4}$ |
| - | - | D10 | 0.999613 | 0.0638126 | 1.00372 | 1.27262 | 0.774643 | 10.2533 | $10^{-5}$ |
| 5 | 15 | D6 | 0.990933 | 0.0380704 | - | - | - | - | $10^{-1}$ |
| - | - | D6$^2$ | 1.03351 | 0.0415125 | - | - | - | - | $10^0$ |
| - | - | D8 | 1.00163 | 0.04148 | 0.813574 | 0.679671 | - | - | $10^{-2}$ |
| - | - | D6D8 | 0.997397 | 0.041748 | 1.09566 | 0.864998 | - | - | $10^{-2}$ |
| - | - | D10 | 0.998511 | 0.0417325 | 1.01919 | 0.865256 | 0.488463 | 6.30339 | $10^{-4}$ |
| 5 | 18 | D6 | 0.965893 | 0.0260397 | - | - | - | - | $10^0$ |
| - | - | D6$^2$ | 1.02891 | 0.0295697 | - | - | - | - | $10^0$ |
| - | - | D8 | 1.00242 | 0.029555 | 0.669774 | 0.454578 | - | - | $10^{-1}$ |
| - | - | D6D8 | 0.996306 | 0.0298097 | 1.06217 | 0.656493 | - | - | $10^{-1}$ |
| - | - | D10 | 0.996742 | 0.0298038 | 1.03251 | 0.656615 | 0.168076 | 4.1875 | $10^{-4}$ |
| 5 | 21 | D6 | 0.934215 | 0.0188138 | - | - | - | - | $10^1$ |
| - | - | D6$^2$ | 1.02235 | 0.0224234 | - | - | - | - | $10^0$ |
| - | - | D8 | 1.00275 | 0.0224291 | 0.500178 | 0.321628 | - | - | $10^{-1}$ |
| - | - | D6D8 | 0.994565 | 0.0226532 | 1.00184 | 0.548546 | - | - | $10^{-1}$ |
| - | - | D10 | 0.993985 | 0.0226603 | 1.04119 | 0.548338 | -0.193425 | 2.94854 | $10^{-3}$ |
| 5 | 24 | D6 | 0.895982 | 0.0141668 | - | - | - | - | $10^2$ |
| - | - | D6$^2$ | 1.0137 | 0.0178232 | - | - | - | - | $10^0$ |
| - | - | D8 | 1.00221 | 0.017844 | 0.307827 | 0.238731 | - | - | $10^0$ |
| - | - | D6D8 | 0.992008 | 0.0180137 | 0.896514 | 0.501956 | - | - | $10^0$ |
| - | - | D10 | 0.989903 | 0.0180328 | 1.0405 | 0.500969 | -0.619413 | 2.188 | $10^{-3}$ |
| 5 | 27 | D6 | 0.852562 | 0.0110361 | - | - | - | - | $10^2$ |
| - | - | D6$^2$ | 1.00348 | 0.014677 | - | - | - | - | $10^0$ |
| - | - | D8 | 1.00023 | 0.0146916 | 0.0976998 | 0.185623 | - | - | $10^0$ |
| - | - | D6D8 | 0.987778 | 0.0147991 | 0.762986 | 0.490231 | - | - | $10^0$ |
| - | - | D10 | 0.984127 | 0.0148064 | 1.02352 | 0.488319 | -1.16773 | 1.71363 | $10^{-2}$ |
| 5 | 30 | D6 | 0.806484 | 0.00885404 | - | - | - | - | $10^3$ |
| - | - | D6$^2$ | 0.992757 | 0.0123987 | - | - | - | - | $10^1$ |
| - | - | D8 | 0.996115 | 0.0123682 | -0.125553 | 0.151125 | - | - | $10^0$ |
| - | - | D6D8 | 0.977947 | 0.0124495 | 0.840751 | 0.460585 | - | - | $10^0$ |
| - | - | D10 | 0.976295 | 0.0123872 | 0.980555 | 0.460251 | -1.95454 | 1.41855 | $10^{-2}$ |
| 6 | 3 | D6 | 1.029 | 1.41854 | - | - | - | - | $10^{-3}$ |
| - | - | D6$^2$ | 1.03012 | 1.42176 | - | - | - | - | $10^{-3}$ |
| - | - | D8 | 1.00443 | 1.42172 | 0.99779 | 38.6663 | - | - | $10^{-5}$ |
| - | - | D6D8 | 1.00436 | 1.42188 | 1.0049 | 38.8975 | - | - | $10^{-5}$ |
| - | - | D10 | 1.00766 | 1.42184 | 0.670863 | 38.8982 | 3.91389 | 527.092 | $10^{-5}$ |

Table 13: As described in Tab. 11.

| $M_X$ | $\beta$ | dim | $c_6$ | $\delta c_6$ | $c_8$ | $\delta c_8$ | $c_{10}$ | $\delta c_{10}$ | $\chi^2_{\min}$ |
|---|---|---|---|---|---|---|---|---|---|
| 6 | 6 | D6 | 1.0231 | 0.353626 | - | - | - | - | $10^{-2}$ |
| - | - | D6$^2$ | 1.02757 | 0.356846 | - | - | - | - | $10^{-2}$ |
| - | - | D8 | 1.0007 | 0.356806 | 1.03749 | 9.59054 | - | - | $10^{-5}$ |
| - | - | D6D8 | 1.00041 | 0.356982 | 1.06816 | 9.82338 | - | - | $10^{-5}$ |
| - | - | D10 | 1.00192 | 0.356962 | 0.915134 | 9.82374 | 1.75886 | 130.497 | $10^{-5}$ |
| 6 | 9 | D6 | 1.01645 | 0.156419 | - | - | - | - | $10^{-2}$ |
| - | - | D6$^2$ | 1.02658 | 0.159669 | - | - | - | - | $10^{-2}$ |
| - | - | D8 | 1.00025 | 0.159631 | 1.00743 | 4.20693 | - | - | $10^{-5}$ |
| - | - | D6D8 | 0.999566 | 0.159811 | 1.07715 | 4.44411 | - | - | $10^{-5}$ |
| - | - | D10 | 1.0007 | 0.159795 | 0.963307 | 4.44439 | 1.26644 | 57.0715 | $10^{-5}$ |
| 6 | 12 | D6 | 1.00712 | 0.0873974 | - | - | - | - | $10^{-2}$ |
| - | - | D6$^2$ | 1.02536 | 0.0906925 | - | - | - | - | $10^{-1}$ |
| - | - | D8 | 1.00017 | 0.0906579 | 0.952493 | 2.32371 | - | - | $10^{-5}$ |
| - | - | D6D8 | 0.998933 | 0.0908387 | 1.07632 | 2.56762 | - | - | $10^{-5}$ |
| - | - | D10 | 0.999839 | 0.0908262 | 0.985441 | 2.56787 | 0.964559 | 31.3958 | $10^{-5}$ |
| 6 | 15 | D6 | 0.994553 | 0.0554535 | - | - | - | - | $10^{-2}$ |
| - | - | D6$^2$ | 1.02351 | 0.0588048 | - | - | - | - | $10^{-1}$ |
| - | - | D8 | 1.00002 | 0.0587756 | 0.87614 | 1.45338 | - | - | $10^{-3}$ |
| - | - | D6D8 | 0.998072 | 0.0589564 | 1.06774 | 1.70696 | - | - | $10^{-3}$ |
| - | - | D10 | 0.998801 | 0.0589462 | 0.995035 | 1.70717 | 0.724855 | 19.5399 | $10^{-5}$ |
| 6 | 18 | D6 | 0.978148 | 0.0381078 | - | - | - | - | $10^{-1}$ |
| - | - | D6$^2$ | 1.02061 | 0.0415239 | - | - | - | - | $10^{0}$ |
| - | - | D8 | 0.999437 | 0.0415022 | 0.779703 | 0.982276 | - | - | $10^{-3}$ |
| - | - | D6D8 | 0.996613 | 0.0416802 | 1.05023 | 1.24935 | - | - | $10^{-3}$ |
| - | - | D10 | 0.99712 | 0.0416732 | 0.999976 | 1.24952 | 0.46265 | 13.134 | $10^{-5}$ |
| 6 | 21 | D6 | 0.957354 | 0.0276604 | - | - | - | - | $10^{0}$ |
| - | - | D6$^2$ | 1.01624 | 0.0311447 | - | - | - | - | $10^{0}$ |
| - | - | D8 | 0.998114 | 0.0311326 | 0.661893 | 0.700304 | - | - | $10^{-2}$ |
| - | - | D6D8 | 0.994301 | 0.0313028 | 1.01666 | 0.986123 | - | - | $10^{-2}$ |
| - | - | D10 | 0.994494 | 0.0313001 | 0.997552 | 0.986199 | 0.159608 | 9.3128 | $10^{-4}$ |
| 6 | 24 | D6 | 0.931688 | 0.0208979 | - | - | - | - | $10^{1}$ |
| - | - | D6$^2$ | 1.00994 | 0.0244451 | - | - | - | - | $10^{0}$ |
| - | - | D8 | 0.995695 | 0.0244438 | 0.521384 | 0.51987 | - | - | $10^{-1}$ |
| - | - | D6D8 | 0.990869 | 0.0245973 | 0.954089 | 0.831728 | - | - | $10^{-1}$ |
| - | - | D10 | 0.990557 | 0.0246013 | 0.984813 | 0.83158 | -0.229103 | 6.88186 | $10^{-4}$ |
| 6 | 27 | D6 | 0.90095 | 0.0162879 | - | - | - | - | $10^{1}$ |
| - | - | D6$^2$ | 1.00132 | 0.0198795 | - | - | - | - | $10^{0}$ |
| - | - | D8 | 0.991787 | 0.0198876 | 0.356441 | 0.399246 | - | - | $10^{-1}$ |
| - | - | D6D8 | 0.986123 | 0.0200113 | 0.838812 | 0.746804 | - | - | $10^{-1}$ |
| - | - | D10 | 0.984951 | 0.020025 | 0.95527 | 0.746114 | -0.770891 | 5.27224 | $10^{-4}$ |
| 6 | 30 | D6 | 0.865429 | 0.0130248 | - | - | - | - | $10^{2}$ |
| - | - | D6$^2$ | 0.990113 | 0.0166263 | - | - | - | - | $10^{0}$ |
| - | - | D8 | 0.985926 | 0.0166361 | 0.165411 | 0.316445 | - | - | $10^{-1}$ |
| - | - | D6D8 | 0.979825 | 0.016718 | 0.640609 | 0.709609 | - | - | $10^{-1}$ |
| - | - | D10 | 0.977269 | 0.0167405 | 0.901631 | 0.707769 | -1.59143 | 4.18391 | $10^{-3}$ |
| 7 | 3 | D6 | 1.02288 | 1.9313 | - | - | - | - | $10^{-4}$ |
| - | - | D6$^2$ | 1.0237 | 1.93451 | - | - | - | - | $10^{-4}$ |
| - | - | D8 | 1.00665 | 1.93448 | 0.902131 | 71.6873 | - | - | $10^{-5}$ |
| - | - | D6D8 | 1.00662 | 1.93459 | 0.906743 | 72.0024 | - | - | $10^{-5}$ |
| - | - | D10 | 1.01023 | 1.93455 | 0.407461 | 72.0035 | 7.97618 | 1330.36 | $10^{-5}$ |
| 7 | 6 | D6 | 1.01721 | 0.481831 | - | - | - | - | $10^{-3}$ |
| - | - | D6$^2$ | 1.02049 | 0.485035 | - | - | - | - | $10^{-3}$ |
| - | - | D8 | 1.00147 | 0.485006 | 1.00242 | 17.8204 | - | - | $10^{-5}$ |
| - | - | D6D8 | 1.00131 | 0.48513 | 1.02397 | 18.1364 | - | - | $10^{-5}$ |
| - | - | D10 | 1.00259 | 0.485113 | 0.847507 | 18.1368 | 2.7797 | 330.276 | $10^{-5}$ |
| 7 | 9 | D6 | 1.01213 | 0.21341 | - | - | - | - | $10^{-3}$ |
| - | - | D6$^2$ | 1.01955 | 0.216634 | - | - | - | - | $10^{-2}$ |
| - | - | D8 | 1.00053 | 0.216606 | 0.995301 | 7.8458 | - | - | $10^{-5}$ |
| - | - | D6D8 | 1.00017 | 0.216734 | 1.04504 | 8.16588 | - | - | $10^{-5}$ |
| - | - | D10 | 1.00098 | 0.216723 | 0.933747 | 8.16615 | 1.71181 | 145.1 | $10^{-5}$ |
| 7 | 12 | D6 | 1.0053 | 0.119464 | - | - | - | - | $10^{-3}$ |
| - | - | D6$^2$ | 1.0186 | 0.122719 | - | - | - | - | $10^{-2}$ |
| - | - | D8 | 1.00008 | 0.122693 | 0.960405 | 4.35565 | - | - | $10^{-5}$ |
| - | - | D6D8 | 0.999435 | 0.122822 | 1.04923 | 4.68214 | - | - | $10^{-5}$ |
| - | - | D10 | 1.00006 | 0.122813 | 0.964044 | 4.68236 | 1.2666 | 80.3171 | $10^{-5}$ |
| 7 | 15 | D6 | 0.996155 | 0.0759814 | - | - | - | - | $10^{-3}$ |
| - | - | D6$^2$ | 1.01717 | 0.0792776 | - | - | - | - | $10^{-1}$ |
| - | - | D8 | 0.999478 | 0.0792539 | 0.907748 | 2.74141 | - | - | $10^{-4}$ |
| - | - | D6D8 | 0.998457 | 0.0793832 | 1.04596 | 3.0768 | - | - | $10^{-4}$ |
| - | - | D10 | 0.998956 | 0.0793763 | 0.97794 | 3.07699 | 0.96701 | 50.3672 | $10^{-5}$ |
| 7 | 18 | D6 | 0.984248 | 0.0523652 | - | - | - | - | $10^{-1}$ |
| - | - | D6$^2$ | 1.01489 | 0.0557085 | - | - | - | - | $10^{-1}$ |
| - | - | D8 | 0.998356 | 0.0556884 | 0.838465 | 1.86601 | - | - | $10^{-4}$ |
| - | - | D6D8 | 0.996878 | 0.055817 | 1.0349 | 2.21329 | - | - | $10^{-4}$ |
| - | - | D10 | 0.997257 | 0.0558117 | 0.98335 | 2.21344 | 0.692647 | 34.1397 | $10^{-5}$ |
| 7 | 21 | D6 | 0.969136 | 0.0381316 | - | - | - | - | $10^{0}$ |
| - | - | D6$^2$ | 1.01137 | 0.0415263 | - | - | - | - | $10^{-1}$ |
| - | - | D8 | 0.996413 | 0.0415109 | 0.750673 | 1.33996 | - | - | $10^{-3}$ |
| - | - | D6D8 | 0.994416 | 0.0416366 | 1.01074 | 1.70288 | - | - | $10^{-3}$ |
| - | - | D10 | 0.99466 | 0.0416332 | 0.977706 | 1.70299 | 0.414387 | 24.4041 | $10^{-5}$ |

Table 14: As described in Tab. 11.

| $M_X$ | $\beta$ | dim | $c_6$ | $\delta c_6$ | $c_8$ | $\delta c_8$ | $c_{10}$ | $\delta c_{10}$ | $\chi^2_{\min}$ |
|---|---|---|---|---|---|---|---|---|---|
| 7 | 24 | D6 | 0.950354 | 0.0289032 | - | - | - | - | $10^0$ |
| - | - | D6$^2$ | 1.00617 | 0.0323494 | - | - | - | - | $10^{-1}$ |
| - | - | D8 | 0.99327 | 0.0323398 | 0.64258 | 1.0007 | - | - | $10^{-2}$ |
| - | - | D6D8 | 0.990728 | 0.0324588 | 0.965326 | 1.38406 | - | - | $10^{-2}$ |
| - | - | D10 | 0.990763 | 0.0324583 | 0.960662 | 1.38407 | 0.0539281 | 18.1427 | $10^{-5}$ |
| 7 | 27 | D6 | 0.927504 | 0.0225904 | - | - | - | - | $10^1$ |
| - | - | D6$^2$ | 0.998806 | 0.0260823 | - | - | - | - | $10^{-1}$ |
| - | - | D8 | 0.98855 | 0.0260791 | 0.510485 | 0.770686 | - | - | $10^{-2}$ |
| - | - | D6D8 | 0.985523 | 0.0261851 | 0.882197 | 1.18069 | - | - | $10^{-2}$ |
| - | - | D10 | 0.985194 | 0.0261896 | 0.926521 | 1.18049 | -0.467042 | 13.9165 | $10^{-5}$ |
| 7 | 30 | D6 | 0.900336 | 0.0180942 | - | - | - | - | $10^1$ |
| - | - | D6$^2$ | 0.988796 | 0.0216182 | - | - | - | - | $10^{-1}$ |
| - | - | D8 | 0.981838 | 0.0216204 | 0.349993 | 0.609175 | - | - | $10^{-1}$ |
| - | - | D6D8 | 0.978547 | 0.0217043 | 0.733969 | 1.05361 | - | - | $10^{-1}$ |
| - | - | D10 | 0.977557 | 0.0217169 | 0.868176 | 1.0529 | -1.27983 | 10.9691 | $10^{-4}$ |
| 8 | 3 | D6 | 1.01969 | 2.52294 | - | - | - | - | $10^{-5}$ |
| - | - | D6$^2$ | 1.02032 | 2.52614 | - | - | - | - | $10^{-5}$ |
| - | - | D8 | 1.0092 | 2.52613 | 0.768645 | 122.352 | - | - | $10^{-5}$ |
| - | - | D6D8 | 1.00919 | 2.5262 | 0.771495 | 122.764 | - | - | $10^{-5}$ |
| - | - | D10 | 1.01367 | 2.52614 | -0.0367678 | 122.766 | 16.8834 | 2965.98 | $10^{-5}$ |
| 8 | 6 | D6 | 1.01363 | 0.629749 | - | - | - | - | $10^{-3}$ |
| - | - | D6$^2$ | 1.01614 | 0.632943 | - | - | - | - | $10^{-3}$ |
| - | - | D8 | 1.00215 | 0.632922 | 0.963676 | 30.4573 | - | - | $10^{-5}$ |
| - | - | D6D8 | 1.00207 | 0.633013 | 0.979354 | 30.8692 | - | - | $10^{-5}$ |
| - | - | D10 | 1.0034 | 0.632995 | 0.740265 | 30.8698 | 4.94083 | 737.606 | $10^{-5}$ |
| 8 | 9 | D6 | 1.00944 | 0.279159 | - | - | - | - | $10^{-3}$ |
| - | - | D6$^2$ | 1.01511 | 0.282366 | - | - | - | - | $10^{-3}$ |
| - | - | D8 | 1.00081 | 0.282345 | 0.979773 | 13.4404 | - | - | $10^{-5}$ |
| - | - | D6D8 | 1.00061 | 0.28244 | 1.01676 | 13.8562 | - | - | $10^{-5}$ |
| - | - | D10 | 1.00134 | 0.28243 | 0.885373 | 13.8565 | 2.6664 | 324.973 | $10^{-5}$ |
| 8 | 12 | D6 | 1.00409 | 0.156453 | - | - | - | - | $10^{-3}$ |
| - | - | D6$^2$ | 1.01423 | 0.159683 | - | - | - | - | $10^{-2}$ |
| - | - | D8 | 1.00011 | 0.159663 | 0.96136 | 7.48544 | - | - | $10^{-5}$ |
| - | - | D6D8 | 0.99974 | 0.159759 | 1.028 | 7.90736 | - | - | $10^{-5}$ |
| - | - | D10 | 1.00024 | 0.159752 | 0.93779 | 7.90758 | 1.78433 | 180.588 | $10^{-5}$ |
| 8 | 15 | D6 | 0.997018 | 0.0996584 | - | - | - | - | $10^{-3}$ |
| - | - | D6$^2$ | 1.01298 | 0.102919 | - | - | - | - | $10^{-2}$ |
| - | - | D8 | 0.999289 | 0.1029 | 0.924236 | 4.73028 | - | - | $10^{-5}$ |
| - | - | D6D8 | 0.998705 | 0.102997 | 1.02838 | 5.16059 | - | - | $10^{-5}$ |
| - | - | D10 | 0.999094 | 0.102991 | 0.958947 | 5.16078 | 1.3277 | 113.802 | $10^{-5}$ |
| 8 | 18 | D6 | 0.987788 | 0.0688098 | - | - | - | - | $10^{-2}$ |
| - | - | D6$^2$ | 1.01098 | 0.0721055 | - | - | - | - | $10^{-1}$ |
| - | - | D8 | 0.997935 | 0.0720883 | 0.871844 | 3.23503 | - | - | $10^{-4}$ |
| - | - | D6D8 | 0.997091 | 0.072185 | 1.02034 | 3.67624 | - | - | $10^{-4}$ |
| - | - | D10 | 0.997399 | 0.0721807 | 0.96541 | 3.6764 | 1.00712 | 77.5742 | $10^{-5}$ |
| 8 | 21 | D6 | 0.976005 | 0.0502132 | - | - | - | - | $10^{-1}$ |
| - | - | D6$^2$ | 1.00784 | 0.0535471 | - | - | - | - | $10^{-1}$ |
| - | - | D8 | 0.995699 | 0.0535325 | 0.803608 | 2.3351 | - | - | $10^{-4}$ |
| - | - | D6D8 | 0.994557 | 0.0536278 | 1.00129 | 2.79011 | - | - | $10^{-4}$ |
| - | - | D10 | 0.99478 | 0.0536247 | 0.961571 | 2.79023 | 0.69189 | 55.7899 | $10^{-5}$ |
| 8 | 24 | D6 | 0.961253 | 0.0381497 | - | - | - | - | $10^0$ |
| - | - | D6$^2$ | 1.00314 | 0.0415225 | - | - | - | - | $10^{-1}$ |
| - | - | D8 | 0.992216 | 0.0415112 | 0.716856 | 1.75299 | - | - | $10^{-3}$ |
| - | - | D6D8 | 0.990758 | 0.041603 | 0.964379 | 2.22523 | - | - | $10^{-3}$ |
| - | - | D10 | 0.990864 | 0.0416015 | 0.945701 | 2.2253 | 0.306376 | 41.7204 | $10^{-5}$ |
| 8 | 27 | D6 | 0.943141 | 0.0298882 | - | - | - | - | $10^0$ |
| - | - | D6$^2$ | 0.996439 | 0.0332972 | - | - | - | - | $10^{-1}$ |
| - | - | D8 | 0.987127 | 0.0332897 | 0.606923 | 1.35623 | - | - | $10^{-3}$ |
| - | - | D6D8 | 0.985379 | 0.0333745 | 0.896676 | 1.84988 | - | - | $10^{-3}$ |
| - | - | D10 | 0.985302 | 0.0333756 | 0.910389 | 1.84983 | -0.209871 | 32.154 | $10^{-5}$ |
| 8 | 30 | D6 | 0.921307 | 0.0239913 | - | - | - | - | $10^1$ |
| - | - | D6$^2$ | 0.98722 | 0.0274291 | - | - | - | - | $10^{-1}$ |
| - | - | D8 | 0.980012 | 0.0274258 | 0.46873 | 1.07517 | - | - | $10^{-2}$ |
| - | - | D6D8 | 0.978078 | 0.0274981 | 0.77873 | 1.59523 | - | - | $10^{-2}$ |
| - | - | D10 | 0.977671 | 0.0275037 | 0.85077 | 1.59492 | -1.0202 | 25.4018 | $10^{-5}$ |

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
