# Peer review of "Top-down and bottom-up: Studying the SMEFT beyond leading order in 1/Lambda^2"

_SciPost Physics, doi:SciPost Phys. Core 7, 053 (2024)_

## Round 1 · Referee Report · Anonymous (Referee 1) · 2024-6-18

Report
This paper studies examples of four UV extensions of the SM, their imprint on the SMEFT, and asseses the possibility of the SMEFT expansion breakdown in the case of the Drell-Yan process. In addition, the author studies the reliability of fits that rely on a SMEFT expansion truncated at dimension-six squared, compared to fits that include dimension-eight and dimension-ten operators.
After matching the models to dimension-ten, the author studies how the inclusion of dimension-eight and dimension-ten operators in hypothetical effective field theory fits to the full ultraviolet models impacts the fits. This is done in both the top-down approach and in a limited approximation to the bottom up approach of the SMEFT to infer the impact of a fully general fit to the SMEFT. The importance of including higher order terms depends on the nature of the new physics interactions, whether they are weakly coupled models or have stronger interactions or a lighter mass. The author concludes by pointing out some future considerations that would further improve our understanding of the convergence of the SMEFT expansion.
The paper is written very clearly and all the relevant details of this analysis have been discussed at length. There are some minor typos in the paper some of which are listed below. Once these typos are fixed the paper is ready for publication in SciPost.
-
Near the end of page 8 the author states that 1/lambda^4 terms include dim6 amplitude interfered with dim8 as well as dim10 interfered with the SM. I assume this is a typo and that the author means 1/lambda^6?
-
Page 21 right after eq. 24, there is a typo: "indicted" -> indicated .
-
EWPD was first used on page 11 but the acronym wasn't explained. If the acronym refers to "Electroweak Precision Data" the author should write that explicitly when the acronym is first used.
-
Page 29 near the end, M_phi = 3 is missing the unit "TeV".
Recommendation
Ask for minor revision

---

## Round 1 · Referee Report · Anonymous (Referee 2) · 2024-6-22

Strengths
- Large amount of potentially useful calculations within very concrete models relevant for Drell-Yan.
Weaknesses
- Most, if not all, of the conclusions seem to be rather trivial or well-known results.
- There are seemingly important flaws in the fitting procedure.
Report
He finds that in the weakly-interacting regime, the 1/cutoff^2 term in the expansion suffices to parametrize the UV. He also finds that in more strongly-coupled cases, the 1/cutoff^4 term might be needed, though some times it worsens the perturbative result (which is only improved upon including higher-order terms).
Notwithstanding the large amount of work done by a single author, I think these findings are rather trivial and well-known. In particular, the seemingly surprising fact that adding more terms in the EFT expansion does not always make the result closer to the UV computation can be understood, for example, by expanding the function Cos(x) in Taylor series around x=0 and applying it to x=3. The first term in the expansion is closer to the actual value (-0.99...) than the next one. What holds is that the EFT expansion will accurately describe the UV result provided enough terms are included.
The author makes rather similar statements from a different perspective, namely by fitting the EFT to simulated data generated in the UV. The statement is that, if the fit is close to the UV value (by construction equal to 1), the EFT expansion is fine. I do not see how much this adds to the previous considerations. Moreover, I think the fitting procedure presents some flaw, because despite the uncertainties being at times very large (see delta=4.6 in Table 2), the central value does not fluctuate at all (it is always 1).
Recommendation
Reject

---

## Round 1 · Referee Report · Anonymous (Referee 3) · 2024-6-26

Strengths
-
The organization of the paper, and approach to the problem, is extremely clear: a reasonable set of different models were chosen to highlight different results.
-
The subject of the paper is timely; the LHC community is in the midst of intensive work on how to implement SMEFT interpretations, and understanding how/when to include effects of higher-order terms in the EFT expansion is of utmost importance.
Weaknesses
-
The scope is relatively limited: a small number of benchmark models are chosen to illustrate different effects, but only their effects on the Drell-Yan process are considered, and only the dilepton invariant mass spectrum is fit; it's difficult to extrapolate many of the findings to a more sophisticated analysis involving additional kinematic variables or other processes, which would provide other handles to constrain the "nuisance parameters".
-
Some of the graphics could be more clearly presented (the same curves could be on one panel, for instance, and the axis labels could be improved).
Report
The author presents a study of four simple UV models that lead to qualitatively different effects on the Drell-Yan (DY) process at the LHC and their matching onto the SMEFT including operators at dimension-8 and 10. The goal is to understand how well the Wilson coefficients, dictated by the UV theory, can be measured via a fit to the dilepton invariant mass spectrum with different truncations of the SMEFT expansion applied. While other studies of dimension-8 matching effects exist, this paper presents a novel, comparative study between different models, with an emphasis on the DY process and a fit to the full invariant mass spectrum that I find to be a valuable addition to the literature. While the scope is limited (see "Weaknesses", above) the results nicely demonstrate several potential pitfalls in performing SMEFT fits to LHC data if these fits are to be interpreted as constraints on new physics, and highlight some of the challenges of this global approach.
I have several questions and comments regarding the work, listed below, as well as some (minor) changes that should be addressed in a revised manuscript:
-
Regarding the results in Table 2: for the X model, with MX = 5 TeV and \beta = 3.0, it appears that when moving from D8 to D6D8, while the central value of c_8 gets significantly closer to its true value, the determination becomes much less precise (+/- 0.70 -> +/- 2.1). Is there a way to understand this behavior? This seems to be distinct from the accidentally good reproduction of the distribution that appears when including the D6^2 terms discussed elsewhere in the paper.
-
There has been a lot of prior work on Drell-Yan studies in the context of effective theories (albeit much of it limited to "universal" corrections in terms of the W & Y parameters) -- see for instance, arXiv:2008.12978 and arXiv:2103.10532. Given the relevance, I think a more extensive discussion of the relation between the present work (and SMEFT interpretations of Drell-Yan more generally) and these studies is warranted.
-
Related to the above, it would be useful to discuss in particular the potential benefits of doubly- or triply-differential Drell-Yan measurements in the context of higher-dimension operators. My suspicion is that a lot of the "nuisance parameter" behavior of the additional coefficients discussed in this work is due to the fact that, ultimately, this is fitting to a single distribution, and, even if the fit closes, there is a lot of unavoidable overlap in the shape of the deviations allowed in the dilepton invariant mass tail. I would expect the additional dimension in kinematic space to substantially affect this behavior, but the author should comment on the extent to which this is true.
-
There seems to be a typo in the paragraph below Eq. (15). I assume the author means that it is the $1/\Lambda^6$ temrs that include dimension-six amplitudes interfering with dimension-eight, etc.
Assuming these questions and the changes below are addressed, I am happy to recommend the paper for publication.
Requested changes
-
Figure 1 would be much clearer if the three panels showing the ratio R for the different models were combined into one figure---the axis ranges are not too different in each panel, and I think an easy comparison of them is important. They could even be combined with the first panel (the total cross section), e.g., with the same x-axis, though this is more of a personal preference.
-
The y-axis labels are somewhat confusing when not read in lots of context---the label "R" is used to mean a number of different ratios, and Figure 4 is particularly ambiguous. More descriptive titles would be very useful.
-
Include references to doubly- and triply-differential DY papers (and other relevant prior work), as discussed above.
Recommendation
Publish (meets expectations and criteria for this Journal)

---

## Round 2 · Author Response

The following citation was added after the paper was submitted to scipost: ''The other effective fermion compositeness'' Bellazzini, Brando and Riva, Francesco and Serra, Javi and Sgarlata, Francesco

Report 3: I thank the referee for the review of the article and suggestions for improvement. I have implemented all their requested changes and hope that have answered all questions satisfactorily.

Comments in report: 1) For table 2 the increase is a reflection of the fit wanting to find agreement with the high invariant mass bins which would require c8 to differ from 1 (as is found for the fit to D8) and that the c8*c6 interference driving the best fit value to the true value of 1. It is reflected in the other cases as well, such as 8 TeV with beta=3 where the error approximately doubles. It is also present in the beta=3 cases in the tables in the appendix. It is less pronounced for larger masses where the statistics are worse and the fit is better able to accommodate the divergence in the high invariant mass bins.

2) I have added these references to the short discussion of loop calculations of the Drell Yan process at dimension-six. As mentioned in the text (p15) including one-loop results in the IR would require the inclusion of one-loop effects in the UV pseudodata which is beyond the scope of this work (but very much of interest in follow up work).

3) Adding additional distributions is another important aspect I hope to address in follow up projects. I added a short discussion of this at the end of the conclusions again citing the two papers suggested in point 2:

"The inclusion of other distributions may have an interesting impact on the interpretation of higher dimension operators as nuisance parameters as at a given order they may not be able to as readily absorb affects such as angular distributions."

4) I have fixed this mistake as well as including 1/Lambda^4 in Eq15 which was missing.

Requested Changes: 1) I have merged the three plots into a single plot. 2) I have updated "R"->"Ratio to SM" in Figure 1. In Figures 3 and 4 "R"->"Ratio to full UV Model" 3) I have included the citations as mentioned in the above.

Report 2: I thank the referee for their report. While many of these results are expected they have never been clearly explored and written up in this manner of treating the UV model as pseudodata and exploring how a fit to that data behaves order by order. The topics of truncation discussed in this article are the subject of (frequently heated) debate within the community and I believe that clarifying these discussions with concrete examples in the literature is important to helping resolve these differences in the community. The work also establishes the next steps toward further work which will continue to inform the community on these topics.

Report 1: I thank the referee for their report and their careful reading of the text. I have corrected all of the typos mentioned in the report.

---

## Round 2 · List of Changes

As outlined in Author comments.

---

## Editorial Decision

published